# PROVABLE AND PRACTICAL: EFFICIENT EXPLORATION IN REINFORCEMENT LEARNING VIA LANGEVIN MONTE CARLO

**Haque Ishfaq**[*]
Mila, McGill University
haque.ishfaq@mail.mcgill.ca

**Qingfeng Lan**[*]
University of Alberta, Amii
qlan3@ualberta.ca

**Pan Xu**
Duke University

**A. Rupam Mahmood**
University of Alberta
CIFAR AI Chair, Amii

**Doina Precup**
Mila, McGill University
Google DeepMind

**Anima Anandkumar**
California Institute of Technology, Nvidia

**Kamyar Azizzadenesheli**
Nvidia

## ABSTRACT

We present a scalable and effective exploration strategy based on Thompson sampling for reinforcement learning (RL). One of the key shortcomings of existing Thompson sampling algorithms is the need to perform a Gaussian approximation of the posterior distribution, which is not a good surrogate in most practical settings. We instead directly sample the Q function from its posterior distribution, by using Langevin Monte Carlo, an efficient type of Markov Chain Monte Carlo (MCMC) method. Our method only needs to perform noisy gradient descent updates to learn the exact posterior distribution of the Q function, which makes our approach easy to deploy in deep RL. We provide a rigorous theoretical analysis for the proposed method and demonstrate that, in the linear Markov decision process (linear MDP) setting, it has a regret bound of $\widetilde{O}(d^{3/2}H^{3/2}\sqrt{T})$, where $d$ is the dimension of the feature mapping, $H$ is the planning horizon, and $T$ is the total number of steps. We apply this approach to deep RL, by using Adam optimizer to perform gradient updates. Our approach achieves better or similar results compared with state-of-the-art deep RL algorithms on several challenging exploration tasks from the Atari57 suite.[1]

## 1 INTRODUCTION

Balancing exploration with exploitation is a fundamental problem in reinforcement learning (RL) (Sutton and Barto, 2018). Numerous exploration algorithms have been proposed (Jaksch et al., 2010; Osband and Van Roy, 2017; Ostrovski et al., 2017; Azizzadenesheli et al., 2018; Jin et al., 2018). However, there is a big discrepancy between provably efficient algorithms, which are typically limited to tabular or linear MDPs with a focus on achieving tighter regret bound, and more heuristic-based algorithms for exploration in deep RL, which scale well but have no guarantees.

A generic and widely used solution to the exploration-exploitation dilemma is the use of optimism in the face of uncertainty (OFU) (Auer et al., 2002). Most works of this type inject optimism through bonuses added to the rewards or estimated Q functions (Jaksct et al., 2010; Azar et al., 2017; Jin et al., 2018; 2020). These bonuses, which are typically decreasing functions of counts on the number of visits of state-action pairs, allow the agent to build upper confidence bounds (UCBs) on the optimal Q functions and act greedily with respect to them. While UCB-based methods provide strong theoretical guarantees in tabular and linear settings, they often perform poorly in practice

---

[*]Equal contribution

[1]Our code is available at https://github.com/hmishfaq/LMC-LSVI

(Osband et al., 2013; Osband and Van Roy, 2017). Generalizations to non-tabular and non-linear settings have also been explored (Bellemare et al., 2016; Tang et al., 2017; Ostrovski et al., 2017; Burda et al., 2018).

Inspired by the well-known Thompson sampling (Thompson, 1933) for multi-armed bandits, another line of work proposes posterior sampling for RL (PSRL) (Osband et al., 2013; Agrawal and Jia, 2017), which maintains a posterior distribution over the MDP model parameters of the problem at hand. At the beginning of each episode, PSRL samples new parameters from this posterior, solves the sampled MDP, and follows its optimal policy until the end of the episode. However, generating exact posterior samples is only tractable in simple environments, such as tabular MDPs where Dirichlet priors can be used over transition probability distribution. Another closely related algorithm is randomized least-square value iteration (RLSVI), which induces exploration through noisy value iteration (Osband et al., 2016a; Russo, 2019; Ishfaq et al., 2021). Concretely, Gaussian noise is added to the reward before applying the Bellman update. This results in a Q function estimate that is equal to an empirical Bellman update with added Gaussian noise, which can be seen as approximating the posterior distribution of the Q function using a Gaussian distribution. However, in practical problems, Gaussian distributions may not be a good approximation of the true posterior of the Q function. Moreover, choosing an appropriate variance is an onerous task; and unless the features are fixed, the incremental computation of the posterior distribution is not possible.

Algorithms based on Langevin dynamics are widely used for training neural networks in Bayesian settings (Welling and Teh, 2011). For instance, by adding a small amount of exogenous noise, Langevin Monte Carlo (LMC) provides regularization and allows quantifying the degree of uncertainty on the parameters of the function approximator. Furthermore, the celebrated stochastic gradient descent, resembles a Langevin process (Cheng et al., 2020). Despite its huge influence in Bayesian deep learning, the application of LMC in sequential decision making problems is relatively unexplored. Mazumdar et al. (2020) proposed an LMC-based approximate Thompson sampling algorithm that achieves optimal instance-dependent regret for the multi-armed bandit (MAB) problem. Recently, Xu et al. (2022) used LMC to approximately sample model parameters from the posterior distribution in contextual bandits and showed that their approach can achieve the same regret bound as the best Thompson sampling algorithms for linear contextual bandits. Motivated by the success of the LMC approach in bandit problems, in this paper, we study the use of LMC to approximate the posterior distribution of the Q function, and thus provide an exploration approach which is principled, maintains the simplicity and scalability of LMC, and can be easily applied in deep RL algorithms.

**Main contributions.** We propose a practical and efficient online RL algorithm, Langevin Monte Carlo Least-Squares Value Iteration (LMC-LSVI), which simply performs noisy gradient descent updates to induce exploration. LMC-LSVI is easily implementable and can be used in high-dimensional RL tasks, such as image-based control. Along with providing empirical evaluation in the RiverSwim environment and simulated linear MDPs, we prove that LMC-LSVI achieves a $\widetilde{O}(d^{3/2}H^{3/2}\sqrt{T})$ regret in the linear MDP setting, where $d$ is the dimension of the feature mapping, $H$ is the planning horizon, and $T$ is the total number of steps. This bound provides the best possible dependency on $d$ and $H$ for any known randomized algorithms and achieves sublinear regret in $T$.

Because preconditioned Langevin algorithms (Li et al., 2016) can avoid pathological curvature problems and saddle points in the optimization landscape, we also propose Adam Langevin Monte Carlo Deep Q-Network (Adam LMCDQN), a preconditioned variant of LMC-LSVI based on the Adam optimizer (Kingma and Ba, 2014). In experiments on both $N$-chain (Osband et al., 2016b) and challenging Atari environments (Bellemare et al., 2013) that require deep exploration, Adam LMCDQN performs similarly or better than state-of-the-art exploration approaches in deep RL.

Unlike many other provably efficient algorithms with function approximations (Yang and Wang, 2020; Cai et al., 2020; Zanette et al., 2020a; Xu and Gu, 2020; Wu et al., 2020; Ayoub et al., 2020; Zanette et al., 2020b; Zhou et al., 2021; He et al., 2023), LMC-LSVI can easily be extended to deep RL settings (Adam LMCDQN). We emphasize that such unification of theory and practice is rare (Feng et al., 2021; Kitamura et al., 2023; Liu et al., 2023) in the current literature of both theoretical RL and deep RL.

## 2 PRELIMINARY

**Notation.** For any positive integer $n$, we denote the set $\{1, 2, \ldots, n\}$ by $[n]$. For any set $A$, $\langle \cdot, \cdot \rangle_A$ denotes the inner product over set $A$. $\odot$ and $\oslash$ represent element-wise vector product and division respectively. For function growth, we use $\widetilde{\mathcal{O}}(\cdot)$, ignoring poly-logarithmic factors.

We consider an episodic discrete-time Markov decision process (MDP) of the form $(\mathcal{S}, \mathcal{A}, H, \mathbb{P}, r)$ where $\mathcal{S}$ is the state space, $\mathcal{A}$ is the action space, $H$ is the episode length, $\mathbb{P} = \{\mathbb{P}_h\}_{h=1}^H$ are the state transition probability distributions, and $r = \{r_h\}_{h=1}^H$ are the reward functions. Moreover, for each $h \in [H]$, $\mathbb{P}_h(\cdot \mid x, a)$ denotes the transition kernel at step $h \in [H]$, which defines a non-stationary environment. $r_h : \mathcal{S} \times \mathcal{A} \to [0, 1]$ is the deterministic reward function at step $h$.[2] A policy $\pi$ is a collection of $H$ functions $\{\pi_h : \mathcal{S} \to \mathcal{A}\}_{h \in [H]}$ where $\pi_h(x)$ is the action that the agent takes in state $x$ at the $h$-th step in the episode. Moreover, for each $h \in [H]$, we define the value function $V_h^\pi : \mathcal{S} \to \mathbb{R}$ as the expected value of cumulative rewards received under policy $\pi$ when starting from an arbitrary state $x_h = x$ at the $h$-th time step. In particular, we have

$$V_h^\pi(x) = \mathbb{E}_\pi \left[ \sum_{h'=h}^H r_{h'}(x_{h'}, a_{h'}) \mid x_h = x \right].$$

Similarly, we define the action-value function (or the Q function) $Q_h^\pi : \mathcal{S} \times \mathcal{A} \to \mathbb{R}$ as the expected value of cumulative rewards given the current state and action where the agent follows policy $\pi$ afterwards. Concretely,

$$Q_h^\pi(x, a) = \mathbb{E}_\pi \left[ \sum_{h'=h}^H r_{h'}(x_{h'}, a_{h'}) \mid x_h = x, a_h = a \right].$$

We denote $V_h^*(x) = V_h^{\pi^*}(x)$ and $Q_h^*(x, a) = Q_h^{\pi^*}(x, a)$ where $\pi^*$ is the optimal policy. To simplify notation, we denote $[\mathbb{P}_h V_{h+1}](x, a) = \mathbb{E}_{x' \sim \mathbb{P}_h(\cdot \mid x, a)} V_{h+1}(x')$. Thus, we write the Bellman equation associated with a policy $\pi$ as

$$Q_h^\pi(x, a) = (r_h + \mathbb{P}_h V_{h+1}^\pi)(x, a), \qquad V_h^\pi(x) = Q_h^\pi(x, \pi_h(x)), \qquad V_{H+1}^\pi(x) = 0. \quad (1)$$

Similarly, the Bellman optimality equation is

$$Q_h^*(x, a) = (r_h + \mathbb{P}_h V_{h+1}^*)(x, a), \qquad V_h^*(x) = Q_h^*(x, \pi_h^*(x)), \qquad V_{H+1}^*(x) = 0. \quad (2)$$

The agent interacts with the environment for $K$ episodes with the aim of learning the optimal policy. At the beginning of each episode $k$, an adversary picks the initial state $x_1^k$, and the agent chooses a policy $\pi^k$. We measure the suboptimality of an agent by the total regret defined as

$$\text{Regret}(K) = \sum_{k=1}^K \left[ V_1^*(x_1^k) - V_1^{\pi^k}(x_1^k) \right].$$

**Langevin Monte Carlo (LMC).** LMC is an iterative algorithm (Rossky et al., 1978; Roberts and Stramer, 2002; Neal et al., 2011), which adds isotropic Gaussian noise to the gradient descent update at each step:

$$w_{k+1} = w_k - \eta_k \nabla L(w_k) + \sqrt{2\eta_k \beta^{-1}} \epsilon_k, \quad (3)$$

where $L(w)$ is the objective function, $\eta_k$ is the step-size parameter, $\beta$ is the inverse temperature parameter, and $\epsilon_k$ is an isotropic Gaussian random vector in $\mathbb{R}^d$. Under certain assumptions, the LMC update will generate a Markov chain whose distribution converges to a target distribution $\propto \exp(-\beta L(w))$ (Roberts and Tweedie, 1996; Bakry et al., 2014). In practice, one can also replace the true gradient $\nabla L(w_k)$ with some stochastic gradient estimators, resulting in the famous stochastic gradient Langevin dynamics (SGLD) (Welling and Teh, 2011) algorithm.

## 3 LANGEVIN MONTE CARLO FOR REINFORCEMENT LEARNING

In this section, we propose Langevin Monte Carlo Least-Squares Value Iteration (LMC-LSVI), as shown in Algorithm 1. Assume we have collected data trajectories in the first $k - 1$ episodes as $\{(x_1^\tau, a_1^\tau, r(x_1^\tau, a_1^\tau), \ldots, x_H^\tau, a_H^\tau, r(x_H^\tau, a_H^\tau))\}_{\tau=1}^{k-1}$. To estimate the Q function for stage $h$ at the $k$-th episode of the learning process, we define the following loss function:

$$L_h^k(w_h) = \sum_{\tau=1}^{k-1} \left[ r_h(x_h^\tau, a_h^\tau) + \max_{a \in \mathcal{A}} Q_{h+1}^k(x_{h+1}^\tau, a) - Q(w_h; \phi(x_h^\tau, a_h^\tau)) \right]^2 + \lambda \|w_h\|^2, \quad (4)$$

---

[2]We study the deterministic reward functions for notational simplicity. Our results can be easily generalized to the case when rewards are stochastic.

where $\phi(\cdot, \cdot)$ is a feature vector of the corresponding state-action pair and $Q(w_h; \phi(x_h^\tau, a_h^\tau))$ denotes any possible approximation of the Q function that is parameterized by $w_h$ and takes $\phi(x_h^\tau, a_h^\tau)$ as input. At stage $h$, we perform noisy gradient descent on $L_h^k(\cdot)$ for $J_k$ times as shown in Algorithm 1, where $J_k$ is also referred to as the update number for episode $k$. Note that the LMC-LSVI algorithm displayed here is a generic one, which works for all types of function approximation of the Q function. Similar to the specification of Langevin Monte Carlo Thompson Sampling (LMCTS) to linear bandits, generalized linear bandits, and neural contextual bandits (Xu et al., 2022), we can also derive different variants of LMC-LSVI for different types of function approximations by replacing the functions $Q(w_h; \phi(x_h^\tau, a_h^\tau))$ and the loss function $L_h^k(w_h)$.

In this paper, we will derive the theoretical analysis of LMC-LSVI under linear function approximations. In particular, when the function approximation of the Q function is linear, the model approximation of the Q function, denoted by $Q_h^k$ in Line 11 of Algorithm 1 becomes

$$Q_h^k(\cdot, \cdot) \leftarrow \min\{\phi(\cdot, \cdot)^\top w_h^{k, J_k}, H - h + 1\}^+. \tag{5}$$

Denoting $V_{h+1}^k(\cdot) = \max_{a \in \mathcal{A}} Q_{h+1}^k(\cdot, a)$, we have $\nabla L_h^k(w_h) = 2(\Lambda_h^k w_h - b_h^k)$, where

$$\Lambda_h^k = \sum_{\tau=1}^{k-1} \phi(x_h^\tau, a_h^\tau) \phi(x_h^\tau, a_h^\tau)^\top + \lambda I \text{ and } b_h^k = \sum_{\tau=1}^{k-1} \left[ r_h(x_h^\tau, a_h^\tau) + V_{h+1}^k(x_{h+1}^\tau) \right] \phi(x_h^\tau, a_h^\tau). \tag{6}$$

By setting $\nabla L_h^k(w_h) = 0$, we get the minimizer of $L_h^k$ as $\widehat{w}_h^k = (\Lambda_h^k)^{-1} b_h^k$.

We can prove that the iterate $w_h^{k, J_k}$ in Equation (5) follows the following Gaussian distribution.

**Proposition 3.1.** *The parameter $w_h^{k, J_k}$ used in episode $k$ of Algorithm 1 follows a Gaussian distribution $\mathcal{N}(\mu_h^{k, J_k}, \Sigma_h^{k, J_k})$, with mean and covariance matrix:*

$$\mu_h^{k, J_k} = A_k^{J_k} \ldots A_1^{J_1} w_h^{1,0} + \sum_{i=1}^{k} A_k^{J_k} \ldots A_{i+1}^{J_{i+1}} \left( I - A_i^{J_i} \right) \widehat{w}_h^i,$$

$$\Sigma_h^{k, J_k} = \sum_{i=1}^{k} \frac{1}{\beta_i} A_k^{J_k} \ldots A_{i+1}^{J_{i+1}} \left( I - A_i^{2J_i} \right) \left( \Lambda_h^i \right)^{-1} \left( I + A_i \right)^{-1} A_{i+1}^{J_{i+1}} \ldots A_k^{J_k},$$

*where $A_i = I - 2\eta_i \Lambda_h^i$ for $i \in [k]$.*

Proposition 3.1 shows that in linear setting the parameter $w_h^{k, J_k}$ follows a tractable distribution. This allows us to provide a high probability bound for the parameter $w_h^{k, J_k}$ in Lemma B.3, which is then used in Lemma B.7 to show that the estimated $Q_h^k$ function is optimistic with high probability.

We note that the parameter update in Algorithm 1 is presented as a full gradient descent step plus an isotropic noise for the purpose of theoretical analysis in Section 4. However, in practice, one can use a stochastic gradient (Welling and Teh, 2011; Zou et al., 2021) or a variance-reduced stochastic gradient (Dubey et al., 2016; Xu et al., 2018; Zou et al., 2018; 2019) of the loss function $L_h^k(w_h^{k, j-1})$ to improve the sample efficiency of LMC-LSVI.

## 4 THEORETICAL ANALYSIS

We now provide a regret analysis of LMC-LSVI under the linear MDP setting (Jin et al., 2020; Yang and Wang, 2020; 2019). First, we formally define a linear MDP.

**Definition 4.1** (Linear MDP). *A linear MDP is an MDP $(\mathcal{S}, \mathcal{A}, H, \mathbb{P}, r)$ with a feature $\phi : \mathcal{S} \times \mathcal{A} \to \mathbb{R}^d$, if for any $h \in [H]$, there exist $d$ unknown (signed) measures $\mu_h = (\mu_h^{(1)}, \mu_h^{(1)}, \ldots, \mu_h^{(d)})$ over $\mathcal{S}$ and an unknown vector $\theta_h \in \mathbb{R}^d$, such that for any $(x, a) \in \mathcal{S} \times \mathcal{A}$, we have*

$$\mathbb{P}_h(\cdot \mid x, a) = \langle \phi(x, a), \mu_h(\cdot) \rangle \text{ and } r_h(x, a) = \langle \phi(x, a), \theta_h \rangle.$$

Without loss of generality, we assume $\|\phi(x, a)\|_2 \leq 1$ for all $(x, a) \in \mathcal{S} \times \mathcal{A}$, and $\max\{\|\mu_h(\mathcal{S})\|_2, \|\theta_h\|_2\} \leq \sqrt{d}$ for all $h \in [H]$.

---

**Algorithm 1** Langevin Monte Carlo Least-Squares Value Iteration (LMC-LSVI )

---

1: Input: step sizes $\{\eta_k > 0\}_{k \geq 1}$, inverse temperature $\{\beta_k\}_{k \geq 1}$, loss function $L_k(w)$
2: Initialize $w_h^{1,0} = \mathbf{0}$ for $h \in [H]$, $J_0 = 0$
3: **for** episode $k = 1, 2, \ldots, K$ **do**
4:     Receive the initial state $s_1^k$
5:     **for** step $h = H, H-1, \ldots, 1$ **do**
6:         $w_h^{k,0} = w_h^{k-1, J_{k-1}}$
7:         **for** $j = 1, \ldots, J_k$ **do**
8:             $\epsilon_h^{k,j} \sim \mathcal{N}(0, I)$
9:             $w_h^{k,j} = w_h^{k,j-1} - \eta_k \nabla L_h^k(w_h^{k,j-1}) + \sqrt{2\eta_k \beta_k^{-1}} \epsilon_h^{k,j}$
10:         **end for**
11:         $Q_h^k(\cdot, \cdot) \leftarrow \min\{Q(w_h^{k,J_k}; \phi(\cdot, \cdot)), H - h + 1\}^+$
12:         $V_h^k(\cdot) \leftarrow \max_{a \in \mathcal{A}} Q_h^k(\cdot, a)$
13:     **end for**
14:     **for** step $h = 1, 2, \ldots, H$ **do**
15:         Take action $a_h^k \leftarrow \operatorname{argmax}_{a \in \mathcal{A}} Q_h^k(s_h^k, a)$, observe reward $r_h^k(s_h^k, a_h^k)$ and next state $s_{h+1}^k$
16:     **end for**
17: **end for**

---

We refer the readers to Wang et al. (2020), Lattimore et al. (2020), and Van Roy and Dong (2019) for related discussions on such a linear representation. Next, we introduce our main theorem.

**Theorem 4.2.** *Let $\lambda = 1$ in Equation (4), $\frac{1}{\sqrt{\beta_k}} = \widetilde{O}(H\sqrt{d})$ in Algorithm 1, and $\delta \in (\frac{1}{2\sqrt{2e\pi}}, 1)$. For any $k \in [K]$, let the learning rate $\eta_k = 1/(4\lambda_{\max}(\Lambda_h^k))$, the update number $J_k = 2\kappa_k \log(4HKd)$ where $\kappa_k = \lambda_{\max}(\Lambda_h^k)/\lambda_{\min}(\Lambda_h^k)$ is the condition number of $\Lambda_h^k$. Under Definition 4.1, the regret of Algorithm 1 satisfies*

$$Regret(K) = \widetilde{O}(d^{3/2}H^{3/2}\sqrt{T}),$$

*with probability at least $1 - \delta$.*

We compare the regret bound of our algorithm with the state-of-the-art results in the literature of theoretical reinforcement learning in Table 1. Compared to the lower bound $\Omega(dH\sqrt{T})$ proved in Zhou et al. (2021), our regret bound is worse off by a factor of $\sqrt{dH}$ under the linear MDP setting. However, the gap of $\sqrt{d}$ in worst-case regret between UCB and TS-based methods is a long standing open problem, even in a simpler setting of linear bandit (Hamidi and Bayati, 2020). When converted to linear bandits by setting $H = 1$, our regret bound matches that of LMCTS (Xu et al., 2022) and the best-known regret upper bound for LinTS from Agrawal and Goyal (2013) and Abeille and Lazaric (2017).

*Remark* 4.3. In Theorem 4.2, we require that the failure probability $\delta > \frac{1}{2\sqrt{2e\pi}}$. However, in frequentist regret analysis it is desirable that the regret bound holds for arbitrarily small failure probability. This arises from Lemma B.7, where we get an optimistic estimation with a constant probability. However, this result can be improved by using optimistic reward sampling scheme proposed in Ishfaq et al. (2021). Concretely, we can generate $M$ estimates for Q function $\{Q_h^{k,m}\}_{m \in [M]}$ through maintaining $M$ samples of $w$: $\{w_h^{k,J_k,m}\}_{m \in [M]}$. Then, we can make an optimistic estimate of Q function by setting $Q_h^k(\cdot, \cdot) = \min\{\max_{m \in [M]}\{Q_h^{k,m}(\cdot, \cdot)\}, H - h + 1\}$. We provide the regret analysis for this approach, whose proof essentially follows the same steps as in that of Theorem 4.2, in Appendix D. However, for the simplicity of the algorithm design, we use the currently proposed algorithm.

## 5   DEEP Q-NETWORK WITH LMC EXPLORATION

In this section, we investigate the case where deep Q-networks (DQNs) (Mnih et al., 2015) are used, which is used as the backbone of many deep RL algorithms and prevalent in real-world RL applications due to its scalability and implementation ease.

Table 1: Regret upper bound for episodic, non-stationary, linear MDPs. Here, computational tractability refers to the ability of a computational problem to be solved in a reasonable amount of time using a feasible amount of computational resources.

| Algorithm | Regret | Exploration | Computational Tractability |
|---|---|---|---|
| LSVI-UCB (Jin et al., 2020) | $\widetilde{\mathcal{O}}(d^{3/2}H^{3/2}\sqrt{T})$ | UCB | Yes |
| OPT-RLSVI (Zanette et al., 2020a) | $\widetilde{\mathcal{O}}(d^2H^2\sqrt{T})$ | TS | Yes |
| ELEANOR (Zanette et al., 2020b) | $\widetilde{\mathcal{O}}(dH^{3/2}\sqrt{T})$ | Optimism | No |
| LSVI-PHE (Ishfaq et al., 2021) | $\widetilde{\mathcal{O}}(d^{3/2}H^{3/2}\sqrt{T})$ | TS | Yes |
| LMC-LSVI (this paper) | $\widetilde{\mathcal{O}}(d^{3/2}H^{3/2}\sqrt{T})$ | LMC | Yes |

While LMC and SGLD have been shown to converge to the true posterior under idealized settings (Chen et al., 2015; Teh et al., 2016; Dalalyan, 2017), in practice, most deep neural networks often exhibit pathological curvature and saddle points (Dauphin et al., 2014), which render the first-order gradient-based algorithms inefficient, such as SGLD. To mitigate this issue, Li et al. (2016) proposed RMSprop (Tieleman et al., 2012) based preconditioned SGLD. Similarly, Kim et al. (2022) proposed Adam based adaptive SGLD algorithm, where an adaptively adjusted bias term is included in the drift function to enhance escape from saddle points and accelerate the convergence in the presence of pathological curvatures.

Similarly, in sequential decision problems, there have been studies that show that deep RL algorithms suffer from training instability due to the usage of deep neural networks (Sinha et al., 2020; Ota et al., 2021; Sullivan et al., 2022). Henderson et al. (2018) empirically analyzed the effects of different adaptive gradient descent optimizers on the performance of deep RL algorithms and suggest that while being sensitive to the learning rate, RMSProp or Adam (Kingma and Ba, 2014) provides the best performance overall. Moreover, even though the original DQN algorithm (Mnih et al., 2015) used RMSProp optimizer with Huber loss, Ceron and Castro (2021) showed that Adam optimizer with mean-squared error (MSE) loss provides overwhelmingly superior performance.

Motivated by these developments both in the sampling community and the deep RL community, we now endow DQN-style algorithms (Mnih et al., 2015) with Langevin Monte Carlo. In particular, we propose Adam Langevin Monte Carlo Deep Q-Network (Adam LMCDQN) in Algorithm 2, where we replace LMC in Algorithm 1 with the Adam SGLD (aSGLD) (Kim et al., 2022) algorithm in learning the posterior distribution.

In Algorithm 2, $\nabla\widetilde{L}_h^k(w)$ denotes an estimate of $\nabla L_h^k(w)$ based on one mini-batch of data sampled from the replay buffer. $\alpha_1$ and $\alpha_2$ are smoothing factors for the first and second moments of stochastic gradients, respectively. $a$ is the bias factor and $\lambda_1$ is a small constant added to avoid zero-divisors. Here, $v_h^{k,j}$ can be viewed as an approximator of the true second-moment matrix $\mathbb{E}(\nabla\widetilde{L}_h^k(w_h^{k,j-1})\nabla\widetilde{L}_h^k(w_h^{k,j-1})^\top)$ and the bias term $m_h^{k,j-1} \oslash \sqrt{v_h^{k,j-1} + \lambda_1\mathbf{1}}$ can be viewed as the rescaled momentum which is isotropic near stationary points. Similar to Adam, the bias term, with an appropriate choice of the bias factor $a$, is expected to guide the sampler to converge to a global optimal region quickly.

## 6 EXPERIMENTS

In this section, we present an empirical evaluation of Adam LMCDQN. For the empirical evaluation of LMC-LSVI in the RiverSwim environment (Strehl and Littman, 2008; Osband et al., 2013) and simulated linear MDPs, we refer the reader to Appendix F.1. First, we consider a hard exploration problem and demonstrate the ability of deep exploration for our algorithm. We then proceed to experiments with 8 hard Atari games, showing that Adam LMCDQN is able to outperform several strong baselines. Note that for implementation simplicity, in the following experiments, we set all the update numbers $J_k$ and the inverse temperature values $\beta_k$ to be the same number for all $k \in [K]$. We also emphasize that even though in Theorem 4.2, we specify a theoretical value for $J_k$, as we show in this section, in practice, a small value for $J_k$ (we use $J_k = 4$ for N-Chain and $J_k = 1$ for

---

**Algorithm 2** Adam LMCDQN

1: Input: step sizes $\{\eta_k > 0\}_{k \geq 1}$, inverse temperature $\{\beta_k\}_{k \geq 1}$, smoothing factors $\alpha_1$ and $\alpha_2$, bias factor $a$, loss function $L_k(w)$.
2: Initialize $w_h^{1,0}$ from appropriate distribution for $h \in [H]$, $J_0 = 0$, $m_h^{1,0} = 0$ and $v_h^{1,0} = 0$ for $h \in [H]$ and $k \in [K]$.
3: **for** episode $k = 1, 2, \ldots, K$ **do**
4:     Receive the initial state $s_1^k$.
5:     **for** step $h = H, H-1, \ldots, 1$ **do**
6:         $w_h^{k,0} = w_h^{k-1,J_{k-1}}, m_h^{k,0} = m_h^{k-1,J_{k-1}}, v_h^{k,0} = v_h^{k-1,J_{k-1}}$
7:         **for** $j = 1, \ldots, J_k$ **do**
8:             $\epsilon_h^{k,j} \sim \mathcal{N}(0, I)$
9:             $w_h^{k,j} = w_h^{k,j-1} - \eta_k \left( \nabla \widetilde{L}_h^k(w_h^{k,j-1}) + am_h^{k,j-1} \oslash \sqrt{v_h^{k,j-1} + \lambda_1 \mathbf{1}} \right) + \sqrt{2\eta_k \beta_k^{-1}} \epsilon_h^{k,j}$
10:            $m_h^{k,j} = \alpha_1 m_h^{k,j-1} + (1 - \alpha_1) \nabla \widetilde{L}_h^k(w_h^{k,j-1})$
11:            $v_h^{k,j} = \alpha_2 v_h^{k,j-1} + (1 - \alpha_2) \nabla \widetilde{L}_h^k(w_h^{k,j-1}) \odot \nabla \widetilde{L}_h^k(w_h^{k,j-1})$
12:        **end for**
13:        $Q_h^k(\cdot, \cdot) \leftarrow Q(w_h^{k,J_k}; \phi(\cdot, \cdot))$
14:        $V_h^k(\cdot) \leftarrow \max_{a \in \mathcal{A}} Q_h^k(\cdot, a)$
15:    **end for**
16:    **for** step $h = 1, 2, \ldots, H$ **do**
17:        Take action $a_h^k \leftarrow \text{argmax}_{a \in \mathcal{A}} Q_h^k(s_h^k, a)$, observe reward $r_h^k(s_h^k, a_h^k)$ and next state $s_{h+1}^k$.
18:    **end for**
19: **end for**

---

Atari) can yield good performance for our algorithm. We also emphasize that in our implementation of Adam LMCDQN we use fixed values of $\alpha_1 = 0.9$, $\alpha_2 = 0.99$, and $\lambda_1 = 10^{-8}$ instead of tuning them.

*Remark* 6.1. We note that in our experiments in this section, as baselines, we use commonly used algorithms from deep RL literature as opposed to methods presented in Table 1. This is because while these methods are provably efficient under linear MDP settings, in most cases, it is not clear how to scale them to deep RL settings. More precisely, these methods assume that a good feature is known in advance and Q values can be approximated as a linear function over this feature. If the provided feature is not good and fixed, the empirical performance of these methods is often poor. For example, LSVI-UCB (Jin et al., 2020) computes UCB bonus function of the form $\|\phi(s, a)\|_{\Lambda^{-1}}$, where $\Lambda \in \mathbb{R}^{d \times d}$ is the empirical feature covariance matrix. When we update the feature over iterations in deep RL, the computational complexity of LSVI-UCB becomes unbearable as it needs to repeatedly compute the feature covariance matrix to update the bonus function. In the same vein, while estimating the Q function, OPT-RSLVI (Zanette et al., 2020a) needs to rely on the feature norm with respect to the inverse covariance matrix. Lastly, even though LSVI-PHE (Ishfaq et al., 2021) is computationally implementable in deep RL settings, it requires sampling independent and identically distributed (i.i.d.) noise for the whole history every time to perturb the reward, which appears to be computationally burdensome in most practical settings.

## 6.1 DEMONSTRATION OF DEEP EXPLORATION

We first conduct experiments in $N$-Chain (Osband et al., 2016b) to show that Adam LMCDQN is able to perform deep exploration. The environment consists of a chain of $N$ states, namely $s_1, s_2, \ldots, s_N$. The agent always starts in state $s_2$, from where it can either move left or right. The agent receives a small reward $r = 0.001$ in state $s_1$ and a larger reward $r = 1$ in state $s_N$. The horizon length is $N + 9$, so the optimal return is 10. Please refer to Appendix F.2 for a depiction of the environment.

In our experiments, we consider $N$ to be 25, 50, 75, or 100. For each chain length, we train different algorithms for $10^5$ steps across 20 seeds. We use DQN (Mnih et al., 2015), Bootstrapped DQN (Osband et al., 2016b) and Noisy-Net (Fortunato et al., 2017) as the baseline algorithms. We use DQN with $\epsilon$-greedy exploration strategy, where $\epsilon$ decays linearly from 1.0 to 0.01 for the first 1,000

training steps and then is fixed as $0.01$. For evaluation, we set $\epsilon = 0$ in DQN. We measure the performance of each algorithm in each run by the mean return of the last 10 evaluation episodes. For all algorithms, we sweep the learning rate and pick the one with the best performance. For Adam LMCDQN , we sweep $a$ and $\beta_k$ in small ranges. For more details, please check Appendix F.2.

In Figure 1, we show the performance of Adam LMCDQN and the baseline methods under different chain lengths. The solid lines represent the averaged return over 20 random seeds and the shaded areas represent standard errors. Note that for Adam LMCDQN , we set $J_k = 4$ for all chain lengths. As $N$ increases, the hardness of exploration increases, and Adam LMCDQN is able to maintain high performance while the performance of other baselines especially Bootstrapped DQN and Noisy-Net drop quickly. Clearly, Adam LMCDQN achieves significantly more robust performance than other baselines as $N$ increases, showing its deep exploration ability.

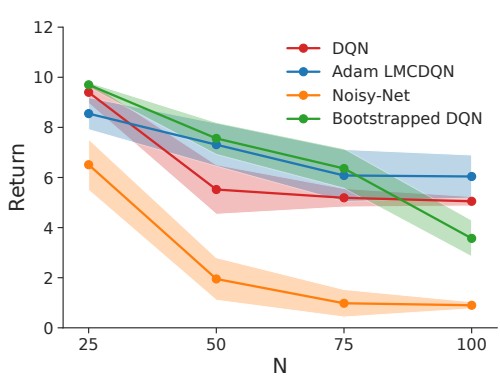

Figure 1: A comparison of Adam LMCDQN and other baselines in $N$-chain with different chain lengths $N$. All results are averaged over 20 runs and the shaded areas represent standard errors. As $N$ increases, the exploration hardness increases.

## 6.2 EVALUATION IN ATARI GAMES

To further evaluate our algorithm, we conduct experiments in Atari games (Bellemare et al., 2013). Specifically, 8 visually complicated hard exploration games (Taiga et al., 2019) are selected, including Alien, Freeway, Gravitar, H.E.R.O., Pitfall, Qbert, Solaris, and Venture. Among these games, Alien, H.E.R.O., and Qbert are dense reward environments, while Freeway, Gravitar, Pitfall, Solaris, and Venture are sparse reward environments, according to Taiga et al. (2019).

**Main Results.** We consider 7 baselines: Double DQN (Van Hasselt et al., 2016), Prioritized DQN (Schaul et al., 2015), C51 (Bellemare et al., 2017), QR-DQN (Dabney et al., 2018a), IQN (Dabney et al., 2018b), Bootstrapped DQN (Osband et al., 2016b) and Noisy-Net (Fortunato et al., 2017). Since a large $J_k$ greatly increases training time, we set $J_k = 1$ in Adam LMCDQN so that all experiments can be finished in a reasonable time. We also incorporate the double Q trick (Van Hasselt, 2010; Van Hasselt et al., 2016), which is shown to slightly boost performance. We train Adam LMCDQN for 50M frames (i.e., 12.5M steps) and summarize results across over 5 random seeds. Please check Appendix F.3.1 for more details about the training and hyper-parameters settings.

In Figure 2, we present the learning curves of all methods in 8 Atari games. The solid lines correspond to the median performance over 5 random seeds, while the shaded areas represent 90% confidence intervals. Overall, the results show that our algorithm Adam LMCDQN is quite competitive compared to the baseline algorithms. In particular, Adam LMCDQN exhibits a strong advantage against all other methods in Gravitar and Venture.

**Sensitivity Analysis.** In Figure 3a, we draw the learning curves of Adam LMCDQN with different bias factors $a$ in Qbert. The performance of our algorithm is greatly affected by the value of the bias factor. Overall, by setting $a = 0.1$, Adam LMCDQN achieves good performance in Qbert as well as in other Atari games. On the contrary, Adam LMCDQN is less sensitive to the inverse temperature $\beta_k$, as shown in Figure 3b.

**Ablation Study.** In Appendix F.3.2, we also present results for Adam LMCDQN without applying double Q functions. The performance of Adam LMCDQN is only slightly worse without using double Q functions, proving the effectiveness of our approach. Moreover, we implement Langevin DQN (Dwaracherla and Van Roy, 2020) with double Q functions and compare it with our algorithm Adam LMCDQN . Empirically, we observed that Adam LMCDQN usually outperforms Langevin DQN in sparse-reward hard-exploration games, while in dense-reward hard-exploration games, Adam LMCDQN and Langevin DQN achieve similar performance.

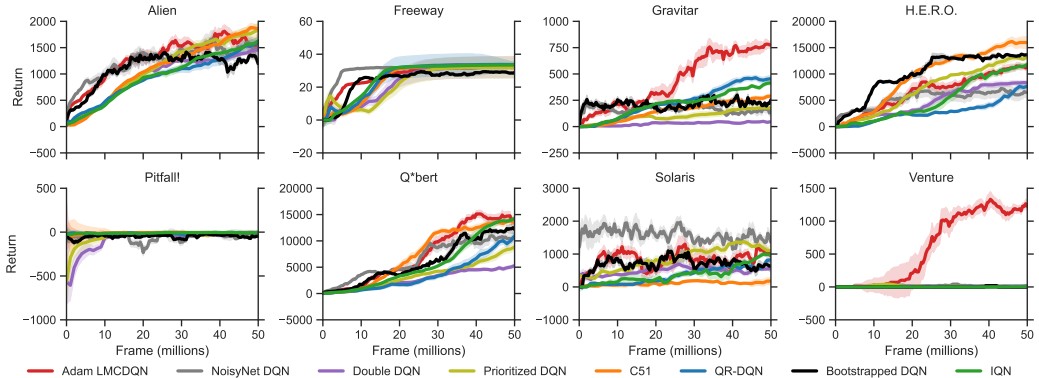

Figure 2: The return curves of various algorithms in eight Atari tasks over 50 million training frames. Solid lines correspond to the median performance over 5 random seeds, and the shaded areas correspond to 90% confidence interval.

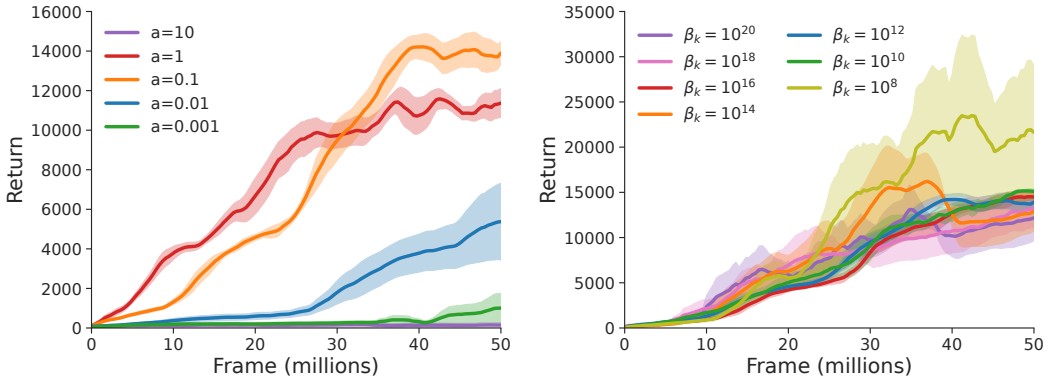

(a) Different bias factor $a$ in Adam LMCDQN

(b) Different temperatures $\beta_k$ in Adam LMCDQN

Figure 3: (a) A comparison of Adam LMCDQN with different bias factor $a$ in Qbert. Solid lines correspond to the average performance over 5 random seeds, and shaded areas correspond to standard errors. The performance of Adam LMCDQN is greatly affected by the value of the bias factor. (b) A comparison of Adam LMCDQN with different values of inverse temperature parameter $\beta_k$ in Qbert. Adam LMCDQN is not very sensitive to inverse temperature $\beta_k$.

## 7 CONCLUSION AND FUTURE WORK

We proposed the LMC-LSVI algorithm for reinforcement learning that uses Langevin Monte Carlo to directly sample a Q function from the posterior distribution with arbitrary precision. LMC-LSVI achieves the best-available regret bound for randomized algorithms in the linear MDP setting. Furthermore, we proposed Adam LMCDQN, a practical variant of LMC-LSVI, that demonstrates competitive empirical performance in challenging exploration tasks. There are several avenues for future research. It would be interesting to explore if one can improve the suboptimal dependence on $H$ for randomized algorithms. Extending the current results to more practical and general settings (Zhang et al., 2022; Ouhamma et al., 2023; Weisz et al., 2023) is also an exciting future direction. On the empirical side, it would be interesting to see whether LMC based approaches can be used in continuous control tasks for efficient exploration.

### ACKNOWLEDGMENTS

We gratefully acknowledge funding from the Canada CIFAR AI Chairs program, the Reinforcement Learning and Artificial Intelligence (RLAI) laboratory, the Alberta Machine Intelligence Institute (Amii), Mila - Quebec Artificial Intelligence Institute, the Natural Sciences and Engineering Re-

search Council (NSERC) of Canada, the US National Science Foundation (DMS-2323112) and the Whitehead Scholars Program at the Duke University School of Medicine. The authors would like to thank Vikranth Dwaracherla, Sehwan Kim, and Fabian Pedregosa for their helpful discussions. The authors would also like to thank Ziniu Li for giving great advice about Atari experiments.

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

CONTENTS

# A   RELATED WORK

**Posterior Sampling in Reinforcement Learning.** Our work is closely related to a line of work that uses posterior sampling, i.e., Thompson sampling in RL (Strens, 2000). Osband et al. (2016a), Russo (2019) and Xiong et al. (2022) propose randomized least-squares value iteration (RLSVI) with frequentist regret analysis under tabular MDP setting. RLSVI carefully injects tuned random noise into the value function in order to induce exploration. Recently, Zanette et al. (2020a) and Ishfaq et al. (2021) extended RLSVI to the linear setting. While RLSVI enjoys favorable regret bound under tabular and linear settings, it can only be applied when a good feature is known and fixed during training, making it impractical for deep RL (Li et al., 2021). Osband et al. (2016b; 2018) addressed this issue by training an ensemble of randomly initialized neural networks and viewing them as approximate posterior samples of Q functions. However, training an ensemble of neural networks is computationally prohibitive. Another line of work directly injects noise to parameters (Fortunato et al., 2017; Plappert et al., 2018). Noisy-Net (Fortunato et al., 2017) learns noisy parameters using gradient descent, whereas Plappert et al. (2018) added constant Gaussian noise to the parameters of the neural network. However, Noisy-Net is not ensured to approximate the posterior distribution (Fortunato et al., 2017). Proposed by Dwaracherla and Van Roy (2020), Langevin DQN is the closest algorithm to our work. Even though Langevin DQN is also inspired by SGLD (Welling and Teh, 2011), Dwaracherla and Van Roy (2020) did not provide any theoretical study nor regret bound for their algorithm under any setting. In a concurrent work, Kuang et al. (2024) studied linear MDP with delayed feedback, where they also used an LMC-based posterior sampling algorithm, which theoretically resembles similarity to part of our theoretical study with no deep RL variant studied.

A recently proposed model-free posterior sampling algorithm is Conditional Posterior Sampling (CPS) algorithm (Dann et al., 2021). Similar to Feel-Good Thompson Sampling proposed in (Zhang, 2022), CPS considers an opimistic prior term which helps with initial exploration. However, the proposed posterior formulation in CPS does not allow computationally tractable sampling. Another recently proposed algorithm Bayes-UCBVI (Tiapkin et al., 2022) uses the quantile of a Q-value function posterior as UCBs on the optimal Q-value functions which can be thought of as a deterministic version of PSRL (Osband et al., 2013). While Bayes-UCBVI is extendable to deep RL, it does not have any theory for the function approximation case and their analysis only works in the tabular setting. Even for Bayes-UCBVI to work in deep RL, it needs a well-chosen posterior, and sampling from the posterior is not addressed in their algorithm. Moreover, the Atari experiment in Tiapkin et al. (2022) only shows average comparison against Bootstrapped DQN (Osband et al., 2016b) and Double DQN (Van Hasselt et al., 2016) across all 57 games. However, in-average comparison across 57 games is not a sufficient demonstration of the utility of Bayes-UCBVI, especially when it comes to hard exploration tasks.

**Comparison to Dwaracherla and Van Roy (2020).** Here we provide a detailed comparison of Langevin DQN (Dwaracherla and Van Roy, 2020) and our work. On the algorithmic side, at each time step, Langevin DQN performs only one gradient update, while we perform multiple (i.e., $J_k$) noisy gradient updates, as shown in Algorithm 1 and Algorithm 2. This is a crucial difference as a large enough value for $J_k$ allows us to learn the exact posterior distribution of the parameters $\{w_h\}_{h \in [H]}$ up to high precision. Moreover, they also proposed to use preconditioned SGLD optimizer which is starkly different from our Adam LMCDQN. Their optimizer is more akin to a heuristic variant of the original Adam optimizer (Kingma and Ba, 2014) with a Gaussian noise term added to the gradient term. Moreover, they do not use any temperature parameter in the noise term. On the contrary, Adam LMCDQN is inspired by Adam SGLD (Kim et al., 2022), which enjoys convergence guarantees in the supervised learning setting. Overall, the design of Adam LMCDQN is heavily inspired from the theoretical study of LMC-LSVI and Adam SGLD (Kim et al., 2022). Lastly, while Dwaracherla and Van Roy (2020) provided some empirical study in the tabular deep sea environment (Osband et al., 2019a;b), they did not perform any experiment in challenging pixel-based environment (e.g., Atari). We conducted a comparison in such environments in Appendix F.3.2. Empirically, we observed that Adam LMCDQN usually outperforms Langevin DQN in sparse-reward hard-exploration games, such as Gravitar, Solaris, and Venture; while in dense-reward hard-exploration games such as Alien, H.E.R.O and Qbert, Adam LMCDQN and Langevin DQN achieve similar performance.

# B    PROOF OF THE REGRET BOUND OF LMC-LSVI

**Additional Notation.** For any set $A$, $\langle \cdot, \cdot \rangle_A$ denotes the inner product over set $A$. For a vector $x \in \mathbb{R}^d$, $\|x\|_2 = \sqrt{x^\top x}$ is the Euclidean norm of $x$. For a matrix $V \in \mathbb{R}^{m \times n}$, we denote the operator norm and Frobenius norm by $\|V\|_2$ and $\|V\|_F$ respectively. For a positive definite matrix $V \in \mathbb{R}^{d \times d}$ and a vector $x \in \mathbb{R}^d$, we denote $\|x\|_V = \sqrt{x^\top V x}$.

## B.1    SUPPORTING LEMMAS

Before deriving the regret bound of LMC-LSVI, we first outline the necessary technical lemmas that are helpful in our regret analysis. The first result below shows that the parameter obtained from LMC follows a Gaussian distribution.

**Proposition B.1.** *The parameter $w_h^{k,J_k}$ used in episode $k$ of Algorithm 1 follows a Gaussian distribution $\mathcal{N}(\mu_h^{k,J_k}, \Sigma_h^{k,J_k})$, where the mean vector and the covariance matrix are defined as*

$$\mu_h^{k,J_k} = A_k^{J_k} \dots A_1^{J_1} w_h^{1,0} + \sum_{i=1}^{k} A_k^{J_k} \dots A_{i+1}^{J_{i+1}} \left(I - A_i^{J_i}\right) \widehat{w}_h^i, \tag{7}$$

$$\Sigma_h^{k,J_k} = \sum_{i=1}^{k} \frac{1}{\beta_i} A_k^{J_k} \dots A_{i+1}^{J_{i+1}} \left(I - A_i^{2J_i}\right) \left(\Lambda_h^i\right)^{-1} \left(I + A_i\right)^{-1} A_{i+1}^{J_{i+1}} \dots A_k^{J_k}, \tag{8}$$

*where $A_i = I - 2\eta_i \Lambda_h^i$ for $i \in [k]$.*

**Definition B.2** (Model prediction error)**.** For any $(k, h) \in [K] \times [H]$, we define the model prediction error associated with the reward $r_h$,

$$l_h^k(x, a) = r_h(x, a) + \mathbb{P}_h V_{h+1}^k(x, a) - Q_h^k(x, a).$$

**Lemma B.3.** *Let $\lambda = 1$ in Algorithm 1. For any $(k, h) \in [K] \times [H]$, we have*

$$\left\| w_h^{k,J_k} \right\|_2 \leq \frac{16}{3} H d\sqrt{K} + \sqrt{\frac{2K}{3\beta_K \delta}} d^{3/2} := B_\delta,$$

*with probability at least $1 - \delta$.*

**Lemma B.4.** *Let $\lambda = 1$ in Algorithm 1. For any fixed $0 < \delta < 1$, with probability $1 - \delta$, we have for all $(k, h) \in [K] \times [H]$,*

$$\left\| \sum_{\tau=1}^{k-1} \phi(x_h^\tau, a_h^\tau) \left[ V_{h+1}^k(x_{h+1}^\tau) - \mathbb{P}_h V_{h+1}^k(x_h^\tau, a_h^\tau) \right] \right\|_{(\Lambda_h^k)^{-1}}$$

$$\leq 3H\sqrt{d} \left[ \frac{1}{2} \log(K+1) + \log\left( \frac{2\sqrt{2}K B_{\delta/2}}{H} \right) + \log\frac{2}{\delta} \right]^{1/2},$$

*where $B_\delta = \frac{16}{3} H d\sqrt{K} + \sqrt{\frac{2K}{3\beta_K \delta}} d^{3/2}$.*

**Lemma B.5.** *Let $\lambda = 1$ in Algorithm 1. Define the following event*

$$\mathcal{E}(K, H, \delta)$$

$$= \left\{ \left| \phi(x, a)^\top \widehat{w}_h^k - r_h(x, a) - \mathbb{P}_h V_{h+1}^k(x, a) \right| \leq 5H\sqrt{d} C_\delta \|\phi(x, a)\|_{(\Lambda_h^k)^{-1}}, \right.$$

$$\left. \forall (h, k) \in [H] \times [K] \text{ and } \forall (x, a) \in \mathcal{S} \times \mathcal{A} \right\}, \tag{9}$$

*where we denote $C_\delta = \left[ \frac{1}{2} \log(K+1) + \log\left( \frac{2\sqrt{2}K B_{\delta/2}}{H} \right) + \log\frac{2}{\delta} \right]^{1/2}$ and $B_\delta$ is defined in Lemma B.4. Then we have $\mathbb{P}(\mathcal{E}(K, H, \delta)) \geq 1 - \delta$.*

**Lemma B.6** (Error bound). *Let $\lambda = 1$ in Algorithm 1. For any $\delta \in (0,1)$ conditioned on the event $\mathcal{E}(K, H, \delta)$, for all $(h, k) \in [H] \times [K]$ and $(x, a) \in \mathcal{S} \times \mathcal{A}$, with probability at least $1 - \delta^2$, we have*

$$-l_h^k(x, a) \leq \left( 5H\sqrt{d}C_\delta + 5\sqrt{\frac{2d\log(1/\delta)}{3\beta_K}} + 4/3 \right) \|\phi(x, a)\|_{(\Lambda_h^k)^{-1}}, \qquad (10)$$

*where $C_\delta$ is defined in Lemma B.5.*

**Lemma B.7** (Optimism). *Let $\lambda = 1$ in Algorithm 1. Conditioned on the event $\mathcal{E}(K, H, \delta)$, for all $(h, k) \in [H] \times [K]$ and $(x, a) \in \mathcal{S} \times \mathcal{A}$, with probability at least $\frac{1}{2\sqrt{2e\pi}}$, we have*

$$l_h^k(x, a) \leq 0. \qquad (11)$$

## B.2 REGRET ANALYSIS

We first restate the main theorem as follows.

**Theorem B.8.** *Let $\lambda = 1$ in (4), $\frac{1}{\beta_k} = \widetilde{O}(H\sqrt{d})$ in Algorithm 1 and $\delta \in (\frac{1}{2\sqrt{2e\pi}}, 1)$. For any $k \in [K]$, let the learning rate $\eta_k = 1/(4\lambda_{\max}(\Lambda_h^k))$, the update number $J_k = 2\kappa_k \log(4HKd)$ where $\kappa_k = \lambda_{\max}(\Lambda_h^k)/\lambda_{\min}(\Lambda_h^k)$ is the condition number of $\Lambda_h^k$. Under Definition 4.1, the regret of Algorithm 1 satisfies*

$$Regret(K) = \widetilde{O}(d^{3/2}H^{3/2}\sqrt{T}),$$

*with probability at least $1 - \delta$.*

*Proof of Theorem B.8.* By Lemma 4.2 in Cai et al. (2020), it holds that

$$
\begin{aligned}
\text{Regret}(T) &= \sum_{k=1}^{K} \left( V_1^*(x_1^k) - V_1^{\pi^k}(x_1^k) \right) \\
&= \underbrace{\sum_{k=1}^{K}\sum_{t=1}^{H} \mathbb{E}_{\pi^*} \left[ \langle Q_h^k(x_h, \cdot), \pi_h^*(\cdot \mid x_h) - \pi_h^k(\cdot \mid x_h) \rangle \mid x_1 = x_1^k \right]}_{(i)} + \underbrace{\sum_{k=1}^{K}\sum_{t=1}^{H} \mathcal{D}_h^k}_{(ii)} \\
&\quad + \underbrace{\sum_{k=1}^{K}\sum_{t=1}^{H} \mathcal{M}_h^k}_{(iii)} + \underbrace{\sum_{k=1}^{K}\sum_{h=1}^{H} \left( \mathbb{E}_{\pi^*} \left[ l_h^k(x_h, a_h) \mid x_1 = x_1^k \right] - l_h^k(x_h^k, a_h^k) \right)}_{(iv)}, \qquad (12)
\end{aligned}
$$

where $\mathcal{D}_h^k$ and $\mathcal{M}_h^k$ are defined as

$$\mathcal{D}_h^k := \langle (Q_h^k - Q_h^{\pi^k})(x_h^k, \cdot), \pi_h^k(\cdot, x_h^k) \rangle - (Q_h^k - Q_h^{\pi^k})(x_h^k, a_h^k), \qquad (13)$$

$$\mathcal{M}_h^k := \mathbb{P}_h((V_{h+1}^k - V_{h+1}^{\pi^k}))(x_h^k, a_h^k) - (V_{h+1}^k - V_{h+1}^{\pi^k})(x_h^k). \qquad (14)$$

Next, we will bound the above terms respectively.

**Bounding Term (i):** For the policy $\pi_h^k$ at time step $h$ of episode $k$, we will prove that

$$\sum_{k=1}^{K}\sum_{h=1}^{H} \mathbb{E}_{\pi^*} [\langle Q_h^k(x_h, \cdot), \pi_h^*(\cdot \mid x_h) - \pi_h^k(\cdot \mid x_h) \rangle \mid x_1 = x_1^k] \leq 0. \qquad (15)$$

To this end, note that $\pi_h^k$ acts greedily with respect to action-value function $Q_h^k$. If $\pi_h^k = \pi_h^*$, then the difference $\pi_h^*(\cdot \mid x_h) - \pi_h^k(\cdot \mid x_h)$ is 0. Otherwise, the difference is negative since $\pi_h^k$ is deterministic with respect to $Q_h^k$. Concretely, $\pi_h^k$ takes a value of 1 where $\pi_h^*$ would take a value of 0. Moreover, $Q_h^k$ would have the greatest value at the state-action pair where $\pi_h^k$ equals 1. This completes the proof.

**Bounding Terms (ii) and (iii):** From (5), note that we truncate $Q_h^k$ to the range $[0, H - h + 1]$. This implies for any $(h, k) \in [K] \times [H]$, we have $|\mathcal{D}_h^k| \leq 2H$. Moreover, $\mathbb{E}[\mathcal{D}_h^k | \mathcal{F}_h^k] = 0$, where $\mathcal{F}_h^k$

is a corresponding filtration. Thus, $\mathcal{D}_h^k$ is a martingale difference sequence. So, applying Azuma-Hoeffding inequality, we have with probability $1 - \delta/3$,

$$\sum_{k=1}^{K}\sum_{h=1}^{H}\mathcal{D}_h^k \leq \sqrt{2H^2T\log(3/\delta)},$$

where $T = KH$. Similarly, we can show that $\mathcal{M}_h^k$ is a martingale difference sequence. Applying Azuma-Hoeffding inequality, we have with probability $1 - \delta/3$,

$$\sum_{k=1}^{K}\sum_{h=1}^{H}\mathcal{M}_h^k \leq \sqrt{2H^2T\log(3/\delta)}.$$

Therefore, by applying union bound, we have that for any $\delta > 0$, with probability $1 - 2\delta/3$, it holds that

$$\sum_{k=1}^{K}\sum_{h=1}^{H}\mathcal{D}_h^k + \sum_{k=1}^{K}\sum_{h=1}^{H}\mathcal{M}_h^k \leq 2\sqrt{2H^2T\log(3/\delta)}, \tag{16}$$

where $T = KH$.

**Bounding Term (iv):**

Suppose the event $\mathcal{E}(K, H, \delta')$ holds. by union bound, with probability $1 - (\delta'^2 + \frac{1}{2\sqrt{2e\pi}})$, we have,

$$\sum_{k=1}^{K}\sum_{h=1}^{H}\left(\mathbb{E}_{\pi^*}[l_h^k(x_h, a_h) \mid x_1 = x_1^k] - l_h^k(x_h^k, a_h^k)\right)$$

$$\leq \sum_{k=1}^{K}\sum_{h=1}^{H} -l_h^k(x_h^k, a_h^k)$$

$$\leq \sum_{k=1}^{K}\sum_{h=1}^{H}\left(5H\sqrt{d}C_{\delta'} + 5\sqrt{\frac{2d\log(1/\delta')}{3\beta_K}} + 4/3\right)\|\phi(x_h^k, a_h^k)\|_{(\Lambda_h^k)^{-1}}$$

$$= \left(5H\sqrt{d}C_{\delta'} + 5\sqrt{\frac{2d\log(1/\delta')}{3\beta_K}} + 4/3\right)\sum_{k=1}^{K}\sum_{h=1}^{H}\|\phi(x_h^k, a_h^k)\|_{(\Lambda_h^k)^{-1}}$$

$$\leq \left(5H\sqrt{d}C_{\delta'} + 5\sqrt{\frac{2d\log(1/\delta')}{3\beta_K}} + 4/3\right)\sum_{h=1}^{H}\sqrt{K}\left(\sum_{k=1}^{K}\|\phi(x_h^k, a_h^k)\|_{(\Lambda_h^k)^{-1}}^2\right)^{1/2}$$

$$\leq \left(5H\sqrt{d}C_{\delta'} + 5\sqrt{\frac{2d\log(1/\delta')}{3\beta_K}} + 4/3\right)H\sqrt{2dK\log(1+K)}$$

$$= \left(5H\sqrt{d}C_{\delta'} + 5\sqrt{\frac{2d\log(1/\delta')}{3\beta_K}} + 4/3\right)\sqrt{2dHT\log(1+K)}$$

$$= \widetilde{O}(d^{3/2}H^{3/2}\sqrt{T}).$$

Here the first, the second, and the third inequalities follow from Lemma B.7, Lemma B.6 and the Cauchy-Schwarz inequality respectively. The last inequality follows from Lemma E.4 The last equality follows from $\frac{1}{\sqrt{\beta_K}} = 10H\sqrt{d}C_{\delta'} + \frac{8}{3}$ which we defined in Lemma B.7.

By Lemma B.5, the event $\mathcal{E}(K, H, \delta')$ occurs with probability $1 - \delta'$. Thus, by union bound, the event $\mathcal{E}(K, H, \delta')$ occurs and it holds that

$$\sum_{k=1}^{K}\sum_{h=1}^{H}\left(\mathbb{E}_{\pi^*}[l_h^k(x_h, a_h) \mid x_1 = x_1^k] - l_h^k(x_h^k, a_h^k)\right) \leq \widetilde{O}(d^{3/2}H^{3/2}\sqrt{T})$$

with probability at least $(1 - (\delta' + \delta'^2 + \frac{1}{2\sqrt{2e\pi}}))$. Since $\delta \in (0, 1)$, setting $\delta' = \delta/6$, we have

$$1 - (\delta' + \delta'^2 + \frac{1}{2\sqrt{2e\pi}}) \geq 1 - \frac{\delta}{3} - \frac{1}{2\sqrt{2e\pi}}.$$

The martingale inequalities from Equation (16) occur with probability $1 - 2\delta/3$. By Equation (15) and applying union bound, we get that the final regret bound is $\widetilde{O}(d^{3/2}H^{3/2}\sqrt{T})$ with probability at least $1 - (\delta + \frac{1}{2\sqrt{2e\pi}})$. In other words, the regret bound holds with probability at least $1 - \delta$ where $\delta \in (\frac{1}{2\sqrt{2e\pi}}, 1)$.

$\square$

## C  PROOF OF SUPPORTING LEMMAS

In this section, we provide the proofs of the lemmas that we used in the regret analysis of LMC-LSVI in the previous section.

### C.1  PROOF OF PROPOSITION B.1

*Proof of Proposition B.1.* First note that for linear MDP, we have
$$\nabla L_h^k(w_h^k) = 2(\Lambda_h^k w_h^k - b_h^k).$$
The update rule is:
$$w_h^{k,j} = w_h^{k,j-1} - \eta_k \nabla L_h^k(w_h^{k,j-1}) + \sqrt{2\eta_k \beta_k^{-1}} \epsilon_h^{k,j},$$
which leads to
$$
\begin{aligned}
w_h^{k,J_k} &= w_h^{k,J_k-1} - 2\eta_k \left( \Lambda_h^k w_h^{k,J_k-1} - b_h^k \right) + \sqrt{2\eta_k \beta_k^{-1}} \epsilon_h^{k,J_k} \\
&= \left( I - 2\eta_k \Lambda_h^k \right) w_h^{k,J_k-1} + 2\eta_k b_h^k + \sqrt{2\eta_k \beta_k^{-1}} \epsilon_h^{k,J_k} \\
&= \left( I - 2\eta_k \Lambda_h^k \right)^{J_k} w_h^{k,0} + \sum_{l=0}^{J_k-1} \left( I - 2\eta_k \Lambda_h^k \right)^l \left( 2\eta_k b_h^k + \sqrt{2\eta_k \beta_k^{-1}} \epsilon_h^{k,J_k-l} \right) \\
&= \left( I - 2\eta_k \Lambda_h^k \right)^{J_k} w_h^{k,0} + 2\eta_k \sum_{l=0}^{J_k-1} \left( I - 2\eta_k \Lambda_h^k \right)^l b_h^k + \sqrt{2\eta_k \beta_k^{-1}} \sum_{l=0}^{J_k-1} \left( I - 2\eta_k \Lambda_h^k \right)^l \epsilon_h^{k,J_k-l}.
\end{aligned}
$$
Note that in Line 6 of Algorithm 1, we warm-start from previous episode and set $w_h^{k,0} = w_h^{k-1,J_{k-1}}$. Denoting $A_i = I - 2\eta_i \Lambda_h^i$, we note that $A_i$ is symmetric. Moreover, when the step size is chosen such that $0 < \eta_i < 1/(2\lambda_{\max}(\Lambda_h^i))$, $A_i$ satisfies $I \succ A_i \succ 0$. Therefore, we further have
$$
\begin{aligned}
w_h^{k,J_k} &= A_k^{J_k} w_h^{k-1,J_{k-1}} + 2\eta_k \sum_{l=0}^{J_k-1} A_k^l \Lambda_h^k \widehat{w}_h^k + \sqrt{2\eta_k \beta_k^{-1}} \sum_{l=0}^{J_k-1} A_k^l \epsilon_h^{k,J_k-l} \\
&= A_k^{J_k} w_h^{k-1,J_{k-1}} + (I - A_k)\left( A_k^0 + A_k^1 + \ldots + A_k^{J_k-1} \right) \widehat{w}_h^k + \sqrt{2\eta_k \beta_k^{-1}} \sum_{l=0}^{J_k-1} A_k^l \epsilon_h^{k,J_k-l} \\
&= A_k^{J_k} w_h^{k-1,J_{k-1}} + \left( I - A_k^{J_k} \right) \widehat{w}_h^k + \sqrt{2\eta_k \beta_k^{-1}} \sum_{l=0}^{J_k-1} A_k^l \epsilon_h^{k,J_k-l} \\
&= A_k^{J_k} \ldots A_1^{J_1} w_h^{1,0} + \sum_{i=1}^{k} A_k^{J_k} \ldots A_{i+1}^{J_{i+1}} \left( I - A_i^{J_i} \right) \widehat{w}_h^i + \sum_{i=1}^{k} \sqrt{2\eta_i \beta_i^{-1}} A_k^{J_k} \ldots A_{i+1}^{J_{i+1}} \sum_{l=0}^{J_i-1} A_i^l \epsilon_h^{i,J_i-l},
\end{aligned}
$$
where in the first equality we used $b_h^k = \Lambda_h^k \widehat{w}_h^k$, in the second equality we used the definition of $\Lambda_h^k$, and in the third equality we used the fact that $I + A + \ldots + A^{n-1} = (I - A^n)(I - A)^{-1}$. We recall a property of multivariate Gaussian distribution: if $\epsilon \sim \mathcal{N}(0, I_{d \times d})$, then we have $A\epsilon + \mu \sim \mathcal{N}(\mu, AA^T)$ for any $A \in \mathbb{R}^{d \times d}$ and $\mu \in \mathbb{R}^d$. This implies $w_h^{k,J_k}$ follows the Gaussian distribution $\mathcal{N}(\mu_h^{k,J_k}, \Sigma_h^{k,J_k})$, where

$$\mu_h^{k,J_k} = A_k^{J_k} \ldots A_1^{J_1} w_h^{1,0} + \sum_{i=1}^{k} A_k^{J_k} \ldots A_{i+1}^{J_{i+1}} \left( I - A_i^{J_i} \right) \widehat{w}_h^i. \tag{17}$$

We now derive the covariance matrix $\Sigma_h^{k,J_k}$. For a fixed $i$, denote $M_i = \sqrt{2\eta_i \beta_i^{-1}} A_k^{J_k} \dots A_{i+1}^{J_{i+1}}$. Then we have,

$$M_i \sum_{l=0}^{J_i-1} A_i^l \epsilon_h^{i,J_i-l} = \sum_{l=0}^{J_i-1} M_i A_i^l \epsilon_h^{i,J_i-l} \sim \mathcal{N}\left(0, \sum_{l=0}^{J_i-1} M_i A_i^l (M_i A_i^l)^\top\right) \sim \mathcal{N}\left(0, M_i \left(\sum_{l=0}^{J_i-1} A_i^{2l}\right) M_i^\top\right).$$

Thus we further have

$$\begin{aligned}
\Sigma_h^{k,J_k} &= \sum_{i=1}^{k} M_i \left(\sum_{l=0}^{J_i-1} A_i^{2l}\right) M_i^\top \\
&= \sum_{i=1}^{k} 2\eta_i \beta_i^{-1} A_k^{J_k} \dots A_{i+1}^{J_{i+1}} \left(\sum_{l=0}^{J_i-1} A_i^{2l}\right) A_{i+1}^{J_{i+1}} \dots A_k^{J_k} \\
&= \sum_{i=1}^{k} 2\eta_i \beta_i^{-1} A_k^{J_k} \dots A_{i+1}^{J_{i+1}} \left(I - A_i^{2J_i}\right) \left(I - A_i^2\right)^{-1} A_{i+1}^{J_{i+1}} \dots A_k^{J_k} \\
&= \sum_{i=1}^{k} \frac{1}{\beta_i} A_k^{J_k} \dots A_{i+1}^{J_{i+1}} \left(I - A_i^{2J_i}\right) \left(\Lambda_h^i\right)^{-1} (I + A_i)^{-1} A_{i+1}^{J_{i+1}} \dots A_k^{J_k}.
\end{aligned}$$

This completes the proof. $\qquad\square$

## C.2 PROOF OF LEMMA B.3

Before presenting the proof, we first need to prove the following two technical lemmas.

**Lemma C.1.** *For any $(k,h) \in [K] \times [H]$, we have*

$$\|\widehat{w}_h^k\| \le 2H\sqrt{kd/\lambda}.$$

*Proof of Lemma C.1.* We have

$$\begin{aligned}
\|\widehat{w}_h^k\| &= \left\|(\Lambda_h^k)^{-1} \sum_{\tau=1}^{k-1} \left[r_h(x_h^\tau, a_h^\tau) + V_{h+1}^k(x_{h+1}^\tau)\right] \cdot \phi(s_h^\tau, a_h^\tau)\right\| \\
&\le \frac{1}{\sqrt{\lambda}} \sqrt{k-1} \left(\sum_{\tau=1}^{k-1} \left\|\left[r_h(x_h^\tau, a_h^\tau) + V_{h+1}^k(x_{h+1}^\tau)\right] \cdot \phi(x_h^\tau, a_h^\tau)\right\|_{(\Lambda_h^k)^{-1}}^2\right)^{1/2} \\
&\le \frac{2H}{\sqrt{\lambda}} \sqrt{k-1} \left(\sum_{\tau=1}^{k-1} \|\phi(x_h^\tau, a_h^\tau)\|_{(\Lambda_h^k)^{-1}}^2\right)^{1/2} \\
&\le 2H\sqrt{kd/\lambda},
\end{aligned}$$

where the first inequality follows from Lemma E.5, the second inequality is due to $0 \le V_h^k \le H$ and the reward function being bounded by 1, and the last inequality follows from Lemma E.3. $\quad\square$

**Lemma C.2.** *Let $\lambda = 1$ in Algorithm 1. For any $(h,k) \in [H] \times [K]$ and $(x,a) \in \mathcal{S} \times \mathcal{A}$, we have*

$$\left|\phi(x,a)^\top w_h^{k,J_k} - \phi(x,a)^\top \widehat{w}_h^k\right| \le \left(5\sqrt{\frac{2d\log(1/\delta)}{3\beta_K}} + \frac{4}{3}\right) \|\phi(x,a)\|_{(\Lambda_h^k)^{-1}},$$

*with probability at least $1 - \delta^2$.*

*Proof of Lemma C.2.* By the triangle inequality, we have

$$\left|\phi(x,a)^\top w_h^{k,J_k} - \phi(x,a)^\top \widehat{w}_h^k\right| \le \left|\phi(x,a)^\top \left(w_h^{k,J_k} - \mu_h^{k,J_k}\right)\right| + \left|\phi(x,a)^\top \left(\mu_h^{k,J_k} - \widehat{w}_h^k\right)\right|. \quad (18)$$

**Bounding the term** $\left| \phi(x,a)^\top \left( w_h^{k,J_k} - \mu_h^{k,J_k} \right) \right|$: we have

$$\left| \phi(x,a)^\top \left( w_h^{k,J_k} - \mu_h^{k,J_k} \right) \right| \leq \left\| \phi(x,a)^\top \left( \Sigma_h^{k,J_k} \right)^{1/2} \right\|_2 \left\| \left( \Sigma_h^{k,J_k} \right)^{-1/2} \left( w_h^{k,J_k} - \mu_h^{k,J_k} \right) \right\|_2.$$

Since $w_h^{k,J_k} \sim \mathcal{N}(\mu_h^{k,J_k}, \Sigma_h^{k,J_k})$, we have $\left( \Sigma_h^{k,J_k} \right)^{-1/2} \left( w_h^{k,J_k} - \mu_h^{k,J_k} \right) \sim \mathcal{N}(0, I_{d\times d})$. Thus, we have

$$\mathbb{P}\left( \left\| \left( \Sigma_h^{k,J_k} \right)^{-1/2} \left( w_h^{k,J_k} - \mu_h^{k,J_k} \right) \right\|_2 \geq \sqrt{4d \log(1/\delta)} \right) \leq \delta^2. \tag{19}$$

When we choose $\eta_k \leq 1/(4\lambda_{\max}(\Lambda_h^k))$ for all $k$, we have

$$\begin{aligned}
\frac{1}{2}I &< A_k = I - 2\eta_k\Lambda_h^k < \left( 1 - 2\eta_k\lambda_{\min}(\Lambda_h^k) \right) I, \\
\frac{3}{2}I &< I + A_k = 2I - 2\eta_k\Lambda_h^k < 2I.
\end{aligned} \tag{20}$$

Also note that $A_k$ and $(\Lambda_h^k)^{-1}$ commute. Therefore, we have

$$\begin{aligned}
A_k^{2J_k} \left( \Lambda_h^k \right)^{-1} &= \left( I - 2\eta_k\Lambda_h^k \right) \ldots \left( I - 2\eta_k\Lambda_h^k \right) \left( I - 2\eta_k\Lambda_h^k \right) \left( \Lambda_h^k \right)^{-1} \\
&= \left( I - 2\eta_k\Lambda_h^k \right) \ldots \left( I - 2\eta_k\Lambda_h^k \right) \left( \Lambda_h^k \right)^{-1} \left( I - 2\eta_k\Lambda_h^k \right) \\
&= A_k^{J_k} \left( \Lambda_h^k \right)^{-1} A_k^{J_k}.
\end{aligned} \tag{21}$$

Recall the definition of $\Sigma_h^{k,J_k}$. Then

$$\begin{aligned}
&\phi(x,a)^\top \Sigma_h^{k,J_k} \phi(x,a) \\
&= \sum_{i=1}^k \frac{1}{\beta_i} \phi(x,a)^\top A_k^{J_k} \ldots A_{i+1}^{J_{i+1}} \left( I - A^{2J_i} \right) \left( \Lambda_h^i \right)^{-1} \left( I + A_i \right)^{-1} A_{i+1}^{J_{i+1}} \ldots A_k^{J_k} \phi(x,a) \\
&\leq \frac{2}{3\beta_i} \sum_{i=1}^k \phi(x,a)^\top A_k^{J_k} \ldots A_{i+1}^{J_{i+1}} \left( \left( \Lambda_h^i \right)^{-1} - A_k^{J_k} \left( \Lambda_h^i \right)^{-1} A_k^{J_k} \right) A_{i+1}^{J_{i+1}} \ldots A_k^{J_k} \phi(x,a) \\
&= \frac{2}{3\beta_K} \sum_{i=1}^k \phi(x,a)^\top A_k^{J_k} \ldots A_{i+1}^{J_{i+1}} \left( \left( \Lambda_h^i \right)^{-1} - \left( \Lambda_h^{i+1} \right)^{-1} \right) A_{i+1}^{J_{i+1}} \ldots A_k^{J_k} \phi(x,a) \\
&\quad - \frac{2}{3\beta_K} \phi(x,a)^\top A_k^{J_k} \ldots A_1^{J_1} \left( \Lambda_h^1 \right)^{-1} A_1^{J_1} \ldots A_k^{J_k} \phi(x,a) + \frac{2}{3\beta_K} \phi(x,a)^\top \left( \Lambda_h^k \right)^{-1} \phi(x,a),
\end{aligned}$$

where the first inequality is due to (20) and the last equality is due to setting $\beta_i = \beta_K$ for all $i \in [K]$. By Sherman-Morrison formula and (6), we have

$$\begin{aligned}
\left( \Lambda_h^i \right)^{-1} - \left( \Lambda_h^{i+1} \right)^{-1} &= \left( \Lambda_h^i \right)^{-1} - \left( \Lambda_h^i + \phi(x_h^i, a_h^i)\phi(x_h^i, a_h^i)^\top \right)^{-1} \\
&= \frac{\left( \Lambda_h^i \right)^{-1} \phi(x_h^i, a_h^i)\phi(x_h^i, a_h^i)^\top \left( \Lambda_h^i \right)^{-1}}{1 + \|\phi(x_h^i, a_h^i)\|^2_{(\Lambda_h^i)^{-1}}}.
\end{aligned}$$

This implies

$$\begin{aligned}
&\phi(x,a)^\top A_k^{J_k} \ldots A_{i+1}^{J_{i+1}} \left( \left( \Lambda_h^i \right)^{-1} - \left( \Lambda_h^{i+1} \right)^{-1} \right) A_{i+1}^{J_{i+1}} \ldots A_k^{J_k} \phi(x,a) \\
&= \phi(x,a)^\top A_k^{J_k} \ldots A_{i+1}^{J_{i+1}} \frac{\left( \Lambda_h^i \right)^{-1} \phi(x_h^i, a_h^i)\phi(x_h^i, a_h^i)^\top \left( \Lambda_h^i \right)^{-1}}{1 + \|\phi(x_h^i, a_h^i)\|^2_{(\Lambda_h^i)^{-1}}} A_{i+1}^{J_{i+1}} \ldots A_k^{J_k} \phi(x,a) \\
&\leq \left( \phi(x,a)^\top A_k^{J_k} \ldots A_{i+1}^{J_{i+1}} \left( \Lambda_h^i \right)^{-1} \phi(x_h^i, a_h^i) \right)^2 \\
&\leq \left\| A_k^{J_k} \ldots A_{i+1}^{J_{i+1}} \left( \Lambda_h^i \right)^{-1/2} \phi(x,a) \right\|^2_2 \cdot \left\| \left( \Lambda_h^i \right)^{-1/2} \phi(x_h^i, a_h^i) \right\|^2_2 \\
&\leq \prod_{j=i+1}^k \left( 1 - 2\eta_j\lambda_{\min}\left( \Lambda_h^j \right) \right)^{2J_j} \|\phi(x_h^i, a_h^i)\|^2_{(\Lambda_h^i)^{-1}} \|\phi(x,a)\|^2_{(\Lambda_h^i)^{-1}},
\end{aligned}$$

where the last inequality is due to (20). So, we have

$$
\phi(x,a)^\top \Sigma_h^{k,J_k} \phi(x,a) \le \frac{2}{3\beta_K} \sum_{i=1}^{k} \prod_{j=i+1}^{k} \left(1 - 2\eta_j \lambda_{\min}\left(\Lambda_h^j\right)\right)^{2J_j} \left\|\phi(x_h^i, a_h^i)\right\|_{(\Lambda_h^i)^{-1}}^2 \left\|\phi(x,a)\right\|_{(\Lambda_h^i)^{-1}}^2
$$
$$
+ \frac{2}{3\beta_K} \left\|\phi(x,a)\right\|_{(\Lambda_h^k)^{-1}}^2.
$$

Using the inequality $\sqrt{a^2 + b^2} \le a + b$ for $a, b > 0$, we thus get

$$
\|\phi(x,a)\|_{\Sigma_h^{k,J_k}} \le \sqrt{\frac{2}{3\beta_K}} \left( \|\phi(x,a)\|_{(\Lambda_h^k)^{-1}} + \sum_{i=1}^{k} \prod_{j=i+1}^{k} \left(1 - 2\eta_j \lambda_{\min}\left(\Lambda_h^j\right)\right)^{J_j} \left\|\phi(x_h^i, a_h^i)\right\|_{(\Lambda_h^i)^{-1}} \|\phi(x,a)\|_{(\Lambda_h^i)^{-1}} \right)
$$
$$(22)$$

Let's denote the R.H.S. of (22) as $\widehat{g}_h^k(\phi(x,a))$.

Therefore, it holds that

$$
\mathbb{P}\left( \left| \phi(x,a)^\top w_h^{k,J_k} - \phi(x,a)^\top \mu_h^{k,J_k} \right| \ge 2\widehat{g}_h^k(\phi(x,a))\sqrt{d\log(1/\delta)} \right)
$$
$$
\le \mathbb{P}\left( \left| \phi(x,a)^\top w_h^{k,J_k} - \phi(x,a)^\top \mu_h^{k,J_k} \right| \ge 2\sqrt{d\log(1/\delta)} \|\phi(x,a)\|_{\Sigma_h^{k,J_k}} \right)
$$
$$
\le \mathbb{P}\left( \left\| \phi(x,a)^\top \left(\Sigma_h^{k,J_k}\right)^{1/2} \right\|_2 \left\| \left(\Sigma_h^{k,J_k}\right)^{-1/2} \left( w_h^{k,J_k} - \mu_h^{k,J_k} \right) \right\|_2 \ge 2\sqrt{d\log(1/\delta)} \|\phi(x,a)\|_{\Sigma_h^{k,J_k}} \right)
$$
$$
\le \delta^2,
$$
$$(23)$$

where the last inequality follows from (19).

**Bounding the term $\phi(x,a)^\top \left( \mu_h^{k,J_k} - \widehat{w}_h^k \right)$:** Recall that,

$$
\mu_h^{k,J_k} = A_k^{J_k} \dots A_1^{J_1} w_h^{1,0} + \sum_{i=1}^{k} A_k^{J_k} \dots A_{i+1}^{J_{i+1}} \left(I - A_i^{J_i}\right) \widehat{w}_h^i
$$
$$
= A_k^{J_k} \dots A_1^{J_1} w_h^{1,0} + \sum_{i=1}^{k-1} A_k^{J_k} \dots A_{i+1}^{J_{i+1}} \left(\widehat{w}_h^i - \widehat{w}_h^{i+1}\right) - A_k^{J_k} \dots A_1^{J_1} \widehat{w}_h^1 + \widehat{w}_h^k
$$
$$
= A_k^{J_k} \dots A_1^{J_1} \left(w_h^{1,0} - \widehat{w}_h^1\right) + \sum_{i=1}^{k-1} A_k^{J_k} \dots A_{i+1}^{J_{i+1}} \left(\widehat{w}_h^i - \widehat{w}_h^{i+1}\right) + \widehat{w}_h^k.
$$

This implies that

$$
\phi(x,a)^\top \left( \mu_h^{k,J_k} - \widehat{w}_h^k \right) = \underbrace{\phi(x,a)^\top A_k^{J_k} \dots A_1^{J_1} \left( w_h^{1,0} - \widehat{w}_h^1 \right)}_{I_1} + \underbrace{\phi(x,a)^\top \sum_{i=1}^{k-1} A_k^{J_k} \dots A_{i+1}^{J_{i+1}} \left( \widehat{w}_h^i - \widehat{w}_h^{i+1} \right)}_{I_2}
$$
$$(24)$$

In Algorithm 1, we choose $w_h^{1,0} = 0$ and $\widehat{w}_h^1 = (\Lambda_h^1)^{-1} b_h^1 = 0$. Thus we have, $I_1 = 0$. Using inequalities in (20) and Lemma C.1, we have

$$
\begin{aligned}
I_2 &\leq \left| \phi(x,a)^\top \sum_{i=1}^{k-1} A_k^{J_k} \dots A_{i+1}^{J_{i+1}} \left( \widehat{w}_h^i - \widehat{w}_h^{i+1} \right) \right| \\
&= \left| \sum_{i=1}^{k-1} \phi(x,a)^\top A_k^{J_k} \dots A_{i+1}^{J_{i+1}} \left( \widehat{w}_h^i - \widehat{w}_h^{i+1} \right) \right| \\
&\leq \sum_{i=1}^{k-1} \prod_{j=i+1}^{k} \left( 1 - 2\eta_j \lambda_{\min} \left( \Lambda_h^j \right) \right)^{J_j} \| \phi(x,a) \|_2 \| \widehat{w}_h^i - \widehat{w}_h^{i+1} \|_2 \\
&\leq \sum_{i=1}^{k-1} \prod_{j=i+1}^{k} \left( 1 - 2\eta_j \lambda_{\min} \left( \Lambda_h^j \right) \right)^{J_j} \| \phi(x,a) \|_2 \left( \| \widehat{w}_h^i \|_2 + \| \widehat{w}_h^{i+1} \|_2 \right) \\
&\leq \sum_{i=1}^{k-1} \prod_{j=i+1}^{k} \left( 1 - 2\eta_j \lambda_{\min} \left( \Lambda_h^j \right) \right)^{J_j} \| \phi(x,a) \|_2 \left( 2H\sqrt{id/\lambda} + 2H\sqrt{(i+1)d/\lambda} \right) \\
&\leq 4H\sqrt{Kd/\lambda} \sum_{i=1}^{k-1} \prod_{j=i+1}^{k} \left( 1 - 2\eta_j \lambda_{\min} \left( \Lambda_h^j \right) \right)^{J_j} \| \phi(x,a) \|_2.
\end{aligned}
$$

So, it holds that

$$
\phi(x,a)^\top \left( \mu_h^{k,J_k} - \widehat{w}_h^k \right) \leq 4H\sqrt{Kd/\lambda} \sum_{i=1}^{k-1} \prod_{j=i+1}^{k} \left( 1 - 2\eta_j \lambda_{\min} \left( \Lambda_h^j \right) \right)^{J_j} \| \phi(x,a) \|_2. \tag{25}
$$

Substituting (23) and (25) into (18), we get with probability at least $1 - \delta^2$,

$$
\begin{aligned}
&\left| \phi(x,a)^\top w_h^{k,J_k} - \phi(x,a)^\top \widehat{w}_h^k \right| \\
&\leq 4H\sqrt{Kd/\lambda} \sum_{i=1}^{k-1} \prod_{j=i+1}^{k} \left( 1 - 2\eta_j \lambda_{\min} \left( \Lambda_h^j \right) \right)^{J_j} \| \phi(x,a) \|_2 + 2\sqrt{\frac{2d \log(1/\delta)}{3\beta_K}} \| \phi(x,a) \|_{(\Lambda_h^k)^{-1}} \\
&\quad + 2\sqrt{\frac{2d \log(1/\delta)}{3\beta_K}} \sum_{i=1}^{k} \prod_{j=i+1}^{k} \left( 1 - 2\eta_j \lambda_{\min} \left( \Lambda_h^j \right) \right)^{J_j} \| \phi(x_h^i, a_h^i) \|_{(\Lambda_h^i)^{-1}} \| \phi(x,a) \|_{(\Lambda_h^i)^{-1}}.
\end{aligned} \tag{26}
$$

Let's denote the R.H.S. of (26) as $Q$. Recall that, for any $j \in [K]$, we require $\eta_j \leq 1/(4\lambda_{\max}(\Lambda_h^j))$. Choosing $\eta_j = 1/(4\lambda_{\max}(\Lambda_h^j))$ yields

$$
\left( 1 - 2\eta_j \lambda_{\min}(\Lambda_h^j) \right)^{J_j} = (1 - 1/(2\kappa_j))^{J_j},
$$

where $\kappa_j = \lambda_{\max}(\Lambda_h^j)/\lambda_{\min}(\Lambda_h^j)$. In order to have $(1 - 1/(2\kappa_j))^{J_j} < \epsilon$, we need to pick $J_j$ such that

$$
J_j \geq \frac{\log(1/\epsilon)}{\log\left( \frac{1}{1 - 1/(2\kappa_j)} \right)}.
$$

Now we use the well-known fact that $e^{-x} > 1 - x$ for $0 < x < 1$. Since $1/(2\kappa_j) \leq 1/2$, we have $\log(1/(1 - 1/2\kappa_j)) \geq 1/2\kappa_j$. Thus, it suffices to set $J_j \geq 2\kappa_j \log(1/\epsilon)$ to ensure $(1 - 1/2\kappa_j)^{J_j} \leq \epsilon$. Also, note that since $\Lambda_h^i > I$, we have $1 \geq \| \phi(x,a) \|_2 \geq \| \phi(x,a) \|_{(\Lambda_h^i)^{-1}}$. Setting

$\epsilon = 1/(4HKd)$ and $\lambda = 1$, we obtain

$$
\begin{aligned}
Q &\leq \sum_{i=1}^{k-1} \epsilon^{k-i} 4H \sqrt{\frac{Kd}{\lambda}} \|\phi(x,a)\|_2 + 2\sqrt{\frac{2d \log(1/\delta)}{3\beta_K}} \left( \|\phi(x,a)\|_{(\Lambda_h^k)^{-1}} + \sum_{i=1}^{k-1} \epsilon^{k-i} \|\phi(x,a)\|_2 \right) \\
&\leq \sum_{i=1}^{k-1} \epsilon^{k-i} 4H \sqrt{\frac{Kd}{\lambda}} \sqrt{k} \|\phi(x,a)\|_{(\Lambda_h^k)^{-1}} \\
&\quad + 2\sqrt{\frac{2d \log(1/\delta)}{3\beta_K}} \left( \|\phi(x,a)\|_{(\Lambda_h^k)^{-1}} + \sum_{i=1}^{k-1} \epsilon^{k-i} \sqrt{k} \|\phi(x,a)\|_{(\Lambda_h^k)^{-1}} \right) \\
&\leq \sum_{i=1}^{k-1} \epsilon^{k-i-1} \|\phi(x,a)\|_{(\Lambda_h^k)^{-1}} + 2\sqrt{\frac{2d \log(1/\delta)}{3\beta_K}} \left( \|\phi(x,a)\|_{(\Lambda_h^k)^{-1}} + \sum_{i=1}^{k-1} \epsilon^{k-i-1} \|\phi(x,a)\|_{(\Lambda_h^k)^{-1}} \right) \\
&\leq \left( 5\sqrt{\frac{2d \log(1/\delta)}{3\beta_K}} + \frac{4}{3} \right) \|\phi(x,a)\|_{(\Lambda_h^k)^{-1}},
\end{aligned}
$$

where the second inequality is due to $\|\phi(x,a)\|_{(\Lambda_h^k)^{-1}} \geq 1/\sqrt{k} \|\phi(x,a)\|_2$ and the fourth inequality is due to $\sum_{i=1}^{k-1} \epsilon^{k-i-1} = \sum_{i=0}^{k-2} \epsilon^i < 1/(1-\epsilon) \leq 4/3$. So, we have

$$
\begin{aligned}
\mathbb{P} &\left( \left| \phi(x,a)^\top w_h^{k,J_k} - \phi(x,a)^\top \widehat{w}_h^k \right| \leq \left( 5\sqrt{\frac{2d \log(1/\delta)}{3\beta_K}} + \frac{4}{3} \right) \|\phi(x,a)\|_{(\Lambda_h^k)^{-1}} \right) \\
&\geq \mathbb{P} \left( \left| \phi(x,a)^\top w_h^{k,J_k} - \phi(x,a)^\top \widehat{w}_h^k \right| \leq Q \right) \\
&\geq 1 - \delta^2.
\end{aligned}
$$

This completes the proof. $\qquad \square$

*Proof of Lemma B.3.* From Proposition B.1, we know $w_h^{k,J_k}$ follows Gaussian distribution $\mathcal{N}(\mu_h^{k,J_k}, \Sigma_h^{k,J_k})$. Thus we can write,

$$
\left\| w_h^{k,J_k} \right\|_2 = \left\| \mu_h^{k,J_k} + \xi_h^{k,J_k} \right\|_2 \leq \left\| \mu_h^{k,J_k} \right\|_2 + \left\| \xi_h^{k,J_k} \right\|_2,
$$

where $\xi_h^{k,J_k} \sim \mathcal{N}(0, \Sigma_h^{k,J_k})$.

**Bounding $\|\mu_h^{k,J_k}\|_2$:** From Proposition B.1, we have,

$$
\begin{aligned}
\left\| \mu_h^{k,J_k} \right\|_2 &= \left\| A_k^{J_k} \dots A_1^{J_1} w_h^{1,0} + \sum_{i=1}^{k} A_k^{J_k} \dots A_{i+1}^{J_{i+1}} \left( I - A_i^{J_i} \right) \widehat{w}_h^i \right\|_2 \\
&\leq \sum_{i=1}^{k} \left\| A_k^{J_k} \dots A_{i+1}^{J_{i+1}} \left( I - A_i^{J_i} \right) \widehat{w}_h^i \right\|_2,
\end{aligned}
$$

where the inequality follows from the fact that we set $w_h^{1,0} = \mathbf{0}$ in Algorithm 1 and triangle inequality. Denoting the Frobenius of a matrix $X$ by $\|X\|_F$, we have

$$\sum_{i=1}^{k} \left\| A_k^{J_k} \dots A_{i+1}^{J_{i+1}} \left( I - A_i^{J_i} \right) \widehat{w}_h^i \right\|_2$$

$$\leq \sum_{i=1}^{k} \left\| A_k^{J_k} \dots A_{i+1}^{J_{i+1}} \left( I - A_i^{J_i} \right) \right\|_F \left\| \widehat{w}_h^i \right\|_2$$

$$\leq 2H\sqrt{\frac{Kd}{\lambda}} \sum_{i=1}^{k} \left\| A_k^{J_k} \dots A_{i+1}^{J_{i+1}} \left( I - A_i^{J_i} \right) \right\|_F$$

$$\leq 2H\sqrt{\frac{Kd}{\lambda}} \sum_{i=1}^{k} \sqrt{d} \left\| A_k^{J_k} \dots A_{i+1}^{J_{i+1}} \left( I - A_i^{J_i} \right) \right\|_2$$

$$\leq 2Hd\sqrt{\frac{K}{\lambda}} \sum_{i=1}^{k} \|A_k\|_2^{J_k} \dots \|A_{i+1}\|_2^{J_{i+1}} \left\| \left( I - A_i^{J_i} \right) \right\|_2$$

$$\leq 2Hd\sqrt{\frac{K}{\lambda}} \sum_{i=1}^{k} \prod_{j=i+1}^{k} \left( 1 - 2\eta_j \lambda_{\min}(\Lambda_h^j) \right)^{J_j} \left( \|I\|_2 + \|A_i^{J_i}\|_2 \right)$$

$$\leq 2Hd\sqrt{\frac{K}{\lambda}} \sum_{i=1}^{k} \prod_{j=i+1}^{k} \left( 1 - 2\eta_j \lambda_{\min}(\Lambda_h^j) \right)^{J_j} \left( \|I\|_2 + \|A_i\|_2^{J_i} \right)$$

$$\leq 2Hd\sqrt{\frac{K}{\lambda}} \sum_{i=1}^{k} \prod_{j=i+1}^{k} \left( 1 - 2\eta_j \lambda_{\min}(\Lambda_h^j) \right)^{J_j} \left( 1 + \left( 1 - 2\eta_i \lambda_{\min}(\Lambda_h^i) \right)^{J_i} \right)$$

$$\leq 2Hd\sqrt{\frac{K}{\lambda}} \sum_{i=1}^{k} \left( \prod_{j=i+1}^{k} \left( 1 - 2\eta_j \lambda_{\min}(\Lambda_h^j) \right)^{J_j} + \prod_{j=i}^{k} \left( 1 - 2\eta_j \lambda_{\min}(\Lambda_h^j) \right)^{J_j} \right),$$

where the second inequality is from Lemma C.1, the third inequality is due to the fact that $\operatorname{rank}(A_k^{J_k} \dots A_{i+1}^{J_{i+1}} (I - A_i^{J_i})) \leq d$, the fourth one uses the submultiplicativity of matrix norm, and the fifth one is from Lemma E.6 and (20).

As in Lemma C.2, setting $J_j \geq 2\kappa_j \log(1/\epsilon)$ where $\kappa_j = \lambda_{\max}(\Lambda_h^j)/\lambda_{\min}(\Lambda_h^j)$ and $\epsilon = 1/(4HKd)$, $\lambda = 1$, we further get

$$\sum_{i=1}^{k} \left\| A_k^{J_k} \dots A_{i+1}^{J_{i+1}} \left( I - A_i^{J_i} \right) \widehat{w}_h^i \right\|_2 \leq 2Hd\sqrt{\frac{K}{\lambda}} \sum_{i=1}^{k} \left( \epsilon^{k-i} + \epsilon^{k-i+1} \right)$$

$$\leq 4Hd\sqrt{\frac{K}{\lambda}} \sum_{i=0}^{\infty} \epsilon^i$$

$$= 4Hd\sqrt{\frac{K}{\lambda}} \left( \frac{1}{1-\epsilon} \right)$$

$$\leq 4Hd\sqrt{\frac{K}{\lambda}} \cdot \frac{4}{3}$$

$$= \frac{16}{3} Hd\sqrt{\frac{K}{\lambda}}.$$

Thus, setting $\lambda = 1$, we have

$$\|\mu_h^{k,J_k}\|_2 \leq \frac{16}{3} Hd\sqrt{K}.$$

**Bounding** $\|\xi_h^{k,J_k}\|_2$: Since $\xi_h^{k,J_k} \sim \mathcal{N}(0, \Sigma_h^{k,J_k})$, using Lemma E.1, we have

$$\mathbb{P}\left( \left\| \xi_h^{k,J_k} \right\|_2 \leq \sqrt{\frac{1}{\delta} \operatorname{Tr}\left( \Sigma_h^{k,J_k} \right)} \right) \geq 1 - \delta.$$

Recall from Proposition B.1, that

$$\Sigma_h^{k,J_k} = \sum_{i=1}^{k} \frac{1}{\beta_i} A_k^{J_k} \dots A_{i+1}^{J_{i+1}} \left( I - A_i^{2J_i} \right) \left( \Lambda_h^i \right)^{-1} \left( I + A_i \right)^{-1} A_{i+1}^{J_{i+1}} \dots A_k^{J_k}.$$

Thus,

$$\operatorname{Tr} \left( \Sigma_h^{k,J_k} \right) = \sum_{i=1}^{k} \frac{1}{\beta_i} \operatorname{Tr} \left( A_k^{J_k} \dots A_{i+1}^{J_{i+1}} \left( I - A_i^{2J_i} \right) \left( \Lambda_h^i \right)^{-1} \left( I + A_i \right)^{-1} A_{i+1}^{J_{i+1}} \dots A_k^{J_k} \right)$$

$$\leq \sum_{i=1}^{k} \frac{1}{\beta_i} \operatorname{Tr} \left( A_k^{J_k} \right) \dots \operatorname{Tr} \left( A_{i+1}^{J_{i+1}} \right) \operatorname{Tr} \left( I - A_i^{2J_i} \right) \operatorname{Tr} \left( \left( \Lambda_h^i \right)^{-1} \right) \operatorname{Tr} \left( \left( I + A_i \right)^{-1} \right)$$

$$\times \operatorname{Tr} \left( A_{i+1}^{J_{i+1}} \right) \dots \operatorname{Tr} \left( A_k^{J_k} \right),$$

where we used Lemma E.7. Note that if matrix $A$ and $B$ are positive definite matrix such that $A > B > 0$, then $\operatorname{Tr}(A) > \operatorname{Tr}(B)$. Also, recall from (20) that, when $\eta_k \leq 1/(4\lambda_{\max}(\Lambda_h^k))$ for all $k$, we have

$$\frac{1}{2} I < A_k = I - 2\eta_k \Lambda_h^k < \left( 1 - 2\eta_k \lambda_{\min}(\Lambda_h^k) \right) I,$$

$$\frac{3}{2} I < I + A_k = 2I - 2\eta_k \Lambda_h^k < 2I.$$

So, we have $A_i^{J_i} < \left( 1 - 2\eta_k \lambda_{\min}(\Lambda_h^k) \right)^{J_j} I$ and

$$\operatorname{Tr} \left( A_i^{J_i} \right) \leq \operatorname{Tr} \left( \left( 1 - 2\eta_k \lambda_{\min}(\Lambda_h^k) \right)^{J_j} I \right)$$

$$\leq d \left( 1 - 2\eta_k \lambda_{\min}(\Lambda_h^k) \right)^{J_j}$$

$$\leq d\epsilon$$

$$= \frac{d}{4HKd}$$

$$\leq 1,$$

where third inequality follows from the fact that in Lemma C.2, we chose $J_j$ such that $\left( 1 - 2\eta_j \lambda_{\min}(\Lambda_h^j) \right)^{J_j} \leq \epsilon$ and the first equality follows from the choice of $\epsilon = 1/(4HKd)$. Similarly, we have $I - A_i^{2J_i} < \left( 1 - \frac{1}{2^{2J_i}} \right) I$ and thus,

$$\operatorname{Tr} \left( I - A_i^{2J_i} \right) \leq \left( 1 - \frac{1}{2^{2J_i}} \right) d < d.$$

Likewise, using $(I + A_i)^{-1} \leq \frac{2}{3} I$, we have

$$\operatorname{Tr} \left( (I + A_i)^{-1} \right) \leq \frac{2}{3} d.$$

Finally, note that all eigenvalues of $\Lambda_h^i$ are greater than or equal to 1, which implies all eigenvalues of $(\Lambda_h^i)^{-1}$ are less than or equal to 1. Since the trace of a matrix is equal to the sum of its eigenvalues, we have

$$\operatorname{Tr} \left( (\Lambda_h^i)^{-1} \right) \leq d \cdot 1 = d.$$

Using the above observations and the choice of $\beta_i = \beta_K$ for all $i \in [K]$, we have

$$\operatorname{Tr} \left( \Sigma_h^{k,J_k} \right) \leq \sum_{i=1}^{K} \frac{1}{\beta_k} \cdot \frac{2}{3} \cdot d^3 = \frac{2}{3\beta_K} K d^3.$$

Thus we have

$$\mathbb{P} \left( \left\| \xi_h^{k,J_k} \right\|_2 \leq \sqrt{\frac{1}{\delta} \cdot \frac{2}{3\beta_K} K d^3} \right) \geq \mathbb{P} \left( \left\| \xi_h^{k,J_k} \right\|_2 \leq \sqrt{\frac{1}{\delta} \operatorname{Tr} \left( \Sigma_h^{k,J_k} \right)} \right) \geq 1 - \delta.$$

So, with probability at least $1 - \delta$, we have

$$\left\| w_h^{k, J_k} \right\|_2 \le \frac{16}{3} H d \sqrt{K} + \sqrt{\frac{2K}{3\beta_K \delta}} d^{3/2},$$

which completes the proof. $\qquad\square$

### C.3 PROOF OF LEMMA B.4

*Proof of Lemma B.4.* Applying Lemma B.3, with probability $1 - \delta/2$, we have

$$\left\| w_h^{k, J_k} \right\|_2 \le \frac{16}{3} H d \sqrt{K} + \sqrt{\frac{4K}{3\beta_K \delta}} d^{3/2} := B_{\delta/2}. \tag{27}$$

Now, considering the function class $\mathcal{V} := \{\min\{\max_{a \in \mathcal{A}} \phi(\cdot, a)^\top w, H\} : \|w\|_2 \le B_{\delta/2}\}$ and combining Lemma E.8 and Lemma E.10, we have that for any $\varepsilon > 0$ and $\delta > 0$, with probability at least $1 - \delta/2$,

$$\left\| \sum_{\tau=1}^{k-1} \phi(x_h^\tau, a_h^\tau) \left[ V_{h+1}^k(x_{h+1}^\tau) - \mathbb{P}_h V_{h+1}^k(x_h^\tau, a_h^\tau) \right] \right\|_{(\Lambda_h^k)^{-1}}$$

$$\le \left( 4H^2 \left[ \frac{d}{2} \log\left( \frac{k+\lambda}{\lambda} \right) + d \log\left( \frac{B_{\delta/2}}{\varepsilon} \right) + \log\frac{2}{\delta} \right] + \frac{8k^2 \varepsilon^2}{\lambda} \right)^{1/2}$$

$$\le 2H \left[ \frac{d}{2} \log\left( \frac{k+\lambda}{\lambda} \right) + d \log\left( \frac{B_{\delta/2}}{\varepsilon} \right) + \log\frac{2}{\delta} \right]^{1/2} + \frac{2\sqrt{2}k\varepsilon}{\sqrt{\lambda}}. \tag{28}$$

Setting $\lambda = 1$, $\varepsilon = \frac{H}{2\sqrt{2}k}$, we get

$$\left\| \sum_{\tau=1}^{k-1} \phi(x_h^\tau, a_h^\tau) \left[ V_{h+1}^k(x_{h+1}^\tau) - \mathbb{P}_h V_{h+1}^k(x_h^\tau, a_h^\tau) \right] \right\|_{(\Lambda_h^k)^{-1}}$$

$$\le 2H\sqrt{d} \left[ \frac{1}{2} \log(k+1) + \log\left( \frac{B_{\delta/2}}{\frac{H}{2\sqrt{2}k}} \right) + \log\frac{2}{\delta} \right]^{1/2} + H$$

$$\le 3H\sqrt{d} \left[ \frac{1}{2} \log(K+1) + \log\left( \frac{2\sqrt{2}KB_{\delta/2}}{H} \right) + \log\frac{2}{\delta} \right]^{1/2}. \tag{29}$$

with probability $1 - \delta/2$. Now combining (27) and (29) through a union bound, we obtain the stated result. $\qquad\square$

### C.4 PROOF OF LEMMA B.5

*Proof of Lemma B.5.* We denote the inner product over $\mathcal{S}$ by $\langle \cdot, \cdot \rangle_{\mathcal{S}}$. Using Definition 4.1, we have

$$\mathbb{P}_h V_{h+1}^k(x, a) = \phi(x, a)^\top \langle \mu_h, V_{h+1}^k \rangle_{\mathcal{S}}$$

$$= \phi(x, a)^\top \left( \Lambda_h^k \right)^{-1} \Lambda_h^k \langle \mu_h, V_{h+1}^k \rangle_{\mathcal{S}}$$

$$= \phi(x, a)^\top \left( \Lambda_h^k \right)^{-1} \left( \sum_{\tau=1}^{k-1} \phi(x_h^\tau, a_h^\tau) \phi(x_h^\tau, a_h^\tau)^\top + \lambda I \right) \langle \mu_h, V_{h+1}^k \rangle_{\mathcal{S}} \tag{30}$$

$$= \phi(x, a)^\top \left( \Lambda_h^k \right)^{-1} \left( \sum_{\tau=1}^{k-1} \phi(x_h^\tau, a_h^\tau)(\mathbb{P}_h V_{h+1}^k)(x_h^\tau, a_h^\tau) + \lambda I \langle \mu_h, V_{h+1}^k \rangle_{\mathcal{S}} \right).$$

Using (30) we obtain,

$$\phi(x,a)^\top \widehat{w}_h^k - r_h(x,a) - \mathbb{P}_h V_{h+1}^k(x,a)$$

$$= \phi(x,a)^\top \left(\Lambda_h^k\right)^{-1} \sum_{\tau=1}^{k-1} \left[ r_h(x_h^\tau, a_h^\tau) + V_{h+1}^k(x_{h+1}^\tau) \right] \cdot \phi(x_h^\tau, a_h^\tau) - r_h(x,a)$$

$$- \phi(x,a)^\top \left(\Lambda_h^k\right)^{-1} \left( \sum_{\tau=1}^{k-1} \phi(x_h^\tau, a_h^\tau)(\mathbb{P}_h V_{h+1}^k)(x_h^\tau, a_h^\tau) + \lambda I \langle \mu_h, V_{h+1}^k \rangle_\mathcal{S} \right)$$

$$= \underbrace{\phi(x,a)^\top (\Lambda_h^k)^{-1} \left( \sum_{\tau=1}^{k-1} \phi(x_h^\tau, a_h^\tau) \left[ V_{h+1}^k(x_{h+1}^\tau) - \mathbb{P}_h V_{h+1}^k(x_h^\tau, a_h^\tau) \right] \right)}_{(i)}$$

$$+ \underbrace{\phi(x,a)^\top (\Lambda_h^k)^{-1} \left( \sum_{\tau=1}^{k-1} r_h(x_h^\tau, a_h^\tau)\phi(x_h^\tau, a_h^\tau) \right) - r_h(x,a)}_{(ii)}$$

$$\underbrace{- \lambda\phi(x,a)^\top (\Lambda_h^k)^{-1} \langle \mu_h, V_{h+1}^k \rangle_\mathcal{S}}_{(iii)} . \tag{31}$$

We now provide an upper bound for each of the terms in (31).

**Term(i).** Using Cauchy-Schwarz inequality and Lemma B.4, with probability at least $1 - \delta$, we have

$$\phi(x,a)^\top(\Lambda_h^k)^{-1} \left( \sum_{\tau=1}^{k-1} \phi(x_h^\tau, a_h^\tau) \left[ V_{h+1}^k(x_{h+1}^\tau) - \mathbb{P}_h V_{h+1}^k(x_h^\tau, a_h^\tau) \right] \right)$$

$$\leq \left\| \sum_{\tau=1}^{k-1} \phi(x_h^\tau, a_h^\tau) \left[ V_{h+1}^k(x_{h+1}^\tau) - \mathbb{P}V_{h+1}^k(x_h^\tau, a_h^\tau) \right] \right\|_{(\Lambda_h^k)^{-1}} \|\phi(x,a)\|_{(\Lambda_h^k)^{-1}}$$

$$\leq 3H\sqrt{d} \left[ \frac{1}{2}\log(K+1) + \log\left( \frac{2\sqrt{2}KB_{\delta/2}}{H} \right) + \log\frac{2}{\delta} \right]^{1/2} \|\phi(x,a)\|_{(\Lambda_h^k)^{-1}} . \tag{32}$$

**Term (ii).** First note that,

$$\phi(x,a)^\top(\Lambda_h^k)^{-1} \left( \sum_{\tau=1}^{k-1} r_h(x_h^\tau, a_h^\tau)\phi(x_h^\tau, a_h^\tau) \right) - r_h(x,a)$$

$$= \phi(x,a)^\top(\Lambda_h^k)^{-1} \left( \sum_{\tau=1}^{k-1} r_h(x_h^\tau, a_h^\tau)\phi(x_h^\tau, a_h^\tau) \right) - \phi(x,a)^\top\theta_h$$

$$= \phi(x,a)^\top(\Lambda_h^k)^{-1} \left( \sum_{\tau=1}^{k-1} r_h(x_h^\tau, a_h^\tau)\phi(x_h^\tau, a_h^\tau) - \Lambda_h^k\theta_h \right)$$

$$= \phi(x,a)^\top(\Lambda_h^k)^{-1} \left( \sum_{\tau=1}^{k-1} r_h(x_h^\tau, a_h^\tau)\phi(x_h^\tau, a_h^\tau) - \sum_{\tau=1}^{k-1} \phi(x_h^\tau, a_h^\tau)\phi(x_h^\tau, a_h^\tau)^\top\theta_h - \lambda I\theta_h \right)$$

$$= \phi(x,a)^\top(\Lambda_h^k)^{-1} \left( \sum_{\tau=1}^{k-1} r_h(x_h^\tau, a_h^\tau)\phi(x_h^\tau, a_h^\tau) - \sum_{\tau=1}^{k-1} \phi(x_h^\tau, a_h^\tau)r_h(x_h^\tau, a_h^\tau) - \lambda I\theta_h \right)$$

$$= -\lambda\phi(x,a)^\top(\Lambda_h^k)^{-1}\theta_h . \tag{33}$$

Here we used the definition $r_h(x,a) = \langle \phi(x,a), \theta_h \rangle$ from Definition 4.1. Applying Cauchy-Schwarz inequality, we further get,

$$
\begin{aligned}
-\lambda \phi(x,a)^\top (\Lambda_h^k)^{-1} \theta_h &\le \lambda \|\phi(x,a)\|_{(\Lambda_h^k)^{-1}} \|\theta_h\|_{(\Lambda_h^k)^{-1}} \\
&\le \sqrt{\lambda} \|\phi(x,a)\|_{(\Lambda_h^k)^{-1}} \|\theta_h\|_2 \\
&\le \sqrt{\lambda d} \|\phi(x,a)\|_{(\Lambda_h^k)^{-1}}.
\end{aligned}
\tag{34}
$$

Here we used the observation that the largest eigenvalue of $(\Lambda_h^k)^{-1}$ is at most $1/\lambda$ and $\|\theta_h\|_2 \le \sqrt{d}$ from Definition 4.1. Combining (33) and (34), we get,

$$
\phi(x,a)^\top (\Lambda_h^k)^{-1} \left( \sum_{\tau=1}^{k-1} r_h(x_h^\tau, a_h^\tau) \phi(x_h^\tau, a_h^\tau) \right) - r_h(x,a) \le \sqrt{\lambda d} \|\phi(x,a)\|_{(\Lambda_h^k)^{-1}}.
\tag{35}
$$

**Term(iii).** Applying Cauchy-Schwarz inequality, we get,

$$
\begin{aligned}
\lambda \phi(x,a)^\top (\Lambda_h^k)^{-1} \langle \mu_h, V_{h+1}^k \rangle_{\mathcal{S}} &\le \lambda \|\phi(x,a)\|_{(\Lambda_h^k)^{-1}} \|\langle \mu_h, V_{h+1}^k \rangle_{\mathcal{S}} \|_{(\Lambda_h^k)^{-1}} \\
&\le \sqrt{\lambda} \|\phi(x,a)\|_{(\Lambda_h^k)^{-1}} \|\langle \mu_h, V_{h+1}^k \rangle_{\mathcal{S}} \|_2 \\
&\le \sqrt{\lambda} \|\phi(x,a)\|_{(\Lambda_h^k)^{-1}} \left( \sum_{\tau=1}^{d} \|\mu_h^\tau\|_1^2 \right)^{\frac{1}{2}} \|V_{h+1}^k\|_\infty \\
&\le H \sqrt{\lambda d} \|\phi(x,a)\|_{(\Lambda_h^k)^{-1}},
\end{aligned}
\tag{36}
$$

where the the last inequality follows from $\sum_{\tau=1}^{d} \|\mu_h^\tau\|_1^2 \le d$ in Definition 4.1. Combining (32), (35) and (36), and letting $\lambda = 1$, we get, with probability at least $1 - \delta$

$$
\begin{aligned}
&\left| \phi(x,a)^\top \widehat{w}_h^k - r_h(x,a) - \mathbb{P}_h V_{h+1}^k(x,a) \right| \\
&\le \left( 3H\sqrt{d} \left[ \frac{1}{2} \log(K+1) + \log\left( \frac{2\sqrt{2} K B_{\delta/2}}{H} \right) + \log \frac{2}{\delta} \right]^{1/2} + \sqrt{\lambda d} + H\sqrt{\lambda d} \right) \|\phi(x,a)\|_{(\Lambda_h^k)^{-1}} \\
&\le 5H\sqrt{d} \left[ \frac{1}{2} \log(K+1) + \log\left( \frac{2\sqrt{2} K B_{\delta/2}}{H} \right) + \log \frac{2}{\delta} \right]^{1/2} \|\phi(x,a)\|_{(\Lambda_h^k)^{-1}} \\
&= 5H\sqrt{d} C_\delta \|\phi(x,a)\|_{(\Lambda_h^k)^{-1}},
\end{aligned}
$$

where we denote $C_\delta = \left[ \frac{1}{2} \log(K+1) + \log\left( \frac{2\sqrt{2} K B_{\delta/2}}{H} \right) + \log \frac{2}{\delta} \right]^{1/2}$.  $\square$

## C.5   PROOF OF LEMMA B.6

*Proof of Lemma B.6.* First note that,

$$
\begin{aligned}
-l_h^k(x,a) &= Q_h^k(x,a) - r_h(x,a) - \mathbb{P}_h V_{h+1}^k(x,a) \\
&= \min\{ \phi(x,a)^\top w_h^{k,J_k}, H - h + 1 \} - r_h(x,a) - \mathbb{P}_h V_{h+1}^k(x,a) \\
&\le \phi(x,a)^\top w_h^{k,J_k} - r_h(x,a) - \mathbb{P}_h V_{h+1}^k(x,a) \\
&= \phi(x,a)^\top w_h^{k,J_k} - \phi(x,a)^\top \widehat{w}_h^k + \phi(x,a)^\top \widehat{w}_h^k - r_h(x,a) - \mathbb{P}_h V_{h+1}^k(x,a) \\
&\le \underbrace{\left| \phi(x,a)^\top w_h^{k,J_k} - \phi(x,a)^\top \widehat{w}_h^k \right|}_{(i)} + \underbrace{\left| \phi(x,a)^\top \widehat{w}_h^k - r_h(x,a) - \mathbb{P}_h V_{h+1}^k(x,a) \right|}_{(ii)}.
\end{aligned}
$$

Applying Lemma C.2, for any $(h,k) \in [H] \times [K]$ and $(x,a) \in \mathcal{S} \times \mathcal{A}$, we have

$$
\left| \phi(x,a)^\top w_h^{k,J_k} - \phi(x,a)^\top \widehat{w}_h^k \right| \le \left( 5\sqrt{\frac{2d \log(1/\delta)}{3\beta_K}} + \frac{4}{3} \right) \|\phi(x,a)\|_{(\Lambda_h^k)^{-1}},
$$

with probability at least $1 - \delta^2$.

From Lemma B.5, conditioned on the event $\mathcal{E}(K, H, \delta)$, for all $(h, k) \in [H] \times [K]$ and $(x, a) \in \mathcal{S} \times \mathcal{A}$, we have

$$\left| \phi(x, a)^\top \widehat{w}_h^k - r_h(x, a) - \mathbb{P}_h V_{h+1}^k(x, a) \right| \le 5H\sqrt{d}C_\delta \|\phi(x, a)\|_{(\Lambda_h^k)^{-1}}.$$

So, with probability $1 - \delta^2$,

$$-l_h^k(x, a) \le (i) + (ii)$$

$$\le \left( 5H\sqrt{d}C_\delta + 5\sqrt{\frac{2d\log(1/\delta)}{3\beta_K}} + 4/3 \right) \|\phi(x, a)\|_{(\Lambda_h^k)^{-1}}.$$

This completes the proof. $\qquad\square$

## C.6 Proof of Lemma B.7

*Proof of Lemma B.7.* We want to show $Q_h^k(x, a) \ge r_h(x, a) + \mathbb{P}_h V_{h+1}^k(x, a)$ with high probability. Recall that $Q_h^k(x, a) = \min\{\phi(x, a)^\top w_h^{k, J_k}, H - h + 1\}$. Also note that $r_h(x, a) + \mathbb{P}_h V_{h+1}^k(x, a) \le H - h + 1$. Thus, when $\phi(x, a)^\top w_h^{k, J_k} \ge H - h + 1$, we trivially have $Q_h^k(x, a) \ge r_h(x, a) + \mathbb{P}_h V_{h+1}^k(x, a)$. So, we now consider the case, when $\phi(x, a)^\top w_h^{k, J_k} \le H - h + 1$ and thus $Q_h^k(x, a) = \phi(x, a)^\top w_h^{k, J_k}$.

Based on the mean and covariance matrix defined in Proposition B.1, we have that $\phi(x, a)^\top w_h^{k, J_k}$ follows the distribution $\mathcal{N}(\phi(x, a)^\top \mu_h^{k, J_k}, \phi(x, a)^\top \Sigma_h^{k, J_k} \phi(x, a))$.

Define, $Z_k = \frac{r_h(x, a) + \mathbb{P}_h V_{h+1}^k(x, a) - \phi(x, a)^\top \mu_h^{k, J_k}}{\sqrt{\phi(x, a)^\top \Sigma_h^{k, J_k} \phi(x, a)}}$. When $|Z_k| < 1$, by Lemma E.2, we have

$$\mathbb{P}\left( \phi(x, a)^\top w_h^{k, J_k} \ge r_h(x, a) + \mathbb{P}_h V_{h+1}^k(x, a) \right)$$

$$= \mathbb{P}\left( \frac{\phi(x, a)^\top w_h^{k, J_k} - \phi(x, a)^\top \mu_h^{k, J_k}}{\sqrt{\phi(x, a)^\top \Sigma_h^{k, J_k} \phi(x, a)}} \ge \frac{r_h(x, a) + \mathbb{P}_h V_{h+1}^k(x, a) - \phi(x, a)^\top \mu_h^{k, J_k}}{\sqrt{\phi(x, a)^\top \Sigma_h^{k, J_k} \phi(x, a)}} \right)$$

$$\ge \frac{1}{2\sqrt{2\pi}} \exp\left(-Z_k^2/2\right)$$

$$\ge \frac{1}{2\sqrt{2e\pi}}.$$

We now show that $|Z_k| < 1$ under the event $\mathcal{E}(K, H, \delta)$. First note that by triangle inequality, we have

$$\left| r_h(x, a) + \mathbb{P}_h V_{h+1}^k(x, a) - \phi(x, a)^\top \mu_h^{k, J_k} \right|$$

$$\le \left| r_h(x, a) + \mathbb{P}_h V_{h+1}^k(x, a) - \phi(x, a)^\top \widehat{w}_h^k \right| + \left| \phi(x, a)^\top \widehat{w}_h^k - \phi(x, a)^\top \mu_h^{k, J_k} \right|.$$

By definition of the event $\mathcal{E}(K, H, \delta)$ from Lemma B.5, we have,

$$\left| r_h(x, a) + \mathbb{P}_h V_{h+1}^k(x, a) - \phi(x, a)^\top \widehat{w}_h^k \right| \le 5H\sqrt{d}C_\delta \|\phi(x, a)\|_{(\Lambda_h^k)^{-1}},$$

From (25), we have

$$\left| \phi(x, a)^\top \widehat{w}_h^k - \phi(x, a)^\top \mu_h^{k, J_k} \right| \le 4H\sqrt{Kd/\lambda} \sum_{i=1}^{k-1} \prod_{j=i+1}^{k} \left( 1 - 2\eta_j \lambda_{\min}\left(\Lambda_h^j\right) \right)^{J_j} \|\phi(x, a)\|_2.$$

As in proof of Lemma C.2, setting $\eta_j = 1/(4\lambda_{\max}(\Lambda_h^j))$, $J_j \geq 2\kappa_j \log(1/\epsilon)$, we have for all $j \in [K]$, $\left(1 - 2\eta_j \lambda_{\min}\left(\Lambda_h^j\right)\right)^{J_j} \leq \epsilon$. Setting $\epsilon = 1/(4HKD)$, we have,

$$
\begin{aligned}
\left|\phi(x,a)^\top \widehat{w}_h^k - \phi(x,a)^\top \mu_h^{k,J_k}\right| &\leq 4H\sqrt{Kd}\sum_{i=1}^{k-1}\epsilon^{k-i}\|\phi(x,a)\|_2 \\
&\leq \sum_{i=1}^{k-1}\epsilon^{k-i-1}\frac{1}{4HKd}4H\sqrt{Kd}\sqrt{K}\|\phi(x,a)\|_{(\Lambda_h^k)^{-1}} \\
&\leq \sum_{i=1}^{k-1}\epsilon^{k-i-1}\|\phi(x,a)\|_{(\Lambda_h^k)^{-1}} \\
&\leq \sum_{i=0}^{k-2}\epsilon^{i}\|\phi(x,a)\|_{(\Lambda_h^k)^{-1}} \\
&\leq \frac{1}{1-\epsilon}\|\phi(x,a)\|_{(\Lambda_h^k)^{-1}} \\
&\leq \frac{4}{3}\|\phi(x,a)\|_{(\Lambda_h^k)^{-1}}.
\end{aligned}
$$

So, we have

$$
\left|r_h(x,a) + \mathbb{P}_h V_{h+1}^k(x,a) - \phi(x,a)^\top \mu_h^{k,J_k}\right| \leq \left(5H\sqrt{d}C_\delta + \frac{4}{3}\right)\|\phi(x,a)\|_{(\Lambda_h^k)^{-1}}. \tag{37}
$$

Now, recall the definition of $\Sigma_h^{k,J_k}$ from Proposition B.1:

$$
\begin{aligned}
&\phi(x,a)^\top \Sigma_h^{k,J_k}\phi(x,a) \\
&= \sum_{i=1}^{k}\frac{1}{\beta_i}\phi(x,a)^\top A_k^{J_k}\ldots A_{i+1}^{J_{i+1}}\left(I - A^{2J_i}\right)\left(\Lambda_h^i\right)^{-1}\left(I + A_i\right)^{-1}A_{i+1}^{J_{i+1}}\ldots A_k^{J_k}\phi(x,a) \\
&\geq \sum_{i=1}^{k}\frac{1}{2\beta_i}\phi(x,a)^\top A_k^{J_k}\ldots A_{i+1}^{J_{i+1}}\left(I - A^{2J_i}\right)\left(\Lambda_h^i\right)^{-1}A_{i+1}^{J_{i+1}}\ldots A_k^{J_k}\phi(x,a),
\end{aligned}
$$

where we used the fact that $\frac{1}{2}I \prec (I + A_k)^{-1}$. Recall that in (21), we showed $A_k^{2J_k}\left(\Lambda_h^k\right)^{-1} = A_k^{J_k}\left(\Lambda_h^k\right)^{-1}A_k^{J_k}$. So,

$$
\begin{aligned}
&\phi(x,a)^\top \Sigma_h^{k,J_k}\phi(x,a) \\
&\geq \sum_{i=1}^{k}\frac{1}{2\beta_i}\phi(x,a)^\top A_k^{J_k}\ldots A_{i+1}^{J_{i+1}}\left(\left(\Lambda_h^i\right)^{-1} - A_k^{J_k}\left(\Lambda_h^i\right)^{-1}A_k^{J_k}\right)A_{i+1}^{J_{i+1}}\ldots A_k^{J_k}\phi(x,a) \\
&= \frac{1}{2\beta_K}\sum_{i=1}^{k-1}\phi(x,a)^\top A_k^{J_k}\ldots A_{i+1}^{J_{i+1}}\left(\left(\Lambda_h^i\right)^{-1} - (\Lambda_h^{i+1})^{-1}\right)A_{i+1}^{J_{i+1}}\ldots A_k^{J_k}\phi(x,a) \\
&\quad - \frac{1}{2\beta_K}\phi(x,a)^\top A_k^{J_k}\ldots A_1^{J_1}(\Lambda_h^1)^{-1}A_1^{J_1}\ldots A_k^{J_k}\phi(x,a) + \frac{1}{2\beta_K}\phi(x,a)^\top (\Lambda_h^k)^{-1}\phi(x,a),
\end{aligned}
$$

where we used the choice of $\beta_i = \beta_K$ for all $i \in [K]$. By Sherman-Morrison formula and (6), we have

$$
\begin{aligned}
\left(\Lambda_h^i\right)^{-1} - \left(\Lambda_h^{i+1}\right)^{-1} &= \left(\Lambda_h^i\right)^{-1} - \left(\Lambda_h^i + \phi(x_h^i,a_h^i)\phi(x_h^i,a_h^i)^\top\right)^{-1} \\
&= \frac{\left(\Lambda_h^i\right)^{-1}\phi(x_h^i,a_h^i)\phi(x_h^i,a_h^i)^\top\left(\Lambda_h^i\right)^{-1}}{1 + \|\phi(x_h^i,a_h^i)\|_{(\Lambda_h^i)^{-1}}^2},
\end{aligned}
$$

which implies

$$\left| \phi(x,a)^\top A_k^{J_k} \dots A_{i+1}^{J_{i+1}} \left( (\Lambda_h^i)^{-1} - (\Lambda_h^{i+1})^{-1} \right) A_{i+1}^{J_{i+1}} \dots A_k^{J_k} \phi(x,a) \right|$$

$$= \left| \phi(x,a)^\top A_k^{J_k} \dots A_{i+1}^{J_{i+1}} \frac{(\Lambda_h^i)^{-1} \phi(x_h^i, a_h^i) \phi(x_h^i, a_h^i)^\top (\Lambda_h^i)^{-1}}{1 + \|\phi(x_h^i, a_h^i)\|_{(\Lambda_h^i)^{-1}}^2} A_{i+1}^{J_{i+1}} \dots A_k^{J_k} \phi(x,a) \right|$$

$$\leq \left( \phi(x,a)^\top A_k^{J_k} \dots A_{i+1}^{J_{i+1}} (\Lambda_h^i)^{-1} \phi(x_h^i, a_h^i) \right)^2$$

$$\leq \left\| A_k^{J_k} \dots A_{i+1}^{J_{i+1}} (\Lambda_h^i)^{-1/2} \phi(x,a) \right\|_2^2 \left\| (\Lambda_h^i)^{-1/2} \phi(x_h^i, a_h^i) \right\|_2^2$$

$$\leq \prod_{j=i+1}^k \left( 1 - 2\eta_j \lambda_{\min}(\Lambda_h^j) \right)^{2J_j} \|\phi(x_h^i, a_h^i)\|_{(\Lambda_h^i)^{-1}}^2 \|\phi(x,a)\|_{(\Lambda_h^i)^{-1}}^2,$$

where we used $0 < 1/i \leq \|\phi(x,a)\|_{(\Lambda_h^i)^{-1}} \leq 1$. Therefore, we have

$$\phi(x,a)^\top \Sigma_h^{k, J_k} \phi(x,a)$$

$$\geq \frac{1}{2\beta_K} \|\phi(x,a)\|_{(\Lambda_h^k)^{-1}}^2 - \frac{1}{2\beta_K} \prod_{i=1}^k \left( 1 - 2\eta_i \lambda_{\min}(\Lambda_h^i) \right)^{2J_i} \|\phi(x,a)\|_{(\Lambda_h^1)^{-1}}^2$$

$$- \frac{1}{2\beta_K} \sum_{i=1}^{k-1} \prod_{j=i+1}^k \left( 1 - 2\eta_j \lambda_{\min}(\Lambda_h^j) \right)^{2J_j} \|\phi(x_h^i, a_h^i)\|_{(\Lambda_h^i)^{-1}}^2 \|\phi(x,a)\|_{(\Lambda_h^i)^{-1}}^2.$$

Similar to the proof of Lemma C.2, when we choose $J_j \geq \kappa_j \log(3\sqrt{k})$, we have

$$\|\phi(x,a)\|_{\Sigma_h^{k, J_k}} \geq \frac{1}{2\sqrt{\beta_K}} \left( \|\phi(x,a)\|_{(\Lambda_h^k)^{-1}} - \frac{\|\phi(x,a)\|_2}{(3\sqrt{k})^k} - \sum_{i=1}^{k-1} \frac{1}{(\sqrt{3k})^{k-i}} \|\phi(x,a)\|_2 \right)$$

$$\geq \frac{1}{2\sqrt{\beta_K}} \left( \|\phi(x,a)\|_{(\Lambda_h^k)^{-1}} - \frac{1}{3\sqrt{k}} \|\phi(x,a)\|_2 - \frac{1}{6\sqrt{k}} \|\phi(x,a)\|_2 \right) \quad (38)$$

$$\geq \frac{1}{2\sqrt{\beta_K}} \|\phi(x,a)\|_{(\Lambda_h^k)^{-1}},$$

where we used the fact that $\lambda_{\min}((\Lambda_h^k)^{-1}) \geq 1/k$. Therefore, according to (37) and (38), it holds that

$$|Z_k| = \left| \frac{r_h(x,a) + \mathbb{P}_h V_{h+1}^k(x,a) - \phi(x,a)^\top \mu_h^{k, J_k}}{\sqrt{\phi(x,a)^\top \Sigma_h^{k, J_k} \phi(x,a)}} \right|$$

$$\leq \frac{5H\sqrt{d}C_\delta + \frac{4}{3}}{\frac{1}{2\sqrt{\beta_K}}}, \quad (39)$$

which implies $|Z_k| < 1$ when $\frac{1}{\sqrt{\beta_K}} = 10H\sqrt{d}C_\delta + \frac{8}{3}$. $\qquad \square$

## D REMOVING THE INTERVAL CONSTRAINT ON $\delta$ IN THEOREM 4.2

As discussed in Remark 4.3, one shortcoming of Theorem 4.2 is that it requires the failure probability $\delta$ to be in the interval $(\frac{1}{2\sqrt{2e\pi}}, 1)$. However, in frequentist regret analysis it is desirable that the regret bound holds for arbitrarily small failure probability. Motivated from optimistic reward sampling scheme proposed in (Ishfaq et al., 2021), below we propose a modification of LMC-LSVI for which we have the same regret bound that holds with probability $1 - \delta$ for any $\delta \in (0, 1)$. We call this version of the algorithm MS-LMC-LSVI where MS stands for multi-sampling.

### D.1 MULTI-SAMPLING LMC-LSVI

Multi-Sampling LMC-LSVI follows the same algorithm as in LMC-LSVI (Algorithm 1) with one difference. In Algorithm 1, we maintain one parameterization $w_h$ for each step $h \in [H]$. After

performing noisy gradient descent on this parameter $J_k$ times in Line 7 to 10, we use this perturbed parameter to define our estimated Q function in Line 11. We follow exactly the same procedure in MS-LMC-LSVI but instead of maintaining one estimate of Q function, we generate $M$ estimates for Q function $\{Q_h^{k,m}\}_{m \in [M]}$ through maintaining $M$ samples of $w$: $\{w_h^{k,J_k,m}\}_{m \in [M]}$ where $Q_h^{k,m}(\cdot, \cdot) = \phi(\cdot, \cdot)^\top w_h^{k,J_k,m}$ and $\{w_h^{1,0,m}\}_{m \in [M]}$ are intialized as zero vector. Then, we can make an optimistic estimate of Q function by setting the following

$$Q_h^k(\cdot, \cdot) = \min \Big\{ \max_{m \in [M]} \{Q_h^{k,m}(\cdot, \cdot)\}, H - h + 1 \Big\}. \tag{40}$$

## D.2 SUPPORTING LEMMAS

First, we outline the supporting lemmas required for the regret analysis of MS-LMC-LSVI.

**Lemma D.1.** *Let $\lambda = 1$ in (4). For any $(k, h) \in [K] \times [H]$ and $m \in [M]$, we have*

$$\left\| w_h^{k,J_k,m} \right\|_2 \le \frac{16}{3} H d \sqrt{K} + \sqrt{\frac{2K}{3\beta_K \delta}} d^{3/2} := B_\delta,$$

*with probability at least $1 - \delta$.*

*Proof of Lemma D.1.* The proof is identical to that of Lemma B.3. $\square$

**Lemma D.2.** *Let $\lambda = 1$ in (4). For any fixed $0 < \delta < 1$, with probability $1 - \delta$, we have for all $(k, h) \in [K] \times [H]$,*

$$\left\| \sum_{\tau=1}^{k-1} \phi(x_h^\tau, a_h^\tau) \left[ V_{h+1}^k(x_{h+1}^\tau) - \mathbb{P}_h V_{h+1}^k(x_h^\tau, a_h^\tau) \right] \right\|_{(\Lambda_h^k)^{-1}}$$

$$\le 3H\sqrt{d} \left[ \frac{1}{2} \log(K+1) + M \log \left( \frac{2\sqrt{2}KB_{\delta/(2M)}}{H} \right) + \log \frac{2}{\delta} \right]^{1/2},$$

*where $B_\delta = \frac{16}{3} H d \sqrt{K} + \sqrt{\frac{2K}{3\beta_K \delta}} d^{3/2}$.*

*Proof of Lemma D.2.* Applying Lemma D.1 and applying union bound over all $m$, with probability $1 - \delta/2$, for all $m \in [M]$, we have

$$\left\| w_h^{k,J_k,m} \right\|_2 \le \frac{16}{3} H d \sqrt{K} + \sqrt{\frac{4MK}{3\beta_K \delta}} d^{3/2} := B_{\delta/(2M)} \tag{41}$$

Similar to the proof of Lemma B.4, considering the function class $\mathcal{V} := \{\max_{a \in \mathcal{A}} \min \{\max_{m \in [M]} \phi(\cdot, a)^\top w^m, H\} : \forall m \in [M], \|w^m\|_2 \le B_{\delta/(2M)}\}$ and combining Lemma E.8 and Lemma E.11, we have that for any $\varepsilon > 0$ and $\delta > 0$, with probability at least $1 - \delta/2$,

$$\left\| \sum_{\tau=1}^{k-1} \phi(x_h^\tau, a_h^\tau) \left[ V_{h+1}^k(x_{h+1}^\tau) - \mathbb{P}_h V_{h+1}^k(x_h^\tau, a_h^\tau) \right] \right\|_{(\Lambda_h^k)^{-1}}$$

$$\le 2H \left[ \frac{d}{2} \log \left( \frac{k+\lambda}{\lambda} \right) + dM \log \left( \frac{B_{\delta/(2M)}}{\varepsilon} \right) + \log \frac{2}{\delta} \right]^{1/2} + \frac{2\sqrt{2}k\varepsilon}{\sqrt{\lambda}}. \tag{42}$$

As in the proof of Lemma B.4, setting $\lambda = 1$, $\varepsilon = \frac{H}{2\sqrt{2}k}$, we get

$$\left\| \sum_{\tau=1}^{k-1} \phi(x_h^\tau, a_h^\tau) \left[ V_{h+1}^k(x_{h+1}^\tau) - \mathbb{P}_h V_{h+1}^k(x_h^\tau, a_h^\tau) \right] \right\|_{(\Lambda_h^k)^{-1}}$$

$$\le 3H\sqrt{d} \left[ \frac{1}{2} \log(K+1) + M \log \left( \frac{2\sqrt{2}KB_{\delta/(2M)}}{H} \right) + \log \frac{2}{\delta} \right]^{1/2}. \tag{43}$$

with probability $1 - \delta/2$. Applying union bound completes the proof. $\square$

**Lemma D.3.** *Let* $\lambda = 1$ *in* (4). *Define the following event*

$$
\begin{aligned}
&\mathcal{E}(K, H, M, \delta) \\
&= \Big\{ \big| \phi(x,a)^\top \widehat{w}_h^k - r_h(x,a) - \mathbb{P}_h V_{h+1}^k(x,a) \big| \le 5H\sqrt{d}C_{\delta,M} \|\phi(x,a)\|_{(\Lambda_h^k)^{-1}}, \\
&\qquad \forall (h,k) \in [H] \times [K] \text{ and } \forall (x,a) \in \mathcal{S} \times \mathcal{A} \Big\},
\end{aligned}
\tag{44}
$$

*where we denote* $C_{\delta,M} = \left[ \frac{1}{2} \log(K+1) + M \log \left( \frac{2\sqrt{2}KB_{\delta/(2M)}}{H} \right) + \log \frac{2}{\delta} \right]^{1/2}$ *and* $B_\delta$ *is defined in Lemma D.2. Then we have* $\mathbb{P}(\mathcal{E}(K, H, M, \delta)) \ge 1 - \delta$.

*Proof of Lemma D.3.* The proof is exactly the same as that of Lemma B.5. We just need to replace Lemma B.4 with Lemma D.2 when it is used. □

**Lemma D.4** (Error bound). *Let* $\lambda = 1$ *in* (4). *For any* $\delta \in (0,1)$ *conditioned on the event* $\mathcal{E}(K, H, M, \delta)$, *for all* $(h,k) \in [H] \times [K]$ *and* $(x,a) \in \mathcal{S} \times \mathcal{A}$, *with probability at least* $1 - \delta^2$, *we have*

$$
-l_h^k(x,a) \le \left( 5H\sqrt{d}C_\delta + 5\sqrt{\frac{2d \log(1/\delta)}{3\beta_K}} + 4/3 \right) \|\phi(x,a)\|_{(\Lambda_h^k)^{-1}},
\tag{45}
$$

*where* $C_{\delta,M}$ *is defined in Lemma D.3.*

*Proof of Lemma D.4.* The proof is exactly the same as that of Lemma B.6. We just need to replace Lemma B.5 with Lemma D.3 when it is used. □

**Lemma D.5** (Optimism). *Let* $\lambda = 1$ *in* (4) *and* $M = \log(HK/\delta)/\log(1/(1-c))$ *where* $c = \frac{1}{2\sqrt{2e\pi}}$ *and* $\delta \in (0,1)$. *Conditioned on the event* $\mathcal{E}(K, H, M, \delta)$, *for all* $(h,k) \in [H] \times [K]$ *and* $(x,a) \in \mathcal{S} \times \mathcal{A}$, *with probability at least* $1 - \delta$, *we have*

$$
l_h^k(x,a) \le 0.
\tag{46}
$$

*Proof of Lemma D.5.* We have $Q_h^k(x,a) = H - h + 1$, when $\max_{m \in [M]} Q_h^{k,m}(x,a) \ge H - h + 1$, and we trivially have $l_h^k(x,a) \le 0$.

Now, we consider the case, when $Q_h^k(x,a) = \max_{m \in [M]} Q_h^{k,m}$. Using the exact same proof steps as done in Lemma B.7, we can first show that for any $(k,h) \in [K] \times [H]$ and any $m \in [M]$

$$
\mathbb{P}\left( Q_h^{k,m}(x,a) \ge r_h(x,a) + \mathbb{P}_h V_{h+1}^k(x,a) \right) \ge \frac{1}{2\sqrt{2e\pi}},
\tag{47}
$$

where $Q_h^{k,m} = \phi(x,a)^\top w_h^{k,J_k,m}$.

Note that

$$
\begin{aligned}
&\mathbb{P}\left( l_h^k(x,a) \le 0, \ \forall (x,a) \in \mathcal{S} \times \mathcal{A} \right) \\
&= \mathbb{P}\left( Q_h^k(x,a) \ge r_h(x,a) + \mathbb{P}_h V_{h+1}^k(x,a), \ \forall (x,a) \in \mathcal{S} \times \mathcal{A} \right) \\
&= 1 - \mathbb{P}\left( \exists (x,a) \in \mathcal{S} \times \mathcal{A} : Q_h^k(x,a) \le r_h(x,a) + \mathbb{P}_h V_{h+1}^k(x,a) \right)
\end{aligned}
\tag{48}
$$

Now,

$$\mathbb{P}\left(\exists (x,a) \in \mathcal{S} \times \mathcal{A} : Q_h^k(x,a) \le r_h(x,a) + \mathbb{P}_h V_{h+1}^k(x,a)\right)$$

$$= \mathbb{P}\left(\exists (x,a) \in \mathcal{S} \times \mathcal{A} : \max_{m \in [M]} Q_h^{k,m}(x,a) \le r_h(x,a) + \mathbb{P}_h V_{h+1}^k(x,a)\right)$$

$$= \mathbb{P}\left(\exists (x,a) \in \mathcal{S} \times \mathcal{A} : \forall m \in [M], \ Q_h^{k,m}(x,a) \le r_h(x,a) + \mathbb{P}_h V_{h+1}^k(x,a)\right)$$

$$\le \mathbb{P}\left(\forall m \in [M], \ \exists (x_m, a_m) \in \mathcal{S} \times \mathcal{A} : Q_h^{k,m}(x_m, a_m) \le r_h(x_m, a_m) + \mathbb{P}_h V_{h+1}^k(x_m, a_m)\right)$$

$$= \prod_{m=1}^{M} \mathbb{P}\left(\exists (x,a) \in \mathcal{S} \times \mathcal{A} : Q_h^{k,m}(x,a) \le r_h(x,a) + \mathbb{P}_h V_{h+1}^k(x,a)\right)$$

$$= \prod_{m=1}^{M}\left(1 - \mathbb{P}\left(Q_h^{k,m}(x,a) \ge r_h(x,a) + \mathbb{P}_h V_{h+1}^k(x,a), \ \forall (x,a) \in \mathcal{S} \times \mathcal{A}\right)\right)$$

$$\le \left(1 - \frac{1}{2\sqrt{2e\pi}}\right)^M$$

$$:= \frac{\delta}{HK},$$

where the last inequality follows from (47).

Thus, from (48), we have

$$\mathbb{P}\left(l_h^k(x,a) \le 0, \ \forall (x,a) \in \mathcal{S} \times \mathcal{A}\right) \ge 1 - \frac{\delta}{HK}.$$

Denoting $c = \frac{1}{2\sqrt{2e\pi}}$ and solving for $M$, we have $M = \log(HK/\delta)/\log(1/(1-c))$.

Applying union bound over $h$ and $k$, $\forall (x,a) \in \mathcal{S} \times \mathcal{A}$ and $\forall (h,k) \in [H] \times [K]$, with probability $1 - \delta$, we have, $l_h^k(x,a) \le 0$. This completes the proof. $\square$

### D.3 REGRET ANALYSIS OF MS-LMC-LSVI

**Theorem D.6.** *Let $\lambda = 1$ in (4), $\frac{1}{\beta_k} = \widetilde{O}(H\sqrt{d})$. For any $k \in [K]$, let the learning rate $\eta_k = 1/(4\lambda_{\max}(\Lambda_h^k))$, the update number $J_k = 2\kappa_k \log(4HKd)$ where $\kappa_k = \lambda_{\max}(\Lambda_h^k)/\lambda_{\min}(\Lambda_h^k)$ is the condition number of $\Lambda_h^k$. For any $\delta \in (0,1)$, let $M = \log(HK/\delta)/\log(1/(1-c))$ where $c = \frac{1}{2\sqrt{2e\pi}}$. Under Definition 4.1, the regret of MS-LMC-LSVI satisfies*

$$Regret(K) = \widetilde{O}(d^{3/2}H^{3/2}\sqrt{T}),$$

*with probability at least $1 - \delta$.*

*Proof of Theorem D.6.* The proof is essentially same as that of Theorem B.8. The only difference is in computing the probabilites while applying union bound. Nevertheless, we highlight the key steps.

We use the same regret decomposition from (12). We bound the term (i), (ii) and (iii) in (12) exactly in the same way as in the proof of Theorem B.8 and thus with probability $1 - 2\delta/3$, we have

$$\sum_{k=1}^{K}\sum_{h=1}^{H} \mathcal{D}_h^k + \sum_{k=1}^{K}\sum_{h=1}^{H} \mathcal{M}_h^k \le 2\sqrt{2H^2 T \log(3/\delta)}, \tag{49}$$

Now we bound term (iv) in (12). Suppose the event $\mathcal{E}(K, H, M, \delta')$ defined in Lemma D.3 holds. Using Lemma D.4 and Lemma D.5 with union bound, similar to the proof of Theorem B.8, we can show that with probability $1 - (d'^2 + d')$, we have

$$\sum_{k=1}^{K}\sum_{h=1}^{H}\left(\mathbb{E}_{\pi^*}[l_h^k(x_h, a_h) \mid x_1 = x_1^k] - l_h^k(x_h^k, a_h^k)\right) \le \widetilde{O}(d^{3/2}H^{3/2}\sqrt{T}). \tag{50}$$

By Lemma D.3, the event $\mathcal{E}(K, H, M, \delta')$ occurs with probability $1 - \delta'$. By union bound the event $\mathcal{E}(K, H, M, \delta')$ occurs and (50) holds with probability at least $1 - (\delta'^2 + \delta' + \delta')$. As $\delta \in (0, 1)$, setting $\delta' = \delta/9$, we have $1 - (\delta'^2 + \delta' + \delta') \geq 1 - 3\delta' = 1 - \delta/3$.

Applying another union bound over the last inequality and the martingale inequalities from (49), we have, the regret is bounded by $\widetilde{O}(d^{3/2} H^{3/2} \sqrt{T})$ with probability at least $1 - \delta$ and this completes the proof. $\qquad\square$

# E  AUXILIARY LEMMAS

## E.1  GAUSSIAN CONCENTRATION

In this section, we present some auxiliary technical lemmas that are of general interest instead of closely related to our problem setting.

**Lemma E.1.** *Given a multivariate normal distribution $X \sim \mathcal{N}(0, \Sigma_{d \times d})$, we have,*

$$\mathbb{P}\left( \|X\|_2 \leq \sqrt{\frac{1}{\delta} \operatorname{Tr}(\Sigma)} \right) \geq 1 - \delta.$$

*Proof of Lemma E.1.* From the properties of multivariate Gaussian distribution, $X = \Sigma^{1/2} \xi$ for $\xi \sim \mathcal{N}(0, I_{d \times d})$. As $\Sigma^{1/2}$ is symmetric, it can be decomposed as $\Sigma^{1/2} = Q \Lambda Q^\top$, where $Q$ is orthogonal and $\Lambda$ is diagonal. Hence,

$$\mathbb{P}\left( \|X\|_2 \leq C^2 \right) = \mathbb{P}\left( \|X\|_2^2 \leq C^2 \right) = \mathbb{P}\left( \|Q\Lambda Q^T \xi\|_2^2 \leq C^2 \right) = \mathbb{P}\left( \|\Lambda Q^T \xi\|_2^2 \leq C^2 \right),$$

since orthogonal transformation preserves the norm. Another property of standard Gaussian distribution is that it is spherically symmetric. That is, $Q\xi \overset{d}{=} \xi$ for any orthogonal matrix $Q$. So,

$$\mathbb{P}\left( \|\Lambda Q^T \xi\|_2^2 \leq C^2 \right) = \mathbb{P}\left( \|\Lambda \xi\|_2^2 \leq C^2 \right),$$

as $Q^\top$ is also orthogonal. Observe that $\|\Lambda \xi\|_2^2 = \sum_{i=1}^d \lambda_i^2 \xi_i^2$ is the sum of the independent $\chi_1^2$-distributed variables with $\mathrm{E}\left( \|\Lambda \xi\|_2^2 \right) = \sum_{i=1}^d \lambda_i^2 = \operatorname{Tr}(\Lambda^2) = \sum_{i=1}^d \operatorname{Var}(X_i)$. From Markov's inequality,

$$\mathbb{P}\left( \|\Lambda \xi\|_2^2 \leq C^2 \right) \geq 1 - \frac{1}{C^2} \cdot \mathrm{E}\left( \|\Lambda \xi\|_2^2 \right).$$

So,

$$\delta = \frac{1}{C^2} \cdot \mathrm{E}\left( \|\Lambda \xi\|_2^2 \right) \Leftrightarrow C = \sqrt{\frac{1}{\delta} \sum_{i=1}^d \operatorname{Var}(X_i)} = \sqrt{\frac{1}{\delta} \operatorname{Tr}(\Sigma)},$$

which completes the proof. $\qquad\square$

**Lemma E.2** (Abramowitz and Stegun (1964)). *Suppose $Z$ is a Gaussian random variable $Z \sim \mathcal{N}(\mu, \sigma^2)$, where $\sigma > 0$. For $0 \leq z \leq 1$, we have*

$$\mathbb{P}(Z > \mu + z\sigma) \geq \frac{1}{\sqrt{8\pi}} e^{\frac{-z^2}{2}}, \qquad \mathbb{P}(Z < \mu - z\sigma) \geq \frac{1}{\sqrt{8\pi}} e^{\frac{-z^2}{2}}.$$

*And for $z \geq 1$, we have*

$$\frac{e^{-z^2/2}}{2z\sqrt{\pi}} \leq \mathbb{P}(|Z - \mu| > z\sigma) \leq \frac{e^{-z^2/2}}{z\sqrt{\pi}}.$$

## E.2  INEQUALITIES FOR SUMMATIONS

**Lemma E.3** (Lemma D.1 in Jin et al. (2020)). *Let $\Lambda_h = \lambda I + \sum_{i=1}^t \phi_i \phi_i^\top$, where $\phi_i \in \mathbb{R}^d$ and $\lambda > 0$. Then it holds that*

$$\sum_{i=1}^t \phi_i^\top (\Lambda_h)^{-1} \phi_i \leq d.$$

**Lemma E.4** (Lemma 11 in Abbasi-Yadkori et al. (2011)). *Using the same notation as defined in this paper*

$$\sum_{k=1}^{K} \left\| \phi(s_h^k, a_h^k) \right\|_{(\Lambda_h^k)^{-1}}^2 \leq 2d \log\left(\frac{\lambda + K}{\lambda}\right).$$

**Lemma E.5** (Lemma D.5 in Ishfaq et al. (2021)). *Let $A \in \mathbb{R}^{d \times d}$ be a positive definite matrix where its largest eigenvalue $\lambda_{\max}(A) \leq \lambda$. Let $x_1, \ldots, x_k$ be $k$ vectors in $\mathbb{R}^d$. Then it holds that*

$$\left\| A \sum_{i=1}^{k} x_i \right\| \leq \sqrt{\lambda k} \left( \sum_{i=1}^{k} \|x_i\|_A^2 \right)^{1/2}.$$

### E.3 Linear Algebra Lemmas

**Lemma E.6.** *Consider two symmetric positive semidefinite square matrices $A$ and $B$. If $A \geq B$, then $\|A\|_2 \geq \|B\|_2$.*

*Proof of Lemma E.6.* Note that $A - B$ is also positive semidefinite. Now,

$$\|B\|_2 = \sup_{\|x\|=1} x^\top B x \leq \sup_{\|x\|=1} \left( x^\top B x + x^\top (A - B) x \right) = \sup_{\|x\|=1} x^\top A x = \|A\|_2. \tag{51}$$

This completes the proof. $\qquad\qquad\square$

**Lemma E.7** ((Horn and Johnson, 2012)). *If $A$ and $B$ are positive semi-definite square matrices of the same size, then*

$$0 \leq [\text{Tr}(AB)]^2 \leq \text{Tr}(A^2) \text{Tr}(B^2) \leq [\text{Tr}(A)]^2 [\text{Tr}(B)]^2.$$

### E.4 Covering numbers and self-normalized processes

**Lemma E.8** (Lemma D.4 in Jin et al. (2020)). *Let $\{s_i\}_{i=1}^{\infty}$ be a stochastic process on state space $\mathcal{S}$ with corresponding filtration $\{\mathcal{F}_i\}_{i=1}^{\infty}$. Let $\{\phi_i\}_{i=1}^{\infty}$ be an $\mathbb{R}^d$-valued stochastic process where $\phi_i \in \mathcal{F}_{i-1}$, and $\|\phi_i\| \leq 1$. Let $\Lambda_k = \lambda I + \sum_{i=1}^{k} \phi_i \phi_i^\top$. Then for any $\delta > 0$, with probability at least $1 - \delta$, for all $k \geq 0$, and any $V \in \mathcal{V}$ with $\sup_{s \in \mathcal{S}} |V(s)| \leq H$, we have*

$$\left\| \sum_{i=1}^{k} \phi_i \{ V(s_i) - \mathbb{E}[V(s_i) \mid \mathcal{F}_{i-1}] \} \right\|_{\Lambda_k^{-1}}^2 \leq 4H^2 \left[ \frac{d}{2} \log\left(\frac{k+\lambda}{\lambda}\right) + \log \frac{\mathcal{N}_\varepsilon}{\delta} \right] + \frac{8k^2 \epsilon^2}{\lambda},$$

*where $\mathcal{N}_\varepsilon$ is the $\varepsilon$-covering number of $\mathcal{V}$ with respect to the distance $dist(V, V') = \sup_{s \in \mathcal{S}} |V(s) - V'(s)|$.*

**Lemma E.9** (Covering number of Euclidean ball, Vershynin (2018) ). *For any $\varepsilon > 0$, the $\varepsilon$-covering number, $\mathcal{N}_\varepsilon$, of the Euclidean ball of radius $B > 0$ in $\mathbb{R}^d$ satisfies*

$$\mathcal{N}_\varepsilon \leq \left(1 + \frac{2B}{\varepsilon}\right)^d \leq \left(\frac{3B}{\varepsilon}\right)^d.$$

**Lemma E.10.** *Let $\mathcal{V}$ denote a class of functions mapping from $\mathcal{S}$ to $\mathbb{R}$ with the following parametric form*

$$V(\cdot) = \min\left\{ \max_{a \in \mathcal{A}} \phi(\cdot, a)^\top w, H \right\},$$

*where the parameter $w$ satisifies $\|w\| \leq B$ and for all $(x, a) \in \mathcal{S} \times \mathcal{A}$, we have $\|\phi(x, a)\| \leq 1$. Let $N_{\mathcal{V}, \varepsilon}$ be the $\varepsilon$-covering number of $\mathcal{V}$ with respect to the distance $dist(V, V') = \sup_x |V(x) - V'(x)|$. Then*

$$\log N_{\mathcal{V}, \varepsilon} \leq d \log\left(1 + 2B/\varepsilon\right) \leq d \log\left(3B/\varepsilon\right).$$

*Proof of Lemma E.10.* Consider any two functions $V_1, V_2 \in \mathcal{V}$ with parameters $w_1$ and $w_2$ respectively. Since both $\min\{\cdot, H\}$ and $\max_a$ are contraction maps, we have

$$
\begin{aligned}
\mathrm{dist}(V_1, V_2) &\leq \sup_{x,a} \left| \phi(x,a)^\top w_1 - \phi(x,a)^\top w_2 \right| \\
&\leq \sup_{\phi:\|\phi\|\leq 1} \left| \phi^\top w_1 - \phi^\top w_2 \right| \\
&= \sup_{\phi:\|\phi\|\leq 1} \left| \phi^\top (w_1 - w_2) \right| \\
&\leq \sup_{\phi:\|\phi\|\leq 1} \|\phi\|_2 \|w_1 - w_2\|_2 \\
&\leq \|w_1 - w_2\|,
\end{aligned}
\tag{52}
$$

Let $N_{w,\varepsilon}$ denote the $\varepsilon$-covering number of $\{w \in \mathbb{R}^d \mid \|w\| \leq B\}$. Then, Lemma E.9 implies

$$
N_{w,\varepsilon} \leq \left(1 + \frac{2B}{\varepsilon}\right)^d \leq \left(\frac{3B}{\varepsilon}\right)^d.
$$

Let $\mathcal{C}_{w,\varepsilon}$ be an $\varepsilon$-cover of $\{w \in \mathbb{R}^d \mid \|w\| \leq B\}$. For any $V_1 \in \mathcal{V}$, there exists $w_2 \in \mathcal{C}_{w,\varepsilon}$ such that $V_2$ parameterized by $w_2$ satisfies $\mathrm{dist}(V_1, V_2) \leq \varepsilon$. Thus, we have,

$$
\log N_{\mathcal{V},\varepsilon} \leq \log N_{w,\varepsilon} \leq d\log(1 + 2B/\varepsilon) \leq d\log(3B/\varepsilon),
$$

which concludes the proof. $\qquad\square$

**Lemma E.11.** *Let $\mathcal{V}$ denote a class of functions mapping from $\mathcal{S}$ to $\mathbb{R}$ with the following parametric form*

$$
V(\cdot) = \max_{a \in \mathcal{A}} \min \left\{ \max_{m \in [M]} \phi(\cdot, a)^\top w^m, H \right\},
$$

*where the parameter $w^m$ satisfies $\|w^m\| \leq B$ for all $m \in [M]$, and for all $(x,a) \in \mathcal{S} \times \mathcal{A}$, we have $\|\phi(x,a)\| \leq 1$. Let $N_{\mathcal{V},\varepsilon}$ be the $\varepsilon$-covering number of $\mathcal{V}$ with respect to the distance $\mathrm{dist}(V, V') = \sup_x |V(x) - V'(x)|$. Then*

$$
\log N_{\mathcal{V},\varepsilon} \leq dM \log\left(1 + 2B/\varepsilon\right) \leq dM \log\left(3B/\varepsilon\right).
$$

*Proof of Lemma E.11.* The proof is analogous to that of Lemma E.10. We provide the detailed proof for completenes.

Consider any two functions $V_1, V_2 \in \mathcal{V}$ with

$$
V_1 = \max_{a \in \mathcal{A}} \min \left\{ \max_{m \in [M]} \phi(\cdot, a)^\top w_1^m, H \right\}
$$

and

$$
V_2 = \max_{a \in \mathcal{A}} \min \left\{ \max_{m \in [M]} \phi(\cdot, a)^\top w_2^m, H \right\}.
$$

Since both $\min\{\cdot, H\}$ and $\max_a$ are contraction maps, we have

$$
\begin{aligned}
\mathrm{dist}(V_1, V_2) &\leq \sup_{x,a} \left| \max_{m \in [M]} \phi(x,a)^\top w_1^m - \max_{m \in [M]} \phi(x,a)^\top w_2^m \right| \\
&\leq \sup_{\phi:\|\phi\|\leq 1} \left| \max_{m \in [M]} \phi^\top w_1^m - \max_{m \in [M]} \phi^\top w_2^m \right| \\
&\leq \sup_{\phi:\|\phi\|\leq 1} \max_{m \in [M]} \|\phi\|_2 \|w_1^m - w_2^m\|_2 \\
&\leq \max_{m \in [M]} \|w_1^m - w_2^m\|_2,
\end{aligned}
\tag{53}
$$

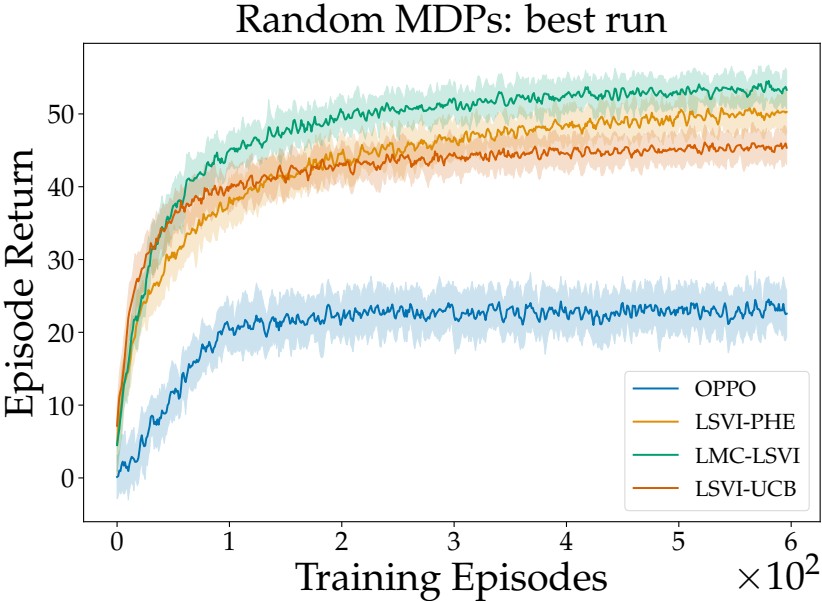

Figure 4: Comparison of LMC-LSVI, OPPO (Cai et al., 2020), LSVI-UCB (Jin et al., 2020) and LSVI-PHE (Ishfaq et al., 2021) in randomly generated non-stationary linearly parameterized MDPs with 10 states, 4 actions, horizon length $H = 100$ and a sparse transition matrix.

For any $m \in [M]$, let $\mathcal{C}^m$ be an $\varepsilon$-cover of $\{w^m \in \mathbb{R}^d \mid \|w^m\| \leq B\}$ with respect to the 2-norm. By Lemma E.9, we know,

$$|\mathcal{C}^m| \leq \left(1 + \frac{2B}{\varepsilon}\right)^d \leq \left(\frac{3B}{\varepsilon}\right)^d.$$

It holds that $N_{\mathcal{V},\varepsilon} \leq \prod_{m=1}^{M} |\mathcal{C}^m|$. Thus, we have,

$$\log N_{\mathcal{V},\varepsilon} \leq \log \prod_{m=1}^{M} |\mathcal{C}^m| \leq dM \log(1 + 2B/\varepsilon) \leq dM \log(3B/\varepsilon),$$

which concludes the proof. □

## F    EXPERIMENT DETAILS

In this section, first, we provide experiments for LMC-LSVI in randomly generated linear MDPs and the riverswim environment (Strehl and Littman, 2008; Osband et al., 2013) and compare it against provably efficient algorithms designed for linear MDPs. Then, we provide more implementation details about experiments in $N$-Chain and Atari games. In total, all experiments (including hyperparameter tuning) took about 2 GPU (V100) years and 20 CPU years.

### F.1    EXPERIMENTS FOR LMC-LSVI

#### F.1.1    RANDOMLY GENERATED LINEAR MDPS

In this section, we use randomly generated non-stationary and linearly parameterized MDPs with 10 states, 4 actions, horizon length of $H = 100$ and a spares transition matrix. As a training setup, we use 4 randomly generated linear MDPs. For each MDP, we use 5 seeds for a total of 20 runs per hyperparameter combination. In Figure 4, we compare our proposed LMC-LSVI against OPPO (Cai et al., 2020), LSVI-UCB (Jin et al., 2020) and LSVI-PHE (Ishfaq et al., 2021).

### F.1.2 THE RIVERSWIM ENVIRONMENT

In the Riverswim environment, there are $N$ states which are lined up in a chain. Figure 5 depicts the case when $N = 6$. The agent begins in the leftmost state $s_1$ and in each state can take one of the two actions – "left" or "right". Swimming to the left, with the current, deterministically moves the agent to the left while swimming to the right against the current often fails. The optimal policy is to swim to the right and reach to the rightmost state $s_N$. Thus, deep exploration is required to obtain the optimal policy in this environment. We experiment with the variant of RiverSwim where $N = 12$ and $H = 40$. We use LSVI-UCB (Jin et al., 2020), LSVI-PHE (Ishfaq et al., 2021), OPPO (Cai et al., 2020) and OPT-RLSVI (Zanette et al., 2020a) as baselines. As shown in Figure 6a, LMC-LSVI achieves similar performace to LSVI-UCB and outperforms other baselines. Figure 6b shows the performance of LMC-LSVI as we vary the update number $J_k$. As we see, even with a relatively small value of $J_k$, LMC-LSVI manages to learn a near-optimal policy quickly.

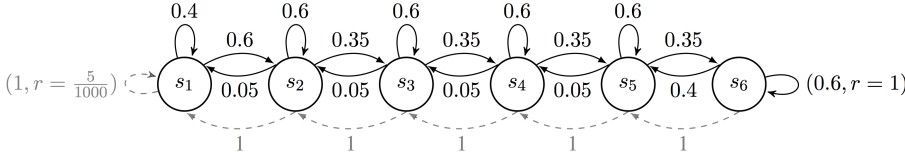

Figure 5: The 6 state RiverSwim environment from Osband et al. (2013). Here, state $s_1$ has a small reward while state $s_6$ has a large reward. The dotted arrows represent the action "left" and deterministically move the agent to the left. The continuous arrows denote the action "right" and move the agent to the right with a relatively high probability. This action represents swimming against the current, hence the name RiverSwim.

### F.2 $N$-CHAIN

There are two kinds of input features $\phi_{1hot}(s_t) = (\mathbf{1}\{x = s_t\})$ and $\phi_{therm}(s_t) = (\mathbf{1}\{x \leq s_t\})$ in $\{0, 1\}^N$. Osband et al. (2016b) found that $\phi_{therm}(s_t)$ has lightly better generalization. So following Osband et al. (2016b), we use $\phi_{therm}(s_t)$ as the input features.

For both DQN and Adam LMCDQN, the Q function is parameterized with a multi-layer perceptron (MLP). The size of the hidden layers in the MLP is $[32, 32]$, and $ReLU$ is used as the activation function. Both algorithms are trained for $10^5$ steps with an experience replay buffer of size $10^4$. We measure the performance of each algorithm by the mean return of the last 10 test episodes. The mini-batch size is 32, and we update the target network for every 100 steps. The discount factor $\gamma = 0.99$.

DQN is optimized by Adam, and we do a hyper-parameter sweep for the learning rate with grid search. Adam LMCDQN is optimized by Adam SGLD with $\alpha_1 = 0.9$, $\alpha_2 = 0.99$, and $\lambda_1 = 10^{-8}$. For Adam LMCDQN, besides the learning rate, we also sweep the bias factor $a$, the inverse temperature $\beta_k$, and the update number $J_k$. We list the details of all swept hyper-parameters in Table 2.

Table 2: The swept hyper-parameter in $N$-Chain.

| HYPER-PARAMETER | VALUES |
|---|---|
| LEARNING RATE $\eta_k$ | $\{10^{-1}, 3 \times 10^{-2}, 10^{-2}, 3 \times 10^{-3}, 10^{-3}, 3 \times 10^{-4}, 10^{-4}\}$ |
| BIAS FACTOR $a$ | $\{1.0, 0.1, 0.01\}$ |
| INVERSE TEMPERATURE $\beta_k$ | $\{10^{16}, 10^{14}, 10^{12}, 10^{10}, 10^{8}\}$ |
| UPDATE NUMBER $J_k$ | $\{1, 4, 16, 32\}$ |

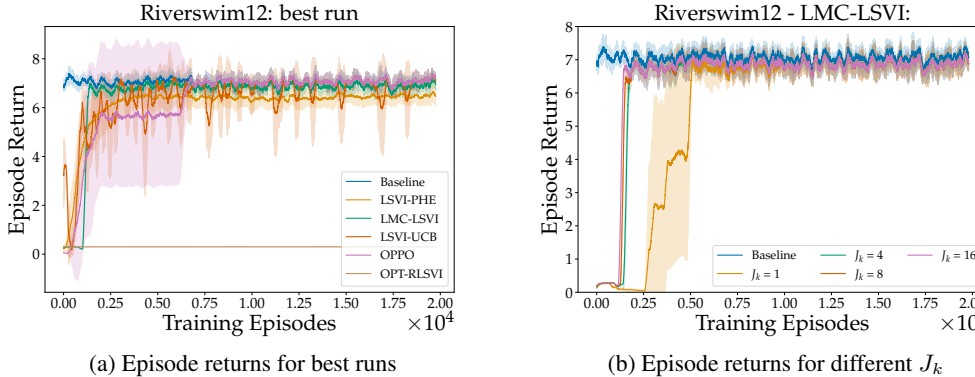

(a) Episode returns for best runs

(b) Episode returns for different $J_k$

Figure 6: Experiment in the Riverswim environment [Strehl & Littman, 2008] with chain length 12. (a) Mean episode returns for best runs, after hyperparameter optimization. (b) Mean episode returns for different values of $J_k$ in LMC-LSVI.

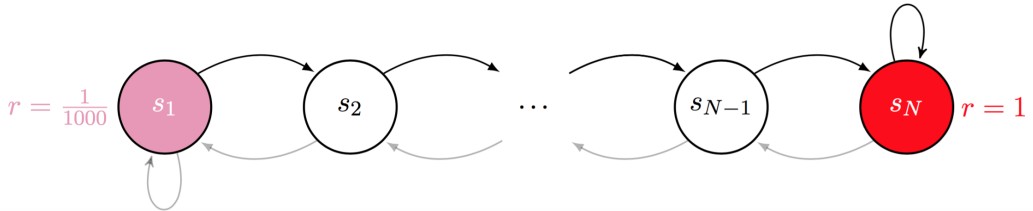

Figure 7: N-Chain environment Osband et al. (2016b).

### F.3 ATARI

#### F.3.1 EXPERIMENT SETUP

We implement DQN, Bootstrapped DQN, Noisy-Net and Adam LMCDQN with tianshou framework (Weng et al., 2022). For the other five baseline algorithms, we take the results from DQN Zoo (Quan and Ostrovski, 2020)[3]. Both DQN and Adam LMCDQN use the same network structure, following the same observation process as in Mnih et al. (2015). To be specific, the observation is 4 stacked frames and is reshaped to $(4, 84, 84)$. The raw reward is clipped to $\{-1, 0, +1\}$ for training, but the test performance is based on the raw reward signals.

Unless mentioned explicitly, we use most of the default hyper-parameters from tianshou's DQN [4]. For each task, there is just one training environment to reduce the exploration effect of training in multiple environments. There are 5 test environments for a robust evaluation. The mini-batch size is 32. The buffer size is $1M$. The discount factor is $0.99$.

For DQN, we use the $\epsilon$-greedy exploration strategy, where $\epsilon$ decays linearly from $1.0$ to $0.01$ for the first $1M$ training steps and then is fixed as $0.05$. During the test, we set $\epsilon = 0$. The DQN agent is optimized by Adam with a fixed learning rate $10^{-4}$.

For our algorithm Adam LMCDQN, since a large $J_k$ significantly increases training time, so we set $J_k = 1$ so that all experiments can be finished in a reasonable time. The Adam LMCDQN agent is optimized by Adam SGLD with learning rate $\eta_k = 10^{-4}$, $\alpha_1 = 0.9$, $\alpha_2 = 0.99$, and $\lambda_1 = 10^{-8}$. We do a hyper-parameter sweep for the bias factor $a$ and the inverse temperature $\beta_k$, as listed in Table 3

---

[3]https://github.com/deepmind/dqn_zoo/blob/master/results.tar.gz
[4]https://github.com/thu-ml/tianshou/blob/master/examples/atari/atari_dqn.py

Table 3: The swept hyper-parameter in Atari games.

| HYPER-PARAMETER | VALUES |
|---|---|
| BIAS FACTOR $a$ | $\{1.0, 0.1, 0.01\}$ |
| INVERSE TEMPERATURE $\beta_k$ | $\{10^{16}, 10^{14}, 10^{12}\}$ |

### F.3.2 ADDITIONAL RESULTS

Our implementation of Adam LMCDQN applies double Q networks by default. In Figure 8, we compare the performance of Adam LMCDQN with and without applying double Q functions. The performance of Adam LMCDQN is only slightly worse without using double Q functions, proving the effectiveness of our approach. Similarly, there is no significant performance difference for Langevin DQN (Dwaracherla and Van Roy, 2020) with and without double Q functions, as shown in Figure 9.

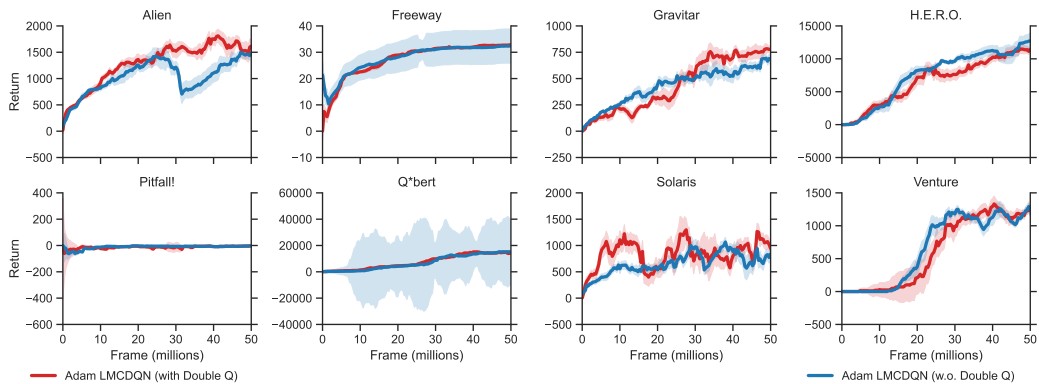

Figure 8: The return curves of Adam LMCDQN in Atari over 50 million training frames, with and without double Q functions. Solid lines correspond to the median performance over 5 random seeds, while shaded areas correspond to 90% confidence interval. The performance of Adam LMCDQN is only slightly worse without using double Q functions, proving the effectiveness of our approach.

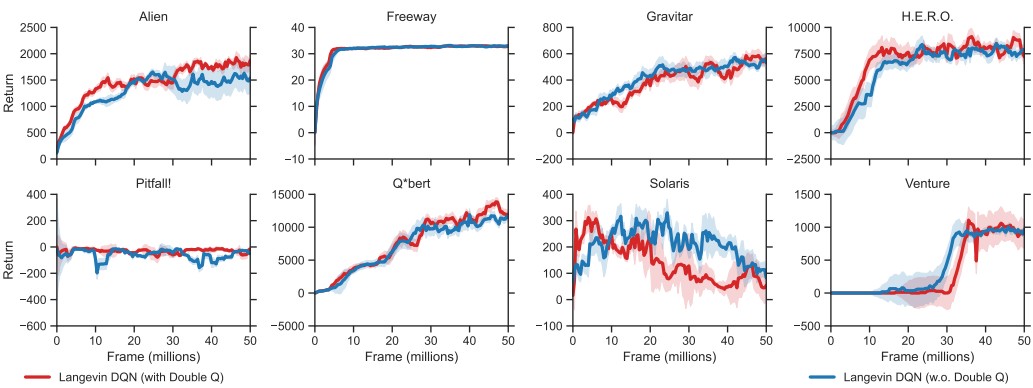

Figure 9: The return curves of Langevin DQN in Atari over 50 million training frames, with and without double Q functions. Solid lines correspond to the median performance over 5 random seeds, while shaded areas correspond to 90% confidence interval. There is no significant performance improvement by applying double Q functions in Langevin DQN.

Moreover, we also compare Langevin DQN with our algorithm Adam LMCDQN in Figure 10. Both algorithms incorporate the double Q trick by default. Overall, Adam LMCDQN usually outperforms Langevin DQN in sparse-reward hard-exploration games, such as Gravitar, Solaris, and Venture,

while in dense-reward hard-exploration games, such as Alien, H.E.R.O and Qbert, Adam LMCDQN and Langevin DQN achieve similar performance.

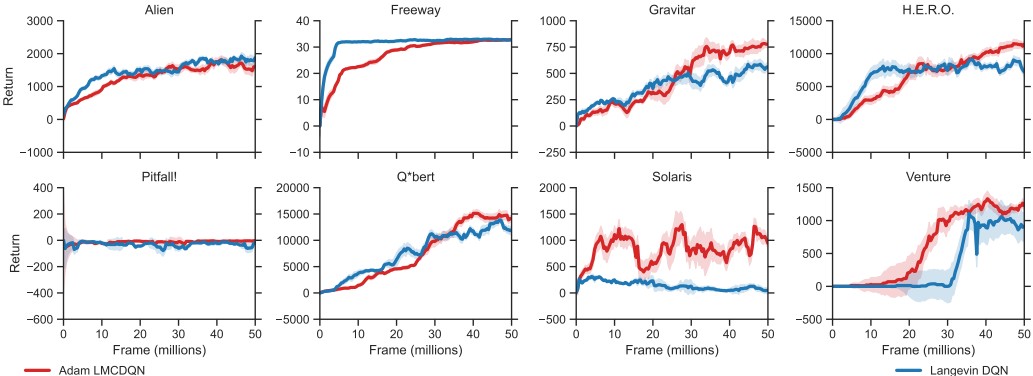

Figure 10: The return curves of Adam LMCDQN and Langevin DQN in Atari over 50 million training frames. Solid lines correspond to the median performance over 5 random seeds, while shaded areas correspond to $90\%$ confidence interval.

