# OpenReview forum: "Provable and Practical: Efficient Exploration in Reinforcement Learning via Langevin Monte Carlo"
_ICLR.cc/2024/Conference — ICLR 2024 poster_

### Official Review · Reviewer_dAXr · 2023-10-28

**Soundness:** 3 good
**Presentation:** 3 good
**Contribution:** 3 good
**Rating:** 8
**Confidence:** 3

**Summary:**

The paper is about online RL algorithm design and theoretical analysis. Different from most previous RL theory works which lack deep RL demonstrations, this work proposes a both practical (scalable to deep RL domains) and provably efficient online RL algorithm (LMC-LSVI) based on the celebrated Langevin Monte Carlo algorithm. Theoretically, it proves that with linear function approximation LMC-LSVI achieves a $\widetilde{\mathcal{O}}(d^{3/2}H^{5/2}T^{1/2})$-online regret. On the practical side, LMC-LSVI is further extended to the Adam LMCDQN algorithm which performs similarly or even better than SOTA explorative deep RL algorithms in some challenging RL domains.

**Strengths:**

1. Bridging the gap between RL theory and practice is of great importance to the advance of RL research. This work gives a possible and positive answer to this question in the specific setting of online RL where exploration-exploitation balance is a key problem.
2. The proposed  Langevin Monte Carlo Least-Squares Value Iteration (LMC-LSVI) algorithm turns out to have a quite clean form which simply adds a noise term to the gradient descent update of the Bellman error (Line 9 of Algorithm 1) to incentivize exploration. This advantage thus allows for a deep RL extension where Adam-based adaptive SGLD is further applied.
3. The proposed algorithm enjoys theoretical guarantees (in the linear function approximation setting) which is missing in most previous deep RL exploration methods even for LFAs.

**Weaknesses:**

The rate of the online regret in linear function approximation setting is far from tight compared with known lower bounds. But from my view this is understandable given that a new approach is derived whose practicability is of higher importance.

**Questions:**

1. Regarding the theoretical analysis (Theorem 4.2), I am curious why the failure probability $\delta$ must be larger than a certain quantity, say $1/(2\sqrt{2e\pi})$? Is this inevitable for a sampling-stype algorithm and analysis? This will narrow the applicability of the theory since for frequentist regret analysis we always hope that the regret bound can hold for arbitrarily small fail probability.
2. The authors say in the contribution part that "unlike any other provably efficient algorithms for linear MDPs, it can easily be extended to deep RL settings",..., "such unification of theory and practice is unique in the current literature of both theoretical RL and deep RL", which to my knowledge is overclaimed. Even though the perspective of Langevin dynamic are less explored in this regime, which is done by this paper, there do exist other works trying to achieve sample efficiency while being compatible with practical deep RL methods, e.g., [1, 2, 3]. So it seems improper to describe the work as unique given this line of research.

**References:**

[1] Feng, Fei, et al. "Provably Correct Optimization and Exploration with Non-linear Policies." *International Conference on Machine Learning*. PMLR, 2021.

[2] Kitamura, Toshinori, et al. "Regularization and Variance-Weighted Regression Achieves Minimax Optimality in Linear MDPs: Theory and Practice." *International Conference on Machine Learning*. PMLR, 2023.

[3] Liu, Zhihan, et al. "One Objective to Rule Them All: A Maximization Objective Fusing Estimation and Planning for Exploration." *arXiv preprint arXiv:2305.18258* (2023).

---

> ### Author Response · Authors · 2023-11-15
> **Response to Reviewer dAXr**
>
> We thank the reviewer for their valuable time and effort in providing detailed feedback on our work. We hope our response will fully address all of the reviewer's points.
>
> ---
> ### Tightness of regret bound.
>
> We thank the reviewer for pointing this out. After the submission, we realized that a minimal change in the proof of Lemma B.4 (Lemma B.5 in the original submission) results in improving the bound by a factor of $H$. We refer the reviewer to our **Overall Response** for more detail on this improvement. Regarding the dependency on $d$, as mentioned in the main paper, *”the gap of $\sqrt{d}$ in worst-case regret between UCB and TS-based methods is a long standing open problem, even in a simpler setting of linear bandit.*” due to Hamidi and Bayati (2020) and is yet to be addressed. In the revised version, LMC-LSVI achieves a $\tilde{O}(d^{3/2}H^{3/2}\sqrt{T})$ online regret which matches **the best known** bound for any randomized algorithm for linear MDP.
>
> ---
> ### Why must the failure probability $\delta$ be larger than a certain quantity?
>
> Indeed, for the frequentist regret analysis we would ideally hope that the regret bound can hold for a small failure probability. This mentioned interval arises from Lemma B.7 (Lemma B.8 in the original submission), where we get an optimistic estimation with a constant probability of $\frac{1}{2\sqrt{2e\pi}}$. This is addressed and improved by using the **optimistic reward sampling** scheme proposed in [4].  Concretely, we can generate $M$ estimates for Q function $\\{Q\_h^{k, m}\\}\_{m \\in[M]}$ through maintaining $M$ samples of $w$: $\\{w\_h^{k, J\_k, m}\\}\_{m \\in [M]}$ throughout the LMC. Then, we can make an optimistic estimate of Q function by setting $Q\_h^{k}(\\cdot, \\cdot)= \\min\\{\\max\_{m \\in [M]}\\{Q\_h^{k, m}(\\cdot, \\cdot)\\}, H-h+1\\}.$ However, for the simplicity of the algorithm design we keep the current algorithm. We added a discussion of this in the revised version. Please refer to Remark 4.3 on page 5.
>
> ---
> ### There are other works trying to achieve sample efficiency while being compatible with practical deep RL methods.
>
> We thank the reviewer for bringing these works to our attention, as they were inadvertently overlooked in our literature survey. We have cited them in our revision and slightly revised our discussion accordingly. The revised paragraph looks as follows:
>
> ```
> Unlike many other provably efficient algorithms with linear function approximations (Yang and Wang, 2020; Cai et al., 2020; Zanette et al., 2020a; Ayoub et al., 2020; Zanette et al., 2020b; Zhou et al., 2021; He et al., 2023), LMC-LSVI can easily be extended to deep RL settings (Adam LMCDQN). We emphasize that such unification of theory and practice is rare (Feng et al., 2021; Kitamura et al., 2023; Liu et al., 2023) in the current literature of both theoretical RL and deep RL.
> ```
>
> We hope we have addressed all of your questions/concerns. If you have any further questions, we would be more than happy to answer them. Finally, we are grateful for the reviewer’s positive support on our paper. Given our updated regret bound and comment regarding how to address the failure probability constraint, if the reviewer thinks the work is worthy of a higher score to be highlighted, we would be immensely grateful.
>
>
> [1] Feng, Fei, et al. "Provably Correct Optimization and Exploration with Non-linear Policies." International Conference on Machine Learning. PMLR, 2021.
>
> [2] Kitamura, Toshinori, et al. "Regularization and Variance-Weighted Regression Achieves Minimax Optimality in Linear MDPs: Theory and Practice." International Conference on Machine Learning. PMLR, 2023.
>
> [3] Liu, Zhihan, et al. "One Objective to Rule Them All: A Maximization Objective Fusing Estimation and Planning for Exploration." arXiv preprint arXiv:2305.18258 (2023).
>
> [4] Haque Ishfaq, Qiwen Cui, Viet Nguyen, Alex Ayoub, Zhuoran Yang, Zhaoran Wang, Doina Precup, and Lin Yang. Randomized exploration in reinforcement learning with general value function approximation. In International Conference on Machine Learning, 2021.

---

> > ### Author Response · Authors · 2023-11-17
> > **Additional result on how to remove the interval constraint on $\delta$ in Theorem 4.2**
> >
> > Once again, we thank the reviewer for their positive support on our paper and engaging questions. We have further updated the paper (Appendix F, Page 41- 45 in revision) with in depth description and theoretical results on how to remove the interval constraint on $\delta$ in Theorem 4.2 by incorporating the optimistic reward sampling scheme proposed in [1] into LMC-LSVI. We call this variant Multi-Sampling LMC-LSVI or MS-LMC-LSVI in short. We believe with this additional detail, the paper further provides a complete picture of Langevin Monte Carlo for linear MDP.
> >
> > In Appendix F.1, we expanded on Remark 4.3 and described in detail how with minimal modification of LMC-LSVI (Algorithm 1), we can get MS-LMC-LSVI.
> >
> > In Appendix F.2, we laid out the supporting lemmas used to prove its regret bound. Each supporting lemma has one to one correspondence with supporting lemmas in Appendix B.1 that are used to prove the regret bound for LMC-LSVI (Algorithm 1). Moreover, for most of them we omit the proofs as they are identical to that of corresponding lemmas in Appendix B.1. We provided some details where necessary for clarity. The only notable difference is in Lemma F.5 (corresponding to Lemma B.7 for LMC-LSVI) on Page 43. After following the same proof and conclusion of Lemma B.7, we derive some probabilistic inequalities and apply union bound to complete the proof.
> >
> > In Appendix F.3, we provide the final regret bound result (Theorem F.7) for MS-LMC-LSVI that holds with probability $1-\delta$ for any $\delta \in (0,1)$. The proof is the same as that of Theorem B.8 with some minor differences at the end when we put everything together using union bound.
> >
> >
> > [1] Haque Ishfaq, Qiwen Cui, Viet Nguyen, Alex Ayoub, Zhuoran Yang, Zhaoran Wang, Doina Precup, and Lin Yang. Randomized exploration in reinforcement learning with general value function approximation. In International Conference on Machine Learning, 2021.

---

> > > ### Author Response · Authors · 2023-11-21
> > > **Thank you for your review**
> > >
> > > Dear Reviewer,
> > >
> > > Thank you so much for taking the time to review our paper. We sincerely appreciate your detailed feedback and appraisal of our work, and have carefully addressed your comments in our response and the revised manuscript. With only a day remaining for the discussion period, we would be grateful if you could acknowledge receipt of our responses and let us know if you have further questions. We are eager to engage in any further discussions if needed.
> > >
> > > Thanks.

---

> > > > ### Comment · Reviewer_dAXr · 2023-11-22
> > > >
> > > > Thanks the authors for the very detailed response to my questions and concerns, and I am glad to see that the regret of the algorithm has been improved to the best known bound. I have no further concerns and I will keep my rate as 8.

---

> > > > > ### Author Response · Authors · 2023-11-22
> > > > > **Thank you for reading our response and consistent support of our paper**
> > > > >
> > > > > We thank the reviewer for reading our response and we are glad that our response addressed all the questions and concerns of the reviewer. Finally, we thank the reviewer again for their consistent support of our paper.

---

### Official Review · Reviewer_SrMF · 2023-11-01

**Soundness:** 3 good
**Presentation:** 3 good
**Contribution:** 3 good
**Rating:** 6
**Confidence:** 2

**Summary:**

They have introduced online RL algorithms incorporating Langevin Monte Carlo.

   *They have established the regret bound for Q-learning with Langevin Monte Carlo under linear MDPs.

   *They have adapted the aforementioned algorithm to use the ADAM optimizer, demonstrating its favorable empirical performance through experiments.

**Strengths:**

While the literature on Bayesian model-free algorithms is extensive, it is indeed lacking in works that offer both robust theoretical guarantees and practical viability. In my view, this paper effectively bridges this gap.

* The paper presents a good comparison, highlighting noteworthy contributions compared to existing works.

* Section 3 offers a non-trivial and novel analysis that significantly enhances the paper.

* The practical version's experimental results in Section 5 demonstrate promising performance.

**Weaknesses:**

* It looks Algorithms they use in practice (Algorithm 2) are not analyzed. So, this algorithm is "practical", but I am not sure it is fair to say this is "provable."

* Some aspects of the comparisons appear to lack complete fairness.

   * In Table 1, it is unclear what precisely the author intends to convey with "computational traceability" and "scalability." These terms should be defined more formally in the caption of the table.

   * Furthermore, it may not be entirely fair to say that OPT-RLSVI and LSVI-PHE lack scalability in Table 1. While I acknolwedge that the proposed algorithms (OPT-RLSVI and LSVI-PHE) may not perform well in practice, one could argue that they could exhibit scability with the incorporation of certain simple heuristics, even if formal guarantees are absent. Indeed, the LSVI-PHE paper includes moderate experiments. The author has similarly extended theoretically justified algorithms to practical versions without formal backing as they did. So, in this sense, I am not sure why it is reasonable to say the author's algorithm is scalable, but their algorithms are not scalable.

**Questions:**

I raised several concerns for the weakness section.

---

> ### Author Response · Authors · 2023-11-15
> **Response to Reviewer SrMF**
>
> We thank the reviewer for their valuable time and effort in providing detailed feedback on our work. We hope our response will fully address all of the reviewer's points.
>
> ---
> ### 1. Why do you say the method is provable and practical?
>
> We would like to clarify that in our submission, we claim LMC-LSVI (Algorithm 1) as provable and propose Adam LMCDQN (Algorithm 2) as a *”practical variant”* of LMC-LSVI which is more well-suited for deep RL due to its usage of Adam based adaptive SGLD. Moreover, the proposed two algorithms can be viewed as a whole framework, which shows us how to provably conduct randomized exploration (under linear function approximation), and is easily generalized to practical implementation.
>
> While it’s interesting to have convergence and regret guarantee for Adam LMCDQN (Algorithm 2), as shown in [1], Adam has issues in convergence and theoretically it has been shown that it may not converge to an optimal solution. We propose the Adam based extension of LMC-LSVI due to its many observed empirical advantages, and leave the theoretical convergence of it for future study.
>
> [1] Sashank J. Reddi, Satyen Kale, and Sanjiv Kumar. On the convergence of Adam and beyond. In Proceedings of the 6th International Conference on Learning Representations, 2018.
>
> ---
> ### 2. What are the definitions of computational tractability and scalability in Table 1?
>
> Thanks a lot for your suggestion. By computational tractability in Table 1, we meant whether the algorithm is “implementable”. By “scalability” we mean whether the algorithm can be easily extended to deep RL setup. In the revision, we precisely defined these terms more formally in the caption of Table 1. The caption currently says the following:
>
> ```
> Regret upper bound for episodic, non-stationary, linear MDPs. Here, computational tractability refers to the ability of a computational problem to be solved in a reasonable amount of time using a feasible amount of computational resources. An algorithm is scalable if it can be easily extended to deep RL setup.
> ```
>
> ---
> ### 3. Why do you say OPT-RLSVI and LSVI-PHE lack scalability in Table 1?
>
> Thanks for asking the questions and providing your insight on the comparison. We added more discussion in Remark 6.1 and the caption of Table 1 to explain our comparison.
>
> Note that the proposed LMC-LSVI algorithm can be easily extended to deep RL by plugging in neural network function approximation of the Q-function, without any change of the algorithm framework. In contrast, one major disadvantage of LSVI-PHE is that it needs to maintain many ensembles of Q-function approximators that are fitted using independent copies of perturbed datasets at each round. This is computationally heavy and memory-wise extremely burdensome. Without changing the algorithm framework of LSVI-PHE, it will be computationally impractical for environments like Atari, for which we may require many ensemble models. Regarding OPT-RLSVI, we believe even with heuristics, it is not clear how to provide its practical implementation for the reasons outlined in Remark 6.1. Concretely, OPT-RLSVI relies on the feature norm with respect to the inverse covariance matrix. When the features are updated over iterations in deep RL, the computational complexity becomes unbearable as it needs to compute the feature covariance matrix repeatedly.
>
> ---
> We hope we have addressed all of your questions/concerns. If you have any further questions, we would be more than happy to answer them and if you don’t, would you kindly consider increasing your score?

---

> > ### Author Response · Authors · 2023-11-21
> > **Thank you for your review**
> >
> > Dear Reviewer,
> >
> > Thank you so much for taking the time to review our paper. We sincerely appreciate your detailed feedback and appraisal of our work, and have carefully addressed your comments in our response and the revised manuscript. With only a day remaining for the discussion period, we would be grateful if you could acknowledge receipt of our responses and let us know if you have further questions. We are eager to engage in any further discussions if needed.
> >
> > Thanks.

---

> > > ### Comment · Reviewer_SrMF · 2023-11-22
> > >
> > > I checked the response.
> > >
> > > >> We would like to clarify that in our submission, we claim LMC-LSVI (Algorithm 1) as provable and propose Adam LMCDQN (Algorithm 2) as a ”practical variant” of LMC-LSVI which is more well-suited for deep RL due to its usage of Adam based adaptive SGLD.
> > >
> > > Then, the current title might be provable "or" practical. But, not "and". Usually. when we use "and", we would think the same algorithm can achieve both properties.  A current title sounds overclaiming.
> > >
> > > >> By “scalability” we mean whether the algorithm can be easily extended to deep RL setup."
> > >
> > > I generally recommend that you remove the "scalability" column after reading your response again. We can take many arbitrary interpretations for "easily extended". This is perceived as an unfair comparison for many readers.  If you want to convey this kind of stuff, the author can just claim in a sentence that we implement our algorithm in these complicated deep environments, where it is unclear whether existing works (cite) work, or something.  But I think this simple table is biased and misleading.
> > >
> > > >> Answer to "Why do you say OPT-RLSVI and LSVI-PHE lack scalability in Table 1?"
> > >
> > > I am still not convinced. Did you check your statement with the authors?  I think they would claim their algorithms can be "easily" extended. For example, this sounds like a biased statement.
> > >
> > > * Note that the proposed LMC-LSVI algorithm can be easily extended to deep RL by plugging in neural network function approximation of the Q-function, without any change of the algorithm framework.  ---> But, you changed anyway as well. We can consider many ways to plug in. So, I am not sure this is treated as "easily extended". Whether it is easy or not is more subjective.
> > >
> > > *  "Without changing the algorithm framework of LSVI-PHE, it will be computationally impractical for environments like Atari, for which we may require many ensemble models".  ---> Did you check with them? Or did you run experiments?

---

> > > > ### Author Response · Authors · 2023-11-23
> > > > **We removed the scalability column from Table 1.**
> > > >
> > > > We thank the reviewer for reading our response and providing constructive suggestions.
> > > >
> > > > ---
> > > > ### 1. Removing the column “scalability” and updating description of OPT-RLSVI and LSVI-PHE
> > > >
> > > > Per the reviewer’s suggestion, we have removed the “scalability” column from Table 1. We also updated the discussion in Remark 6.1 on OPT-RLSVI and LSVI-PHE so that it sounds more unbiased. The revised Remark 6.1 looks as follows:
> > > >
> > > > “We note that in our experiments in this section, as baselines, we use commonly used algorithms from deep RL literature as opposed to methods presented in Table 1. This is because while these methods are provably efficient under linear MDP settings, in most cases, it is not clear how to scale them to deep RL settings. More precisely, these methods assume that a good feature is known in advance and Q values can be approximated as a linear function over this feature. If the provided feature is not good and fixed, the empirical performance of these methods is often poor. For example, LSVI-UCB (Jin et al., 2020)  computes UCB bonus function of the form $\|\phi(s,a)\|_{\Lambda^{-1}}$, where $\Lambda \in \mathbb{R}^{d\times d}$ is the empirical feature covariance matrix. When we update the feature over iterations in deep RL, the computational complexity of LSVI-UCB becomes unbearable as it needs to repeatedly compute the feature covariance matrix to update the bonus function. In the same vein, while estimating the Q function, OPT-RSLVI (Zanette et al., 2020a)  needs to rely on the feature norm with respect to the inverse covariance matrix.  Lastly, even though LSVI-PHE (Ishfaq et al., 2021) is computationally implementable in deep RL settings, it requires sampling independent and identically distributed (i.i.d.) noise for the whole history every time to perturb the reward, which appears to be computationally burdensome in most practical settings.”
> > > >
> > > > ---
> > > > If you have any further questions, we would be more than happy to answer them and if you don’t, would you kindly consider increasing your score?

---

### Official Review · Reviewer_yHqj · 2023-11-06

**Soundness:** 2 fair
**Presentation:** 3 good
**Contribution:** 3 good
**Rating:** 6
**Confidence:** 4

**Summary:**

The paper studies usefulness of LMC algorithms for MDPs. It first shows an LMC algorithm deployed with LSVI for linear MDPs and upper bounds the corresponding regret. It also proposes versions of LMC used with Adam and DQN for practical purposes. The performance of the practical algorithm is illustrated on multiple atari games.

**Strengths:**

1. The paper proposes a regret upper bound for applying LMC with LSVI for linear MDPs.
2. The paper proposes a practical version of LMC to be applied with DQN. It works in practice as shown through multiple experiments.

**Weaknesses:**

1. The theoretical analysis is shown for linear MDPs and the practical algorithm is applied with DQN. Thus, the theory and applications serve two different purposes and does not compliment each other. It makes the paper look like an assortment of results for different settings than a coherent study.
2. If the aim is to design a practical algorithm, why analysing it for linear MDPs, which is known to be unfit for practice (the version stated in the paper) [1]? Why not analysing it for more practical models like [1], [2], [3]?
3. The regret bound for LMC lsvi is loose in terms of both d and H. Why is it so? Is it due to any fundamental hardness in analysis or inherent to LMC algorithms or just a shortcoming of the analysis done in the paper? Can you explain this?
4. The practical version is claimed to be better than the existing LMC algorithm for RL cause the proposal has theoretical guarantees and also employs better practical techniques. The first is not completely valid as the setting plus algorithm for analysis and practice are really different. The second is also not clear as the experimental results leave the question of performance improvement statistically inconclusive . Can you provide a reason where adam LMC DQN would work better than langevin DQN and where it would be worse?

[1] Zhang, Tianjun, Tongzheng Ren, Mengjiao Yang, Joseph Gonzalez, Dale Schuurmans, and Bo Dai. "Making linear mdps practical via contrastive representation learning." In International Conference on Machine Learning, pp. 26447-26466. PMLR, 2022.
[2] Ouhamma, Reda, Debabrota Basu, and Odalric Maillard. "Bilinear exponential family of MDPs: frequentist regret bound with tractable exploration & planning." In Proceedings of the AAAI Conference on Artificial Intelligence, vol. 37, no. 8, pp. 9336-9344. 2023.
[3] Weisz, Gellért, András György, and Csaba Szepesvári. "Online RL in Linearly $ q^\pi $-Realizable MDPs Is as Easy as in Linear MDPs If You Learn What to Ignore." arXiv preprint arXiv:2310.07811 (2023).

**Questions:**

Please check the weaknesses for questions .

---

> ### Author Response · Authors · 2023-11-15
> **Response to Reviewer yHqj (Part 1)**
>
> We thank the reviewer for their valuable time and effort in providing detailed feedback on our work. We hope our response will fully address all of the reviewer's points.
>
> ---
> ### 1. The theory and applications seem to serve two different purposes?
>
> In this paper, along with providing theoretical analysis of LMC-LSVI (Algorithm 1) under linear MDP, we also provide experimental results for it in Appendix E.1 in simulated linear MDPs and the riverswim environment. We later show that we can readily extend the proposed algorithm to deep RL. This is a favorable property of the proposed algorithms that is inspired by stochastic gradient descent and can be generalized to deep learning settings. The theoretical analysis serves as an inspiration to develop the practical methods.
>
> Concretely, the proposed Adam LMCDQN (Algorithm 2) is an instantiation of LMC-LSVI in deep RL by using Adam based adaptive SGLD with DQN as a backbone. DQN being a canonical value based learning method, it makes sense to incorporate Langevin monte carlo based exploration with it when instantiating LMC-LSVI for deep RL setting.
>
> ---
> ### 2. Why analyzing it for linear MDPs instead of other models?
>
> For the study in the paper, we choose linear MDP for its relative generality and potential to provide insights for practical algorithms. We agree that the mentioned variants are more general; however, they also lack the generality needed for real application. As the reviewer knows, there is always a more general setting to be studied. We would also like to highlight that the theoretical analysis of RL algorithms in the linear MDP setting is itself a challenging task, especially for randomized exploration based algorithms such as LMC-LSVI. Nevertheless, analyzing LMC based approaches in newer models like [1], [2], [3] is certainly an interesting direction for future extensions and we have included them in our revised conclusion (Page 9, Section 8) . We provide a copy of the modified version here:
>
>
> ```
> We proposed the LMC-LSVI algorithm for reinforcement learning that uses Langevin Monte Carlo to directly sample a Q function from the posterior distribution with arbitrary precision. LMC-LSVI achieves the best-available regret bound for randomized algorithms in the linear MDP setting. Furthermore, we proposed Adam LMCDQN, a practical variant of LMC-LSVI, that demonstrates competitive empirical performance in challenging exploration tasks. There are several avenues for future research. It would be interesting to explore if one can improve the suboptimal dependence on H for randomized algorithms. Extending the current results to more practical and general settings (Zhang et al., 2022; Ouhamma et al., 2023; Weisz et al., 2023) is also an exciting future direction.
> ```
>
> ---
> ### 3. Why is the regret of LMC-LSVI loose in terms of both $d$ and $H$?
>
> We thank the reviewer for pointing this out. After the submission, we realized that a minimal change in the proof of Lemma B.4 (Lemma B.5 in the original submission) results in improving the bound by a factor of $H$. We refer the reviewer to our **Overall Response** for more details on this improvement. Regarding the dependency on $d$, as mentioned in the main paper (Section 4, page 5), *”the gap of $\sqrt{d}$ in worst-case regret between UCB and TS-based methods is a long standing open problem, even in a simpler setting of linear bandit.*” due to Hamidi and Bayati (2020) and is yet to be addressed. In the revised version, LMC-LSVI achieves a $\tilde{O}(d^{3/2}H^{3/2}\sqrt{T})$ online regret, which matches **the best known** bound for any randomized algorithm for linear MDP.
>
> Here we provide the updated comment from the revised paper (Section 4, page 5)
>
> “We compare the regret bound of our algorithm with the state-of-the-art results in the literature of theoretical reinforcement learning in Table 1. Compared to the lower bound $\Omega(dH\sqrt{T})$ proved in Zhou et al. (2021), our regret bound is worse off by a factor of $\sqrt{dH}$ under the linear MDP setting. However, the gap of $\sqrt{d}$ in worst-case regret between UCB and TS-based methods is a long standing open problem, even in a simpler setting of linear bandit (Hamidi and Bayati 2020). When converted to linear bandits by setting $H = 1$, our regret bound matches that of LMCTS (Xu et al., 2022) and the best-known regret upper bound for LinTS from Agrawal and Goyal (2013) and Abeille et al. (2017).”

---

> > ### Author Response · Authors · 2023-11-15
> > **Response to Reviewer yHqj (Part 2)**
> >
> > ---
> > ### 4. Where Adam LMC DQN would work better than Langevin DQN and where it would be worse
> >
> > We appreciate the reviewers’ comments. In **Comparison to Dwaracherla and Van Roy (2020)”**, in Appendix A, page 18, we change the narrative to imply that the theoretical study of LMC-LSVI inspires the design of practical algorithms, enabling us to avoid overlooking potential components needed for an efficient algorithm. We emphasize that the theoretical guarantee is not the sole reason for the success of the proposed algorithm but rather an intuition and sanity check.
> >
> > In practice, we observed that Adam LMCDQN usually outperforms Langevin DQN in sparse-reward hard-exploration games, such as Gravitar, Solaris, and Venture; while in dense-reward hard-exploration games, Adam LMCDQN and Langevin DQN achieve similar performance. We emphasize that this statement is entirely based on empirical observation at this point. We updated the **Ablation Study** paragraph in Section 6.2 on page 8 which now reads:
> >
> > ```
> > Ablation Study. In Appendix E.3.2, we also present results for Adam LMCDQN without applying
> > double Q functions. The performance of Adam LMCDQN is only slightly worse without using double Q functions, proving the effectiveness of our approach. Moreover, we implement Langevin DQN (Dwaracherla and Van Roy, 2020) with double Q functions and compare it with our algorithm Adam LMCDQN . Empirically, we observed that Adam LMCDQN usually outperforms Langevin DQN in sparse-reward hard-exploration games, while in dense-reward hard-exploration games, Adam LMCDQN and Langevin DQN achieve similar performance.
> > ```
> >
> > We also updated the corresponding description accordingly in the relevant paragraph on page 18 which reads the following:
> >
> >
> > ```
> > We conducted a comparison in such environments in Appendix E.3.2. Empirically, we observed that Adam LMCDQN usually outperforms Langevin DQN in sparse-reward hard-exploration games, such as Gravitar, Solaris, and Venture; while in dense-reward hard-exploration games such as Alien, H.E.R.O and Qbert, Adam LMCDQN and Langevin DQN achieve similar performance.
> > ```
> >
> > We hope we have addressed all of your questions/concerns. If you have any further questions, we would be more than happy to answer them and if you don’t, would you kindly consider increasing your score?
> >
> >
> > [1] Zhang, Tianjun, Tongzheng Ren, Mengjiao Yang, Joseph Gonzalez, Dale Schuurmans, and Bo Dai. "Making linear mdps practical via contrastive representation learning." In International Conference on Machine Learning, pp. 26447-26466. PMLR, 2022.
> >
> >
> >  [2] Ouhamma, Reda, Debabrota Basu, and Odalric Maillard. "Bilinear exponential family of MDPs: frequentist regret bound with tractable exploration & planning." In Proceedings of the AAAI Conference on Artificial Intelligence, vol. 37, no. 8, pp. 9336-9344. 2023.
> >
> >  [3] Weisz, Gellért, András György, and Csaba Szepesvári. "Online RL in Linearly $q^\pi$ -Realizable MDPs Is as Easy as in Linear MDPs If You Learn What to Ignore." arXiv preprint arXiv:2310.07811 (2023).
> >
> > [4] Nima Hamidi and Mohsen Bayati. On worst-case regret of linear thompson sampling. arXiv preprint arXiv:2006.06790, 2020

---

> > > ### Author Response · Authors · 2023-11-21
> > > **Thank you for your review**
> > >
> > > Dear Reviewer,
> > >
> > > Thank you so much for taking the time to review our paper. We sincerely appreciate your detailed feedback and appraisal of our work, and have carefully addressed your comments in our response and the revised manuscript. With only a day remaining for the discussion period, we would be grateful if you could acknowledge receipt of our responses and let us know if you have further questions. We are eager to engage in any further discussions if needed.
> > >
> > > Thanks.

---

### Author Response · Authors · 2023-11-15
**Overall response**

We would like to thank all reviewers for the insightful and detailed reviews and comments. We are grateful that the reviewers have mostly recognized our major contributions, which are

* **An elegant approach to Thompson Sampling:** We propose the idea of using Langevin Monte Carlo (LMC) for performing approximate sampling from the posterior of Q function allowing us to scale up Thompson sampling based exploration to high dimensional state spaces.

* **Provable guarantee for LMC-LSVI:** We propose  Langevin Monte Carlo Least-Squares Value Iteration (LMC-LSVI) which enjoys provable regret guarantee under linear MDP setting.

* **A deep RL variant of LMC-LSVI:** Bridging the gap between RL theory and practice, we extend LMC-LSVI to deep RL settings and propose Adam LMCDQN where we use Adam based adaptive SGLD approach.

* **Thorough experiments:** We provide thorough experiments in simulated linear MDP setting, riverswim environment, N-chain environment and hard Atari games which require deep exploration capabilities from the agent. We also provide an extensive ablation study showing the importance of every component in our algorithms. (Section 6.2 and Appendix E).

We have addressed the comments from the reviewers and revised the manuscript accordingly. We summarize the main changes done in the revised version of the paper:

---
### 1. We slightly updated the proof of Lemma B.4, which leads to an improved regret bound of LMC-LSVI.

One of the most commonly asked questions is why the regret of LMC-LSVI is worse than other TS based algorithms by a factor of $H$.  In the proof of Lemma B.4 in the revised version (Lemma B.5 in the original submission), previously we set $\varepsilon = \frac{H\sqrt{d}}{K\sqrt{\beta_K}}$. However, after submission, we extensively studied the possible choices of $\varepsilon$ and showed that the choice $\varepsilon = \frac{H}{2\sqrt{2}k}$ results in improving the bound by factor of $H$. This minimal change in the $\varepsilon$ assignment removes the extra $H$ factor in the regret bound. We also updated the places in other lemmas where the bound from Lemma B.4 was used. This change does not affect the proof structure of other lemmas. We emphasize that after this modification, LMC-LSVI matches **the best known** bound of randomized algorithms for linear MDP [1, 2]. We thank the reviewers for their suggestions on this matter.

---
### 2. We formally defined  the terms “computational tractability” and “scalability” in the caption of Table 1 (page 5) based on the suggestion from Reviewer SrMF.

---
### 3. We moved the paragraph **Comparison to Dwaracherla and Van Roy (2020)** from the **Related Work** section to the Appendix A under **Additional Related Work** due to space constraint in the main paper after incorporating suggestions from the reviewers.

---
### 4. We added a remark in Section 4 (Remark 4.3 on page 5) where we outlined how one can devise an algorithm that will allow the bound to hold with arbitrarily small failure probability.


[1] Haque Ishfaq, Qiwen Cui, Viet Nguyen, Alex Ayoub, Zhuoran Yang, Zhaoran Wang, Doina Precup, and Lin Yang. Randomized exploration in reinforcement learning with general value function approximation. In International Conference on Machine Learning, 2021.

[2] Andrea Zanette, David Brandfonbrener, Emma Brunskill, Matteo Pirotta, and Alessandro Lazaric. Frequentist regret bounds for randomized least-squares value iteration. In International Conference on Artificial Intelligence and Statistics, 2020.

---

> ### Author Response · Authors · 2023-11-17
> **Additional result on how to remove the interval constraint on $\delta$ in Theorem 4.2**
>
> Once again, we thank all the reviewers for their insightful comments and engaging questions. We have further updated the paper (Appendix F, Page 41- 45 in revision) with in depth description and theoretical results on how to remove the interval constraint on $\delta$ in Theorem 4.2 by incorporating the optimistic reward sampling scheme proposed in [1] into LMC-LSVI. We call this variant Multi-Sampling LMC-LSVI or MS-LMC-LSVI in short. We believe with this additional detail, the paper further provides a complete picture of Langevin Monte Carlo for linear MDP.
>
> In Appendix F.1 we expanded on Remark 4.3 and described in detail how with minimal modification of LMC-LSVI (Algorithm 1), we can get MS-LMC-LSVI.
>
> In Appendix F.2, we laid out the supporting lemmas used to prove its regret bound. Each supporting lemma has one to one correspondence with supporting lemmas in Appendix B.1 that are used to prove the regret bound for LMC-LSVI (Algorithm 1). Moreover, for most of them we omit the proofs as they are identical to that of corresponding lemmas in Appendix B.1. We provided some details where necessary for clarity. The only notable difference is in Lemma F.5 (corresponding to Lemma B.7 for LMC-LSVI) on Page 43. After following the same proof and conclusion of Lemma B.7, we derive some probabilistic inequalities and apply union bound to complete the proof.
>
> In Appendix F.3, we provide the final regret bound result (Theorem F.7) for MS-LMC-LSVI that holds with probability $1-\delta$ for any $\delta \in (0,1)$. The proof is the same as that of Theorem B.8 with some minor differences at the end when we put everything together using union bound.
>
>
> [1] Haque Ishfaq, Qiwen Cui, Viet Nguyen, Alex Ayoub, Zhuoran Yang, Zhaoran Wang, Doina Precup, and Lin Yang. Randomized exploration in reinforcement learning with general value function approximation. In International Conference on Machine Learning, 2021.

---

### Public Comment · ~Yingru_Li1 · 2023-11-24
**Comparison with latest published SOTAs in this direction?**

# Comparison with latest published SOTAs in this direction?
## Deep exploration
For the capability for deep exploration, the HyperDQN [1] is current published STOA. Its performance is much better than BootDQN with randomised prior function.

## Atari Benchmark
HyperDQN [1] also performs much better compared to the baselines the author used in this work. Could you provide some evidence that this algorithm is also better or comparable with HyperDQN?
BBF [2] is currently the STOA value-based method in Atari Benchmark although using a lot of tricks.

[1] Li et al. "HyperDQN: A Randomized Exploration Method for Deep Reinforcement Learning." International Conference on Learning Representations. 2021. Official repository: https://github.com/liziniu/HyperDQN

[2] Schwarzer et al. "Bigger, Better, Faster: Human-level Atari with human-level efficiency." International Conference on Machine Learning. PMLR, 2023. Official repository: https://github.com/google-research/google-research/tree/master/bigger_better_faster

---

### Meta-Review · Area_Chair_ikV3 · 2023-12-03

**Metareview:**

The paper describes a new Thompson sampling algorithm for exploration in RL that leverages Monte Carlo Langevin Dynamics to sample from a broader posterior distribution than Gaussians.  This works advances the state of the art empirically as demonstrated in the experiments and it advances the theory with a state of the art regret bound for the special case of linear reward MDPs.  Unfortunately, the theory does not apply to the practical version of the algorithm, but this is understandable once gradient descent techniques with neural networks are used.  Overall, this is a strong paper.

**Justification For Why Not Higher Score:**

The paper makes important theoretical and empirical contributions, but it simply matches the best regret bounds and the best empirical results.

**Justification For Why Not Lower Score:**

The reviewers unanimously recommend acceptance of the paper.

---

### Decision · Program_Chairs · 2024-01-16

Accept (poster)